# Outcome-Driven Reinforcement Learning via Variational Inference

**Tim G. J. Rudner**[*][†]
University of Oxford

**Vitchyr H. Pong**[*][†]
University of California, Berkeley

**Rowan McAllister**
University of California, Berkeley

**Yarin Gal**
University of Oxford

**Sergey Levine**
University of California, Berkeley

## Abstract

While reinforcement learning algorithms provide automated acquisition of optimal policies, practical application of such methods requires a number of design decisions, such as manually designing reward functions that not only define the task, but also provide sufficient shaping to accomplish it. In this paper, we view reinforcement learning as inferring policies that achieve desired outcomes, rather than as a problem of maximizing rewards. To solve this inference problem, we establish a novel variational inference formulation that allows us to derive a well-shaped reward function which can be learned directly from environment interactions. From the corresponding variational objective, we also derive a new probabilistic Bellman backup operator and use it to develop an off-policy algorithm to solve goal-directed tasks. We empirically demonstrate that this method eliminates the need to hand-craft reward functions for a suite of diverse manipulation and locomotion tasks and leads to effective goal-directed behaviors.

## 1   Introduction

Reinforcement learning (RL) provides an appealing formalism for automated learning of behavioral skills, but requires considerable care and manual design to use in practice. One particularly delicate decision is the design of the reward function, which has a significant impact on the resulting policy but is largely heuristic in practice, often lacks theoretical grounding, can make effective learning difficult, and may lead to reward mis-specification.

To avoid these shortcomings, we propose to circumvent the process of manually specifying a reward function altogether: Instead of framing the reinforcement learning problem as finding a policy that maximizes a heuristically-defined reward function, we express it probabilistically, as inferring a state–action trajectory distribution conditioned on a desired future outcome. By building off of prior work on probabilistic perspectives on RL [10, 23, 35, 46, 47, 58] and goal-directed RL in particular [3, 6, 19, 48], we derive a tractable variational objective, an temporal-difference algorithm that provides a shaping-like effect for effective learning, as well as a reward function that captures the semantics of the underlying decision problem and facilitates effective learning.

We demonstrate that unlike prior works that proposed inference methods for finding policies that achieve desired outcomes [3, 11, 19, 48], the resulting algorithm, Outcome-Driven Actor–Critic (ODAC), is amenable to off-policy learning and applicable to complex, high-dimensional continuous control tasks over finite and infinite horizons. The resulting variational algorithm can be interpreted as an automatic shaping method, where each iteration learns a reward function that automatically provides dense rewards, as we visualize in Figure 1. In tabular settings, ODAC is guaranteed to converge

---

[*]Equal contribution. [†] Corresponding authors: `tim.rudner@cs.ox.ac.uk` and `vitchyr@berkeley.edu`.

35th Conference on Neural Information Processing Systems (NeurIPS 2021).

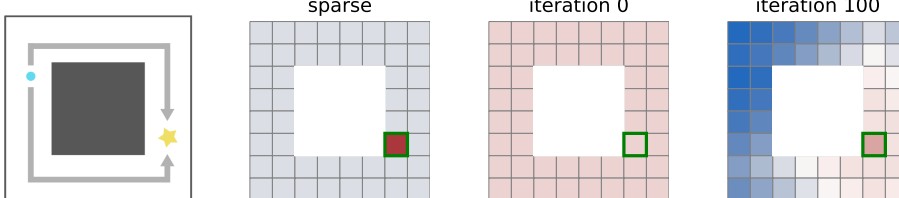

**Figure 1:** Illustration of the shaping effect of the reward function derived from the goal-directed variational inference objective. **Left:** A 2-dimensional grid world with a desired outcome marked by a star. **Center-left** The corresponding sparse reward function provides little shaping. **Center-right:** The reward function derived from our variational inference formulation at initialization. **Right:** The derived reward function after training. We see that the derived reward learns to provide a dense reward signal everywhere in the state space.

to an optimal policy, and in non-tabular settings with linear Gaussian transition dynamics, the derived optimization objective is convex in the policy, facilitating easier learning. In high-dimensional and non-linear domains, our method can be combined with deep neural network function approximators to yield a deep reinforcement learning method that does not require manual specification of rewards, and leads to good performance on a range of benchmark tasks.

**Contributions.** The core contributions of this paper are the probabilistic formulation of a general framework for inferring policies that lead to desired outcomes and the derivation of a variational objective from which we obtain a novel outcome-driven Bellman backup operator. We show that this Bellman backup operator induces a shaping-like effect which ensures a clear and dense learning signal even in the early stages of training. Crucially, unlike heuristic approaches for incorporating shaping often used in standard RL, this "shaping" emerges automatically from variational inference. We demonstrate that the resulting variational objective is a lower bound on the log-marginal likelihood of achieving the outcome given an initial state and that it leads to an off-policy temporal-difference algorithm. We evaluate this algorithm—Outcome-Driven Variational Inference (ODAC)—on a range of reinforcement learning tasks without having to manually specify task-specific reward functions. In our experiments, we find that our method results in significantly faster learning across a variety of robot manipulation and locomotion tasks than alternative approaches.

## 2 Preliminaries

Standard reinforcement learning (RL) addresses reward maximization in a Markov decision process (MDP) defined by the tuple $(\mathcal{S}, \mathcal{A}, p_{\mathbf{S}_0}, p_d, r, \gamma)$ [43, 44], where $\mathcal{S}$ and $\mathcal{A}$ denote the state and action space, respectively, $p_0$ denotes the initial state distribution, $p_d$ is a state transition distribution, $r$ is an immediate reward function, and $\gamma$ is a discount factor. To sample trajectories, an initial state is sampled according to $p_{\mathbf{S}_0}$, and successive states are sampled from the state transition distribution $\mathbf{S}_{t+1} \sim p_d(\cdot \,|\, \mathbf{s}_t, \mathbf{a}_t)$ and actions from a policy $\mathbf{A}_t \sim \pi(\cdot \,|\, \mathbf{s}_t)$. We will write $\mathcal{T}_{0:t} = \{\mathbf{S}_0, \mathbf{A}_0, \mathbf{S}_1, \ldots, \mathbf{S}_t, \mathbf{A}_t\}$ to represent a finite-horizon and $\mathcal{T}_0 \doteq \{\mathbf{A}_t, \mathbf{S}_{t+1}\}_{t=0}^\infty$ to represent an infinite-horizon stochastic state–action trajectory, and write $\boldsymbol{\tau}_{0:t} = \{\mathbf{s}_0, \mathbf{a}_0, \mathbf{s}_1, \ldots, \mathbf{s}_t, \mathbf{a}_t\}$ and $\boldsymbol{\tau}_0 \doteq \{\mathbf{a}_t, \mathbf{s}_{t+1}\}_{t=0}^\infty$ for the respective trajectory realizations. Given a reward function $r : \mathcal{S} \times \mathcal{A} \to \mathbb{R}$ and discount factor $\gamma \in (0, 1)$, the objective in reinforcement learning is to find a policy $\pi$ that maximizes the returns, defined as $\mathbb{E}_{p_\pi} \left[ \sum_{t=0}^\infty \gamma^t r(\mathbf{s}_t, \mathbf{a}_t) \right]$, where $p_\pi$ denotes the distribution of states induced by a policy $\pi$.

**Goal-Conditioned Reinforcement Learning.** In goal-conditioned reinforcement learning [22], which can be considered a special case of the broader class of stochastic boundary value problems [1, 14], the objective is for an agent to reach some pre-specified goal state, $\mathbf{g} \in \mathcal{S}$, so that the policy and reward function introduced above become dependent on the goal and are expressed as $\pi(\mathbf{a} \,|\, \mathbf{s}, \mathbf{g})$ and $r(\mathbf{s}, \mathbf{a}, \mathbf{g})$, respectively. Typically, such a reward function needs to be defined manually, with a common choice being to use a sparse indicator reward $r(\mathbf{s}, \mathbf{g}) = \mathbb{I}\{\mathbf{s} = \mathbf{g}\}$. However, this approach presents a number of challenges both in theory and in practice. From a theoretical perspective, the indicator reward will almost surely equal zero for environments with continuous goal spaces and non-trivial stochastic dynamics. From a practical perspective, such sparse rewards can be slow to learn from, as most transitions provide no reward supervision, while manually designing dense reward functions that provide a better learning signal is time-consuming and often based on heuristics.

In Section 3, we will present a framework that addresses these practical and theoretical considerations by casting goal-conditioned RL as probabilistic inference.

**$Q$-Learning.** Off-policy $Q$-learning algorithms [52] allow training policies from data collected under alternate decision rules by estimating the expected return $Q^\pi$ for a given state–action pair:

$$Q^\pi(\mathbf{s}, \mathbf{a}) \doteq \mathbb{E}_{p_\pi}\left[\sum_{t=0}^{\infty} \gamma^t r(\mathbf{s}_t, \mathbf{a}_t)\Big|\mathbf{S}_0 = \mathbf{s}, \mathbf{A}_0 = \mathbf{a}\right].$$

Crucially, the expected return given a state–action pair can be expressed recursively as

$$Q^\pi(\mathbf{s}, \mathbf{a}) = r(\mathbf{s}, \mathbf{a}) + \gamma\, \mathbb{E}_{p_\pi}[Q^\pi(\mathbf{s}_1, \mathbf{a}_1)\,|\,\mathbf{S}_0 = \mathbf{s}, \mathbf{A}_0 = \mathbf{a}], \tag{1}$$

which makes it possible to estimate the expectation on the right-hand side from single-step transitions. The resulting estimates can then be used to find a policy that results in actions which maximize the expected return $Q^\pi(\mathbf{s}, \mathbf{a})$ for all available state–action pairs.

# 3 Outcome-Driven Reinforcement Learning

In this section, we derive a variational inference objective to infer an approximate posterior policy for achieving desired outcomes. Instead of using the heuristic goal-reaching rewards discussed in Section 2, we will derive a general framework for *inferring actions that lead to desired outcomes* by formulating a probabilistic objective, using the tools of variational inference. As we will show in the following sections, we use this formulation to translate the problem of inferring a policy that leads to a desired outcome into a tractable variational optimization problem, which we show corresponds to an RL problem with a well-shaped, dense reward signal from which the agent can learn more easily.

We start with a warm-up problem that demonstrates how to frame the task of achieving a desired outcome as an inference problem in a simplified setting. We then describe how to extend this approach to more general settings. Finally, we show that the resulting variational objective can be expressed as a recurrence relation, which allows us to derive an outcome-driven variational Bellman operator and prove an outcome-driven probabilistic policy iteration theorem.

## 3.1 Warm-up: Achieving a Desired Outcome at a Fixed Time Step

We first consider a simplified problem, where the desired outcome is to reach a specific state $\mathbf{g} \in \mathcal{S}$ at a specific time step $t^\star$ when starting from initial state $\mathbf{s}_0$. We can think of the starting state $\mathbf{s}_0$ and the desired outcome $\mathbf{g}$ as boundary conditions, and the goal is to learn a stochastic policy that induces a trajectory from $\mathbf{s}_0$ to $\mathbf{g}$. To derive a control law that solves this stochastic boundary value problem,

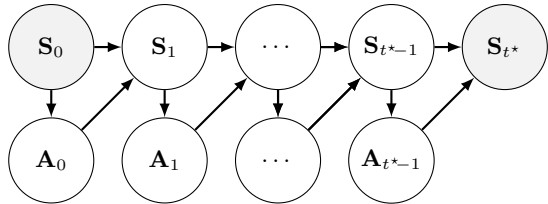

**Figure 2:** A Probabilistic graphical model of a state–action trajectory with observed random variables $\mathbf{S}_0 = \mathbf{s}_0$ and $\mathbf{S}_{t^\star} = \mathbf{g}$

we frame the problem probabilistically, as inferring a state–action trajectory posterior distribution *conditioned on the desired outcome and the initial state*. We will show that, by framing the learning problem this way, we obtain an algorithm for learning outcome-driven policies without needing to manually specify a reward function. We consider a model of the state–action trajectory up to and including the desired outcome $\mathbf{g}$,

$$p_{\tilde{\mathcal{T}}_{0:t}, \mathbf{S}_{t^\star}}(\tilde{\tau}_{0:t}, \mathbf{g}\,|\,\mathbf{s}_0) \doteq p_d(\mathbf{g}\,|\,\mathbf{s}_t, \mathbf{a}_t)p(\mathbf{a}_t\,|\,\mathbf{s}_t)\prod_{t'=0}^{t-1} p_d(\mathbf{s}_{t'+1}\,|\,\mathbf{s}_{t'}, \mathbf{a}_{t'})p(\mathbf{a}_{t'}\,|\,\mathbf{s}_{t'}),$$

where $t^\star \doteq t+1$, $\tilde{\mathcal{T}}_{0:t}$ is the state–action trajectory up to an including $t$ but excluding $\mathbf{S}_0$, $p(\mathbf{a}_t\,|\,\mathbf{s}_t)$ is a conditional action prior, and $p_d(\mathbf{s}_{t+1}\,|\,\mathbf{s}_t, \mathbf{a}_t)$ is the environment's state transition distribution. If the dynamics are simple (e.g., tabular or Gaussian), the posterior over actions can be computed in closed form [3], but we would like to be able to infer outcome-driven posterior policies in any environments, including those where exact inference may be intractable. To do so, we start by expressing posterior inference as the variational minimization problem

$$\min_{q_{\tilde{\mathcal{T}}_{0:t}|\mathbf{S}_0} \in \hat{\mathcal{Q}}} \mathbb{D}_{\mathrm{KL}}(q_{\tilde{\mathcal{T}}_{0:t}|\mathbf{S}_0}(\cdot\,|\,\mathbf{s}_0) \,\|\, p_{\tilde{\mathcal{T}}_{0:t}|\mathbf{S}_0, \mathbf{S}_{t^\star}}(\cdot\,|\,\mathbf{s}_0, \mathbf{g})), \tag{2}$$

where $\mathbb{D}_{\mathrm{KL}}(\cdot \| \cdot)$ is the KL divergence, and $\hat{\mathcal{Q}}$ denotes the variational family over which to optimize. We consider a family of distributions parameterized by a policy $\pi$ and defined by

$$q_{\tilde{\mathcal{T}}_{0:t}|\mathbf{S}_0}(\tilde{\boldsymbol{\tau}}_{0:t} \,|\, \mathbf{s}_0) \doteq \pi(\mathbf{a}_t \,|\, \mathbf{s}_t) \prod_{t'=0}^{t-1} p_d(\mathbf{s}_{t'+1} \,|\, \mathbf{s}_{t'}, \mathbf{a}_{t'}) \pi(\mathbf{a}_{t'} \,|\, \mathbf{s}_{t'}), \qquad (3)$$

where $\pi \in \Pi$, a family of policy distributions, and where $\prod_{t=0}^{t-1} p_d(\mathbf{s}_{t+1} \,|\, \mathbf{s}_t, \mathbf{a}_t)$ is the true action-conditional state transition distribution up to but excluding the state transition at $t^\star - 1$, since $\mathbf{S}_{t^\star} = \mathbf{g}$ is observed. Under this variational family, the inference problem in Equation (2) can be equivalently stated as the problem of maximizing the following objective with respect to the policy $\pi$:

**Proposition 1.** *Given $q_{\tilde{\mathcal{T}}_{0:t}|\mathbf{S}_0}(\tilde{\boldsymbol{\tau}}_{0:t} \,|\, \mathbf{s}_0)$ from Equation (3), any state $\mathbf{s}_0$, termination time $t^\star$, and outcome $\mathbf{g}$, solving Equation (2) is equivalent to maximizing this objective with respect to $\pi \in \Pi$:*

$$\bar{\mathcal{F}}(\pi, \mathbf{s}_0, \mathbf{g}) \doteq \mathbb{E}_{q_{\tilde{\mathcal{T}}_{0:t}|\mathbf{S}_0}(\tilde{\boldsymbol{\tau}}_{0:t} \,|\, \mathbf{s}_0)} \left[ \log p_d(\mathbf{g} \,|\, \mathbf{s}_t, \mathbf{a}_t) - \sum_{t'=0}^{t-1} \mathbb{D}_{\mathrm{KL}}(\pi(\cdot \,|\, \mathbf{s}_{t'}) \,\|\, p(\cdot \,|\, \mathbf{s}_{t'})) \right]. \qquad (4)$$

*Proof.* See Appendix A.1. □

A variational problem of this form—which corresponds to finding a posterior distribution over state–action trajectories—can equivalently be viewed as a reinforcement learning problem:

**Corollary 1.** *The objective in Equation (4) corresponds to KL-regularized reinforcement learning with a time-varying reward function given by $r(\mathbf{s}_{t'}, \mathbf{a}_{t'}, \mathbf{g}, t') \doteq \mathbb{I}\{t' = t\} \log p_d(\mathbf{g} \,|\, \mathbf{s}_{t'}, \mathbf{a}_{t'})$.*

Corollary 1 illustrates how a reward function *emerges automatically* from a probabilistic framing of outcome-driven reinforcement learning problems where the sole specification is which variable ($\mathbf{S}_{t^\star}$) should attain which value ($\mathbf{g}$). In particular, Corollary 1 suggests that we ought to learn the environment's state-transition distribution, and view the log-likelihood of achieving the desired outcome given a state–action pair as a "reward" that can be used for off-policy learning as described in Section 2. Importantly—and unlike in model-based RL—such a transition model would not have to be accurate beyond single-step predictions, as it would not be used for planning (see Appendix B). Instead, $\log p_d(\mathbf{g} \,|\, \mathbf{s}_t, \mathbf{a}_t)$ only needs to be well shaped, which we expect to happen for commonly used model classes. For example, when the dynamics are linear-Gaussian, using a conditional Gaussian model [29] yields a reward function that is quadratic in $\mathbf{S}_{t+1}$, making the objective convex and thus more amenable to optimization.

### 3.2 Outcome-Driven Reinforcement Learning as Variational Inference

Thus far, we assumed that the time at which the outcome is achieved is given. In many problem settings, we do not know (or care) when an outcome is achieved. In this section, we present a variational inference perspective on achieving desired outcomes in settings where *no reward function* and *no termination time* are given, but only a desired outcome is provided. As in the previous section, we derive a variational objective and show that a principled algorithm and reward function emerge automatically when framing the problem as variational inference.

To derive such an objective, we modify the probabilistic model used in the previous section to model that the time at which the outcome is achieved is not known. As before, we define an "outcome" as a point in the state space, but instead of defining the event of "achieving a desired outcome" as a realization $\mathbf{S}_{t^\star} = \mathbf{g}$ for a known $t^\star$, we define it as a realization $\mathbf{S}_{T^\star} = \mathbf{g}$ for an *unknown* termination time $T^\star$ at which the outcome is achieved. Specifically, we model the distribution over the trajectory and the unknown termination time with

$$p_{\tilde{\mathcal{T}}_{0:T}, \mathbf{S}_{T^\star}, T|\mathbf{S}_0}(\tilde{\boldsymbol{\tau}}_{0:t}, \mathbf{g}, t \,|\, \mathbf{s}_0) = p_T(t) p_d(\mathbf{g} \,|\, \mathbf{s}_t, \mathbf{a}_t) p(\mathbf{a}_t \,|\, \mathbf{s}_t) \prod_{t'=0}^{t-1} p_d(\mathbf{s}_{t'+1} \,|\, \mathbf{s}_{t'}, \mathbf{a}_{t'}) p(\mathbf{a}_{t'} \,|\, \mathbf{s}_{t'}), \quad (5)$$

where $p_T(t)$ is the probability of reaching the outcome at $t + 1$. Since the trajectory length is itself a random variable, the joint distribution in Equation (5) is a *transdimensional* distribution defined on $\biguplus_{t=0}^{\infty} \{t\} \times \mathcal{S}^{t+1} \times \mathcal{A}^{t+1}$ [19].

Unlike in the warm-up, the problem of finding an outcome-driven policy that eventually achieves the desired outcome corresponds to finding the posterior distribution over state–action trajectories *and*

the termination time $T$ conditioned on the desired outcome $\mathbf{S}_{T^\star}$ and a starting state. Analogously to Section 3.1, we can express this inference problem variationally as

$$\min_{q_{\tilde{\boldsymbol{\mathcal{T}}}_{0:T},T \mid \mathbf{S}_0} \in \mathcal{Q}} \mathbb{D}_{\mathrm{KL}}(q_{\tilde{\boldsymbol{\mathcal{T}}}_{0:T},T \mid \mathbf{S}_0}(\cdot \mid \mathbf{s}_0) \,\|\, p_{\tilde{\boldsymbol{\mathcal{T}}}_{0:T},T \mid \mathbf{S}_0,\mathbf{S}_{T^\star}}(\cdot \mid \mathbf{s}_0, \mathbf{g})), \tag{6}$$

where $t$ denotes the time immediately *before* the outcome is achieved, $\mathcal{Q}$ denotes the variational family. In general, solving this variational problem in closed form is challenging, but by choosing a variational family $q_{\tilde{\boldsymbol{\mathcal{T}}}_{0:T},T \mid \mathbf{S}_0}(\tilde{\tau}_{0:t}, t \mid \mathbf{s}_0) = q_{\tilde{\boldsymbol{\mathcal{T}}}_{0:T} \mid T,\mathbf{S}_0}(\tilde{\tau}_{0:t} \mid t, \mathbf{s}_0) q_T(t)$, where $q_T$ is a distribution over $T$ in some variational family $\mathcal{Q}_T$ parameterized by

$$q_T(t) = q_{\Delta_{t+1}}(\Delta_{t+1} = 1) \prod_{t'=1}^{t} q_{\Delta_{t'}}(\Delta_{t'} = 0), \tag{7}$$

with Bernoulli random variables $\Delta_t$ denoting the event of "reaching $\mathbf{g}$ at time $t$ given that $\mathbf{g}$ has not yet been reached by time $t-1$," we can equivalently express the variational problem in Equation (6) in a way that is tractable and amenable to off-policy optimization:

**Theorem 1.** *Let $q_T(t)$ and $q_{\tilde{\boldsymbol{\mathcal{T}}}_{0:T} \mid T,\mathbf{S}_0}(\tilde{\tau}_{0:t} \mid t, \mathbf{s}_0)$ be as defined before, and define*

$$V^\pi(\mathbf{s}_t, \mathbf{g}; q_T) \doteq \mathbb{E}_{\pi(\mathbf{a}_t \mid \mathbf{s}_t)}[Q^\pi(\mathbf{s}_t, \mathbf{a}_t, \mathbf{g}; q_T)] - \mathbb{D}_{\mathrm{KL}}(\pi(\cdot \mid \mathbf{s}_t) \,\|\, p(\cdot \mid \mathbf{s}_t)) \tag{8}$$

$$Q^\pi(\mathbf{s}_t, \mathbf{a}_t, \mathbf{g}; q_T) \doteq r(\mathbf{s}_t, \mathbf{a}_t, \mathbf{g}; q_\Delta) + q_{\Delta_{t+1}}(\Delta_{t+1} = 0) \mathbb{E}_{p_d(\mathbf{s}_{t+1} \mid \mathbf{s}_t, \mathbf{a}_t)}[V^\pi(\mathbf{s}_{t+1}, \mathbf{g}; \pi, q_T)] \tag{9}$$

$$r(\mathbf{s}_t, \mathbf{a}_t, \mathbf{g}; q_\Delta) \doteq q_{\Delta_{t+1}}(\Delta_{t+1} = 1) \log p_d(\mathbf{g} \mid \mathbf{s}_t, \mathbf{a}_t) - \mathbb{D}_{\mathrm{KL}}(q_{\Delta_{t+1}} \,\|\, p_{\Delta_{t+1}}). \tag{10}$$

*Then given any initial state $\mathbf{s}_0$ and outcome $\mathbf{g}$,*

$$\mathbb{D}_{\mathrm{KL}}(q_{\tilde{\boldsymbol{\mathcal{T}}}_{0:T},T \mid \mathbf{S}_0}(\cdot \mid \mathbf{s}_0) \,\|\, p_{\tilde{\boldsymbol{\mathcal{T}}}_{0:T},T \mid \mathbf{S}_0,\mathbf{S}_{T^*}}(\cdot \mid \mathbf{s}_0, \mathbf{g})) = -V^\pi(\mathbf{s}_0, \mathbf{g}; q_T) + \log p(\mathbf{g} \mid \mathbf{s}_0),$$

*where $\log p(\mathbf{g} \mid \mathbf{s}_0)$ is independent of $\pi$ and $q_T$ and hence maximizing $V^\pi(\mathbf{s}_0, \mathbf{g}; \pi, q_T)$ is equivalent to minimizing Equation (6).*

*Proof.* See Appendix A.2. $\qquad\qquad\square$

This theorem tells us that the maximizer of $V^\pi(\mathbf{s}_t, \mathbf{g}; q_T)$ is equal to the minimizer of Equation (6). In other words, Theorem 1 presents a variational objective with dense reward functions defined solely in terms of the desired outcome and the environment dynamics, which we can learn directly from environment interactions. It further makes precise that the variational objective, $V^\pi(\mathbf{s}_0, \mathbf{g}; q_T)$, is a lower bound on the log-marginal likelihood, that is, $\log p(\mathbf{g} \mid \mathbf{s}_0) \geq V^\pi(\mathbf{s}_0, \mathbf{g}; q_T)$, where

$$V^\pi(\mathbf{s}_0, \mathbf{g}; q_T) = \mathbb{E}\left[\sum_{t=0}^{\infty} \left(\prod_{t'=1}^{t} q_{\Delta_{t'}}(\Delta_{t'} = 0)\right)\left(r(\mathbf{s}_t, \mathbf{a}_t, \mathbf{g}; q_\Delta) - \mathbb{D}_{\mathrm{KL}}(\pi(\cdot \mid \mathbf{s}_t) \,\|\, p(\cdot \mid \mathbf{s}_t))\right)\right],$$

with the expectation taken with respect to the infinite-horizon trajectory distribution $q_{\tilde{\boldsymbol{\mathcal{T}}}_0 \mid \mathbf{S}_0}(\tilde{\tau}_0 \mid \mathbf{s}_0)$. Thanks to the recursive expression of the variational objective, we can find the optimal variational over $T$ as a function of the current policy and $Q$-function analytically:

**Proposition 2.** *The optimal distribution $q_T^\star$ with respect to Equation (8) is*

$$q_{\Delta_{t+1}}^\star(\Delta_{t+1} = 0; \pi, Q^\pi) = \sigma\left(\Lambda(\mathbf{s}_t, \pi, q_T, Q^\pi) + \sigma^{-1}\left(p_{\Delta_{t+1}}(\Delta_{t+1} = 0)\right)\right), \tag{11}$$

*where*

$$\Lambda(\mathbf{s}_t, \pi, q_T, Q^\pi) \doteq \mathbb{E}_{\pi(\mathbf{a}_{t+1} \mid \mathbf{s}_{t+1}) p_d(\mathbf{s}_{t+1} \mid \mathbf{s}_t, \mathbf{a}_t) \pi(\mathbf{a}_t \mid \mathbf{s}_t)}[Q^\pi(\mathbf{s}_{t+1}, \mathbf{a}_{t+1}, \mathbf{g}; q_T) - \log p_d(\mathbf{g} \mid \mathbf{s}_t, \mathbf{a}_t)]$$

*and $\sigma(\cdot)$ is the sigmoid function, that is, $\sigma(x) = \frac{1}{e^{-x}+1}$ and $\sigma^{-1}(x) = \log \frac{x}{1-x}$.*

*Proof.* See Appendix A.3 $\qquad\qquad\square$

Alternatively, if instead of learning $q_T$ variationally, we fix $q_T$ to the prior $p_T$, we recover the more conventional fixed-discount factor objective [13, 16, 39]:

**Corollary 2.** *Let $q_T = p_T$, assume that $p_T$ is a Geometric distribution with parameter $\gamma \in (0, 1)$. Then the inference problem in Equation (6) of finding a goal-directed variational trajectory distribution simplifies to maximizing the following recursively defined variational objective with respect to $\pi$:*

$$\bar{V}^\pi(\mathbf{s}_0, \mathbf{g}; \gamma) \doteq \mathbb{E}_{\pi(\mathbf{a}_0 \mid \mathbf{s}_0)}[Q(\mathbf{s}_0, \mathbf{a}_0, \mathbf{g}; \gamma)] - \mathbb{D}_{\mathrm{KL}}(\pi(\cdot \mid \mathbf{s}_0) \,\|\, p(\cdot \mid \mathbf{s}_0))), \tag{12}$$

*where*

$$\bar{Q}^\pi(\mathbf{s}_0, \mathbf{a}_0, \mathbf{g}; \gamma) \doteq (1 - \gamma) \log p_d(\mathbf{g} \mid \mathbf{s}_0, \mathbf{a}_0) + \gamma \mathbb{E}_{p_d(\mathbf{s}_1 \mid \mathbf{s}_0, \mathbf{a}_0)}[V(\mathbf{s}_1, \mathbf{g}; \gamma)]. \tag{13}$$

In the next section, we derive a temporal-difference algorithm and discuss how we can learn the $Q$-function in Theorem 1 using off-policy transitions.

# 4 Outcome-Driven Reinforcement Learning

In this section, we show that the variational objective in Theorem 1 is amenable to off-policy learning and that it can be estimated efficiently from single-step transitions. We then describe how to instantiate the resulting outcome-driven algorithm in large environments where function approximation is necessary.

## 4.1 Outcome-Driven Policy Iteration

To develop an outcome-directed off-policy algorithm, we define the following Bellman operator:

**Definition 1.** *Given a function $Q : \mathcal{S} \times \mathcal{A} \times \mathcal{S} \to \mathbb{R}$, define the operator $\mathcal{T}^\pi$ as*

$$\mathcal{T}^\pi Q(\mathbf{s}_t, \mathbf{a}_t, \mathbf{g}; q_T) \doteq r(\mathbf{s}_t, \mathbf{a}_t, \mathbf{g}; q_\Delta) + q_{\Delta_{t+1}}(\Delta_{t+1} = 0) \, \mathbb{E}_{p_d(\mathbf{s}_{t+1} \mid \mathbf{s}_t, \mathbf{a}_t)} \left[ V(\mathbf{s}_{t+1}, \mathbf{g}; q_T) \right], \quad (14)$$

*where $r(\mathbf{s}_t, \mathbf{a}_t, \mathbf{g}; q_\Delta)$ is from Theorem 1 and*

$$V(\mathbf{s}_t, \mathbf{g}; q_T) \doteq \mathbb{E}_{\pi(\mathbf{a}_t \mid \mathbf{s}_t)} \left[ Q(\mathbf{s}_t, \mathbf{a}_t, \mathbf{g}; q_T) \right] + \mathbb{D}_{\mathrm{KL}}(\pi(\cdot \mid \mathbf{s}_t) \,\|\, p(\cdot \mid \mathbf{s}_t)). \quad (15)$$

Unlike the standard Bellman operator, the above operator has a varying weight factor $q_{\Delta_{t+1}}(\Delta_{t+1} = 0)$, with the optimal weight factor given by Equation (11). From Equation (11), we see that as the outcome likelihood $p_d(\mathbf{g} \mid \mathbf{s}, \mathbf{a})$ becomes large relative to the $Q$-function, the weight factor automatically adjusts the target to rely more on the rewards.

Below, we show that repeatedly applying the operator $\mathcal{T}^\pi$ (policy evaluation) and optimizing $\pi$ with respect to $Q^\pi$ (policy improvement) converges to a policy that maximizes the objective in Theorem 1.

**Theorem 2.** *Assume MDP is ergodic and $|\mathcal{A}| < \infty$.*

1. *Outcome-Driven Policy Evaluation (ODPE): Given policy $\pi$ and a function $Q^0 : \mathcal{S} \times \mathcal{A} \times \mathcal{S} \to \mathbb{R}$, define $Q^{i+1} = \mathcal{T}^\pi Q^i$. Then the sequence $Q^i$ converges to the lower bound in Theorem 1.*

2. *Outcome-Driven Policy Improvement (ODPI): The policy*

$$\pi^+ = \arg\max_{\pi' \in \Pi} \{ \mathbb{E}_{\pi'(\mathbf{a}_t \mid \mathbf{s}_t)} \left[ Q^\pi(\mathbf{s}_t, \mathbf{a}_t, \mathbf{g}; q_T) \right] - \mathbb{D}_{\mathrm{KL}}(\pi'(\cdot \mid \mathbf{s}_t) \,\|\, p(\cdot \mid \mathbf{s}_t) \} \quad (16)$$

   *and the variational distribution over $T$ defined in Equation (11) improve the variational objective, that is, $\mathcal{F}(\pi^+, q_T, \mathbf{s}_0) \geq \mathcal{F}(\pi, q_T, \mathbf{s}_0)$ and $\mathcal{F}(\pi, q_T^+, \mathbf{s}_0) \geq \mathcal{F}(\pi, q_T, \mathbf{s}_0)$ for all $\mathbf{s}_0, \pi, q_T$.*

3. *Alternating between ODPE and ODPI converges to a policy $\pi^\star$ and a variational distribution over $T$, $q_T^\star$, such that $Q^{\pi^\star}(\mathbf{s}, \mathbf{a}, \mathbf{g}; q_T^\star) \geq Q^\pi(\mathbf{s}, \mathbf{a}, \mathbf{g}; q_T)$ for all $(\pi, q_T) \in \Pi \times \mathcal{Q}_T$ and any $(\mathbf{s}, \mathbf{a}) \in \mathcal{S} \times \mathcal{A}$.*

*Proof.* See Appendix A.4. □

This result tells us that alternating between applying the outcome-driven Bellman operator in Definition 1 and optimizing the bound in Theorem 1 using the resulting $Q$-function, which can equivalently be viewed as expectation maximization, will lead to a policy that induces an outcome-driven trajectory and solves the inference problem in Equation (6). As we discuss in Appendix A.4, this implies that Variational Outcome-Driven Policy Iteration is theoretically at least as good as or better than standard policy iteration for KL-regularized objectives.

## 4.2 Outcome-Driven Actor–Critic (ODAC)

We now build on previous sections to develop a practical algorithm that handles large and continuous domains. In such domains, the expectation in the Bellman operator in Definition 1 is intractable, and so we approximate the policy $\pi_\theta$ and $Q$-function $Q_\phi$ with neural networks parameterized by parameters $\theta$ and $\phi$, respectively. We train the $Q$-function to minimize

$$\mathcal{F}_Q(\phi) = \mathbb{E}\left[ \left( Q_\phi(\mathbf{s}, \mathbf{a}, \mathbf{g}) - (r(\mathbf{s}, \mathbf{a}, \mathbf{g}; q_\Delta) + q_{\Delta_t}(\Delta_t = 0) \hat{V}(\mathbf{s}', \mathbf{g})) \right)^2 \right], \quad (17)$$

where the expectation is taken with respect to $(\mathbf{s}, \mathbf{a}, \mathbf{g}, \mathbf{s}')$ sampled from a replay buffer, $\mathcal{D}$, of data collected by a policy. We approximate the $\hat{V}$-function using a target $Q$-function $Q_{\bar{\phi}}$:

---

**Algorithm 1** ODAC: Outcome-Driven Actor–Critic

---

1: Initialize policy $\pi_\theta$, replay buffer $\mathcal{R}$, $Q$-function $Q_\phi$, and dynamics model $p_\psi$.
2: **for** iteration $i = 1, 2, ...$ **do**
3:      Collect on-policy samples to add to $\mathcal{R}$ by sampling $\mathbf{g}$ from environment and executing $\pi$.
4:      Sample batch $(\mathbf{s}, \mathbf{a}, \mathbf{s}', \mathbf{g})$ from $\mathcal{R}$.
5:      Compute approximate reward and optimal weights with Equation (20) and Equation (11).
6:      Update $Q_\phi$ with Equation (17), $\pi_\theta$ with Equation (18), and $p_\psi$ with Equation (19).
7: **end for**

---

$\hat{V}(\mathbf{s}', \mathbf{g}) \approx Q_{\bar{\phi}}(\mathbf{s}', \mathbf{a}', \mathbf{g}) - \log \pi(\mathbf{a}' \mid \mathbf{s}'; \mathbf{g})$, where $\mathbf{a}'$ are samples from the amortized variational policy $\pi_\theta(\cdot \mid \mathbf{s}'; \mathbf{g})$. We further assume a uniform prior policy $p(\cdot \mid \mathbf{s}_t)$ in all of our experiments. The parameters $\bar{\phi}$ slowly track the parameters of $\phi$ at each time step via the standard update $\bar{\phi} \leftarrow \tau\bar{\phi} + (1 - \tau)\phi$ [27]. We then train the policy to maximize the approximate $Q$-function by performing gradient descent on

$$\mathcal{F}_\pi(\theta) = -\mathbb{E}_{\mathbf{s}\sim\mathcal{D}, \mathbf{a}\sim\pi_\theta(\cdot\mid\mathbf{s};\mathbf{g})}\left[Q_\phi(\mathbf{s}, \mathbf{a}, \mathbf{g}) - \log \pi_\theta(\mathbf{a}\mid\mathbf{s};\mathbf{g})\right]. \tag{18}$$

We estimate $\hat{q}_{\Delta_{t+1}}(\Delta_{t+1} = 0)$ with a Monte Carlo estimate of Equation (11) obtained via a single Monte Carlo sample $(\mathbf{s}, \mathbf{a}, \mathbf{s}', \mathbf{a}', \mathbf{g})$ from the replay buffer. In practice, a value of $q_{\Delta_{t+1}}(\Delta_{t+1} = 0) = 1$ can lead to numerical instabilities with bootstrapping, and so we also upper bound the estimated $q_{\Delta_{t+1}}(\Delta_{t+1} = 0)$ by the prior distribution $p_{\Delta_{t+1}}(\Delta_{t+1} = 0)$.

To compute the rewards, we need to compute the likelihood of achieving the desired outcome. If the transition dynamics are unknown, we learn a dynamics model from environment interactions by training a neural network $p_\psi$ that parameterizes the mean and scale of a factorized Laplace distribution. We train this model by maximizing the log-likelihood of the data collected by the policy,

$$\mathcal{F}_p(\psi) = \mathbb{E}_{(\mathbf{s},\mathbf{a},\mathbf{s}')\sim\mathcal{D}}[\log p_\psi(\mathbf{s}' \mid \mathbf{s}, \mathbf{a})], \tag{19}$$

and use it to compute the rewards

$$\hat{r}(\mathbf{s}_t, \mathbf{a}_t, \mathbf{g}; q_\Delta) \doteq \hat{q}_{\Delta_{t+1}}(\Delta_{t+1} = 1)\log p_\psi(\mathbf{g} \mid \mathbf{s}_t, \mathbf{a}_t) - \mathbb{D}_{\mathrm{KL}}(q_{\Delta_t} \| p_{\Delta_t}). \tag{20}$$

The complete algorithm is presented in Algorithm 1 and consists of alternating between collecting data via policy $\pi$ and minimizing Equations 17, 18, and 19 via gradient descent. This algorithm alternates between approximating the lower bound in Equation (8) by repeatedly applying the outcome-driven Bellman operator to an approximate $Q$-function, and maximizing this lower bound by performing approximate policy optimization on Equation (18).

## 5 Related Work

Our problem definition is related to goal-conditioned RL, where desired outcomes are often defined in terms of an exact goal-equality indicator [22, 40]. Unlike in our work, goal-conditioned RL typically requires specifying a reward function reflecting the desired outcome. The most natural choice for a reward function in this setting is an exact goal–equality indicator, which gives non-zero reward whenever an outcome is reached. However, this type of reward function makes learning difficult, since the associated reward signal is sparse at best and impossible to attain at worst (in continuous state spaces).

To overcome this limitation, prior work has proposed heuristics for creating dense reward functions, such as the Euclidean distance or a thresholded distance to a goal [2, 26, 28, 32, 33, 41], or estimating auxiliary metrics to encourage learning, such as the mutual information or time between states and goal [9, 18, 34, 50, 51]. In contrast, a dense, generally-applicable reward function results automatically from our variational objective. In Section 6, we demonstrate that this reward function is substantially easier to optimize than sparse rewards, and that it removes the need to choose arbitrary thresholds or distance metrics needed in alternative approaches.

Several prior works cast RL and control as probabilistic inference [10, 11, 19, 25, 35, 42, 45, 58] or KL divergence minimization [24, 31], but with the aim of reformulating standard reward-based RL, assuming that the reward function is given. In other words, these prior works similarly study

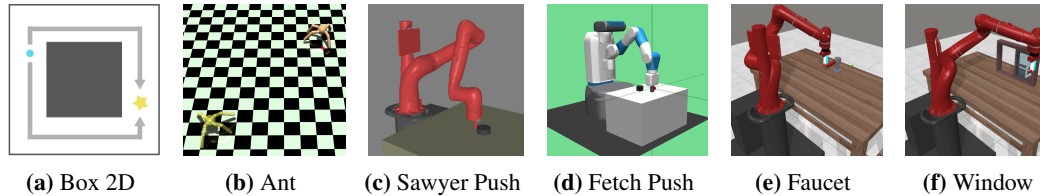

| **(a)** Box 2D | **(b)** Ant | **(c)** Sawyer Push | **(d)** Fetch Push | **(e)** Faucet | **(f)** Window |

**Figure 3:** From left to right, we evaluate on: a 2D environment in which an agent must move around a box, a locomotion task in which a quadruped robot must match a location and pose (yellow), and four manipulation tasks in which the robot must push objects, rotate faucet valve, or open a window.

how optimizing $r_{\text{manual}}(s, a) + \mathcal{H}(\pi)$ corresponds to solving a probabilistic inference problem where $r_{\text{manual}}$ is used to define the outcome's log-likelihood function. However, these prior methods assume that $r_{\text{manual}}$ is provided ex ante. In contrast, our work removes the need to manually specify a task-specific reward or likelihood function, and instead derives both an objective for learning an environment-specific likelihood function and a learning algorithm from the same inference problem.

Past work has also studied control as probabilistic inference in the context of reaching a goal or desired outcome [3, 11, 19, 47]. Toussaint et al. [48] and Hoffman et al. [19] focus on exact inference methods that require time-varying tabular or time-varying Gaussian value functions. In contrast, we propose a variational inference method that eliminates the need to train a time-varying value function, and enables us to use expressive neural networks to represent an approximate value function, making our method applicable to high-dimensional, continuous, and non-linear domains. Unlike the approach in Attias [3], our formulation is not constrained to fixed-horizon settings, obtains a closed-loop rather than open-loop policy, and is applicable to non-tabular dynamic models. More recently, Fu et al. [11] proposed a probabilistic inference method for solving the unknown time-step formulation, but required on-policy trajectory samples. In contrast, we derive an off-policy method by introducing a variational distribution $q_T$ over the time when the outcome is reached.

Lastly, the closely related problem of finding control laws that allow agents to move from an initial state to some desired goal state while incurring minimal cost has been studied in the stochastic control literature [5, 15, 20, 54]. Rawlik et al. [36] consider a continuous-time setting and propose an expectation maximization algorithm that assumes linear Gaussian dynamics, while more recent work has explored finding control laws for non-linear stochastic system dynamics [37, 55]. In contrast to this strand of research, our framework considers a discrete-time setting and does not make assumptions or assumes knowledge of the system dynamics but only requires the ability to interact with the environment to learn an outcome-driven policy.

## 6   Empirical Evaluation

Our experiments compare ODAC to prior methods for learning goal-conditioned policies and evaluate how the components of our variational objective impact the performance of the method. Specifically, we compare ODAC to prior methods on a wide range of manipulation and locomotion tasks that require achieving a desired outcome. To answer the second question, we conduct several ablation studies and visualize the behavior of $q_{\Delta_{t+1}}(\Delta_{t+1} = 0)$. In our experiments, we use a uniform action prior $p(\mathbf{a})$ and the time prior $p_T$ is geometric with parameter 0.01, that is, $p_{\Delta_{t+1}}(\Delta_{t+1} = 0) = 0.99$. We begin by describing prior methods and environments used for the experiments.

### 6.1   Learning to Achieve Desired Outcomes

To avoid over-fitting to any one setting, we compare to these methods across several different robot morphologies and tasks, all illustrated in Figure 3 of the appendix.

**Environments.** We compare ODAC to prior work on a simple 2D navigation task, in which an agent must take non-greedy actions to move around a box, as well as the *Ant*, *Sawyer Push*, and *Fetch Push* simulated robot domains, which have each been studied in prior work on reinforcement learning for reaching goals [2, 28, 30, 34, 41]. For the Ant and Sawyer tasks, desired outcomes correspond to full states (that is, desired positions and joints). For the Fetch task, we use the same goal representation as in prior work [2] and only represent $\mathbf{g}$ with the position of the object. Lastly, we demonstrate the

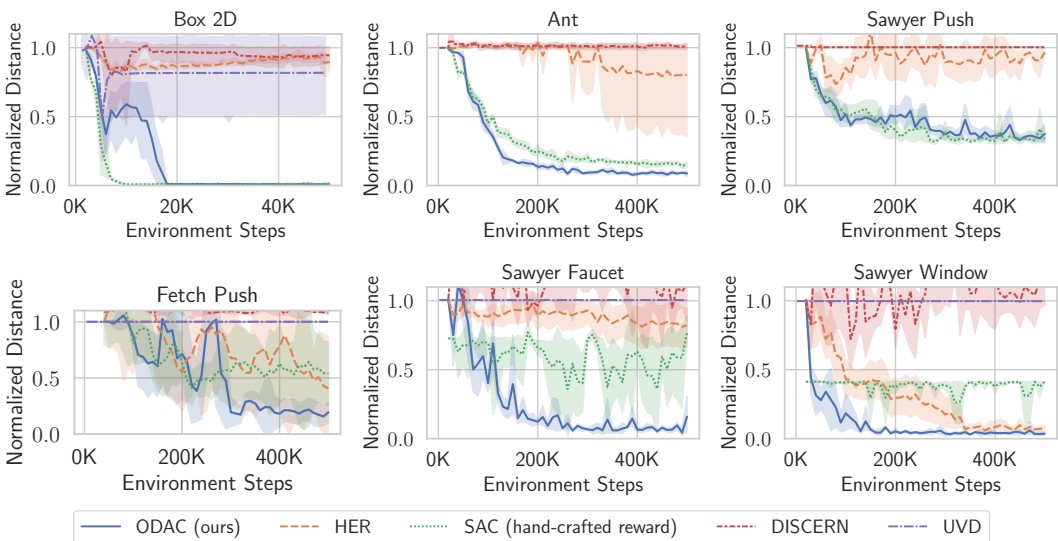

**Figure 4:** Learning curves showing final distance vs. environment steps across all six environments. Only ODAC consistently performs well on all six tasks. Prior methods struggle to learn, especially in the absence of uniform goal sampling. See text for details.

feasibility of replacing manually designed rewards with our outcome-driven paradigm by evaluating the methods on the *Sawyer Window* and *Sawyer Faucet* tasks from the MetaWorld benchmark [56]. These tasks come with manually designed reward functions, which we replace by simply specify a desired outcome g. We plot the mean and standard deviation of the final Euclidean distance to the desired outcome across four random seeds. We normalize the distance to be 1 at the start of training. For further details, see Appendix C.1.

**Goal Sampling.**    In all tasks, rather than assuming that we are able to perform oracle goal sampling (which can be difficult or even impossible in complex real-world environments), a fixed desired outcome is commanded as the exploration goal in each episode. During training, the goals are relabeled using the future-style relabeling scheme from Andrychowicz et al. [2]. Unlike oracle goal sampling, this approach does not assume knowledge of the set of admissible states in the environment, but is more realistic and presents a more challenging exploration problem. To challenge the methods, we choose the desired goal to be far from the starting state.

**Baselines and Prior Work.**    We compare our method to hindsight experience replay (HER) [2], a goal-conditioned method, where the learner receives a reward of $-1$ if it is within an $\epsilon$ distance from the goal, and 0 otherwise, universal value density estimation (UVD) [41], which also uses sparse rewards as well as a generative model of the future occupancy measure to estimate the $Q$-values, and DISCERN [51], which learns a reward function by training a discriminator and using a clipped log-likelihood as the reward. Lastly, we include an oracle Soft Actor–Critic (SAC) baseline that uses a manually designed reward. For the MetaWorld tasks, this baseline uses the benchmark reward for each task. For the remaining environments, this baseline uses the Euclidean distance between the agent's current and the desired outcome for the reward.

**Results.**    In Figure 4, we see that ODAC outperforms virtually every method on all tasks, consistently learning faster and often reaching a final distance that is orders of magnitude closer to the desired outcome. The only exception is that the hand-crafted reward learns slightly faster on the 2D task, but this gap is closed within a few ten thousand steps.

## 6.2   Ablation Study on the Effect of a Variational Discount Factor

Next, we study the importance of the dynamic discount factor $q_{\Delta_{t+1}}(\Delta_{t+1} = 0)$ and the sensitivity of our method to the dynamics model. On all tasks, we evaluate the performance when the posterior exactly matches the prior, that is, $q_{\Delta_{t+1}}(\Delta_{t+1} = 0) = 0.99$ (labeled "fixed $q_T$" in Table 1). Our

analysis in Appendix A.3 suggests that this setting is sub-optimal, and this ablation empirically evaluate its benefits. We also measure how the algorithm's performance depends on the accuracy of the learned dynamics model used for the reward in ODAC. To do this, we evaluate ODAC with the dynamics model fixed to a multivariate Laplace distribution with a fixed variance, centered at the previous state (labeled "fixed $\hat{p}_d$" in Table 1). This ablation represents an extremely crude model, and good performance with such a model would indicate that our method does not depend on obtaining an particularly accurate model.

In Table 1, we see that fixing the distribution $q_T$ to the prior $p_T$ as described in Corollary 3 deteriorates performance, and that using a learned or fixed model both perform relatively well. These results suggest that the derived optimal variational distribution $q^\star_{\Delta_{t+1}}(\Delta_{t+1} = 0)$ given in Proposition 2 is better not only in theory but also in practice, and that ODAC is not sensitive to the accuracy of the dynamics model. Moreover,

**Table 1:** Ablation results, showing mean final normalized distance ($\times 100$) at the end of training across 4 seeds. Best mean is in bold and standard error in parentheses. ODAC is not sensitive to the dynamics models $\hat{p}_d$ but benefits from the dynamic $q_T$ variant.

| Env | ODAC | fixed $\hat{p}_d$ | fixed $q_T$ | fixed $q_T$, $\hat{p}_d$ |
|---|---|---|---|---|
| 2D | 1.7 (1.20) | 1.2 (0.14) | 1.0 (0.24) | 1.3 (0.29) |
| Ant | 9 (0.48) | 11 (0.57) | 12 (0.41) | 13 (0.20) |
| Push | 35 (2.7) | 34 (1.5) | 37 (1.5) | 38 (3.1) |
| Fetch | 19 (6) | 15 (3) | 53 (13) | 66 (15) |
| Window | 5.4 (0.62) | 5.0 (0.62) | 7.9 (0.71) | 6.0 (0.12) |
| Faucet | 13 (4.2) | 15 (3.3) | 37 (8.3) | 38 (7.2) |

we note that a more expressive dynamics model—which could lead to a tighter variational bound— may not necessarily lead to a better variational policy if the functional form of the log-density under the dynamics model does not also provide favorable shaping.

In Appendix B, we provide the full learning curves for this ablation study and present further experiments. For example, we compare ODAC to a variant in which we use the learned dynamics model for model-based planning. We find that using the dynamics model only to compute rewards significantly outperforms the variant where it is used for both computing rewards and model-based planning. This result suggests that ODAC does not require learning a dynamics model that is accurate enough for planning, and that the derived Bellman updates are sufficient for obtaining policies that can achieve desired outcomes. In Figure 6, we also visualize $q_T$ and find that as the policy reaches an irrecoverable state, $q_{\Delta_{t+1}}(\Delta_{t+1} = 0)$ drops in value, suggesting that ODAC *automatically* learns a dynamic discount factor that terminates an episode when an irrecoverable state is reached.

## 7   Conclusion

We proposed a probabilistic approach for achieving desired outcomes in settings where no reward function and no termination condition are given. We showed that by framing the problem of achieving desired outcomes as variational inference, we can derive an off-policy temporal-difference algorithm, a reward function learnable from environment interactions, and a novel Bellman backup that contains a state–action dependent dynamic discount factor for the reward and bootstrap term.

Our experimental results demonstrated that the resulting algorithm, ODAC, leads to efficient outcome-driven approaches to RL. While ODAC requires choosing a dynamics model, we found that it works well even for simple dynamics models and believe that the use of more sophisticated dynamics models that incorporate epistemic uncertainty [7] or domain-specific structure [8, 38, 49, 57] is a promising avenue for future research.

## Acknowledgments and Disclosure of Funding

We thank Marvin Zhang, Michael Janner, Abhishek Gupta, and various RAIL and OATML students for their discussions and feedback on early drafts of this paper. Tim G. J. Rudner is funded by the Rhodes Trust, by a Qualcomm Innovation Fellowship, and by the Engineering and Physical Sciences Research Council (EPSRC). This research was further supported by the Alan Turing Institute, the National Science Foundation, the DARPA Assured Autonomy Program, and ARL DCIST CRA W911NF-17-2-0181.

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
