# Supplementary Material

## Table of Contents

## Appendix A   Proofs & Derivations

### A.1   Finite- and Infinite-Horizon Variational Objectives

In this section, we present detailed derivations and proofs for the results in Sections 3.1 and Section 3.2.

**Proposition 1** (Fixed-Time Outcome-Driven Variational Objective)**.** *Let* $q_{\bar{\mathcal{T}}_{0:t}|\mathbf{S}_0}(\tilde{\tau}_{0:t}\,|\,\mathbf{s}_0)$ *be as defined in Equation* (3)*. Then, given any initial state* $\mathbf{s}_0$*, termination time* $t^\star$*, and outcome* $\mathbf{g}$*,*

$$\mathbb{D}_{\mathrm{KL}}(q_{\bar{\mathcal{T}}_{0:t}|\mathbf{S}_0}(\cdot\,|\,\mathbf{s}_0)\,\|\,p_{\bar{\mathcal{T}}_{0:t}|\mathbf{S}_0,\mathbf{S}_{t^\star}}(\cdot\,|\,\mathbf{s}_0,\mathbf{g})) = \log p(\mathbf{g}\,|\,\mathbf{s}_0) - \bar{\mathcal{F}}(\pi,\mathbf{s}_0,\mathbf{g}), \tag{A.1}$$

*where*

$$\bar{\mathcal{F}}(\pi,\mathbf{s}_0,\mathbf{g}) \doteq \mathbb{E}_{q_{\bar{\mathcal{T}}_{0:t}|\mathbf{S}_0}(\tilde{\tau}_{0:t}\,|\,\mathbf{s}_0)}\left[\log p_d(\mathbf{g}\,|\,\mathbf{s}_t,\mathbf{a}_t) - \sum_{t'=0}^{t-1}\mathbb{D}_{\mathrm{KL}}(\pi(\cdot\,|\,\mathbf{s}_{t'})\,\|\,p(\cdot\,|\,\mathbf{s}_{t'}))\right], \tag{A.2}$$

*and since* $\log p(\mathbf{g}|\mathbf{s}_0)$ *is constant in* $\pi$*,*

$$\arg\min_{\pi\in\Pi}\mathbb{D}_{\mathrm{KL}}(q_{\bar{\mathcal{T}}_{0:t}|\mathbf{S}_0}(\cdot\,|\,\mathbf{s}_0)\,\|\,p_{\bar{\mathcal{T}}_{0:t}|\mathbf{S}_0,\mathbf{S}_{t^\star}}(\cdot\,|\,\mathbf{s}_0,\mathbf{g})) = \arg\max_{\pi\in\Pi}\bar{\mathcal{F}}(\pi,\mathbf{s}_0,\mathbf{g}). \tag{A.3}$$

*Proof.*  To find an approximation to the posterior $p_{\bar{\mathcal{T}}_{0:t}|\mathbf{S}_0,\mathbf{S}_{t^\star}}(\cdot\,|\,\mathbf{s}_0,\mathbf{g})$, we can use variational inference. To do so, we consider the trajectory distribution under $p_{\bar{\mathcal{T}}_{0:t}|\mathbf{S}_0,\mathbf{S}_{t^\star}}(\cdot\,|\,\mathbf{s}_0,\mathbf{g})$, which by Bayes' Theorem is given by

$$p_{\bar{\mathcal{T}}_{0:t}|\mathbf{S}_0,\mathbf{S}_{t^\star}}(\tilde{\tau}_{0:t}\,|\,\mathbf{s}_0,\mathbf{g}) = \frac{p_d(\mathbf{g}\,|\,\mathbf{s}_t,\mathbf{a}_t)p(\mathbf{a}_t\,|\,\mathbf{s}_t)\prod_{t'=0}^{t-1}p_d(\mathbf{s}_{t'+1}\,|\,\mathbf{s}_t,\mathbf{a}_t)p(\mathbf{a}_{t'}\,|\,\mathbf{s}_{t'})}{p(\mathbf{g}\,|\,\mathbf{s}_0)}, \tag{A.4}$$

where  $t = t^\star - 1$,  and  we  denote  the  state–action  trajectory  realization  from  action  $\mathbf{a}_0$  to  $\mathbf{a}_t$  by $\tilde{\tau}_{0:t} \doteq \{\mathbf{a}_0,\mathbf{s}_1,\mathbf{a}_1,...,\mathbf{s}_t,\mathbf{a}_t\}$. Inferring an approximation to the posterior distribution $p_{\bar{\mathcal{T}}_{0:t}|\mathbf{S}_0,\mathbf{S}_{t^\star}}(\cdot\,|\,\mathbf{s}_0,\mathbf{g})$ then becomes equivalent to finding a variational distribution $q_{\bar{\mathcal{T}}_{0:T}|\mathbf{S}_0}(\cdot\,|\,\mathbf{s}_0)$, which induces a trajectory distribution $q_{\bar{\mathcal{T}}_{0:t}|\mathbf{S}_0}(\cdot\,|\,\mathbf{s}_0)$ that minimizes the KL divergence from $q_{\bar{\mathcal{T}}_{0:t}|\mathbf{S}_0}(\cdot\,|\,\mathbf{s}_0)$ to $p_{\bar{\mathcal{T}}_{0:t}|\mathbf{S}_0,\mathbf{S}_{t^\star}}(\cdot\,|\,\mathbf{s}_0,\mathbf{g})$:

$$\min_{q\in\mathcal{Q}}\mathbb{D}_{\mathrm{KL}}(q_{\bar{\mathcal{T}}_{0:t}|\mathbf{S}_0}(\cdot\,|\,\mathbf{s}_0)\,\|\,p_{\bar{\mathcal{T}}_{0:t}|\mathbf{S}_0,\mathbf{S}_{t^\star}}(\cdot\,|\,\mathbf{s}_0,\mathbf{g})). \tag{A.5}$$

If we find a distribution $q_{\tilde{\mathcal{T}}_{0:t}|\mathbf{S}_0}(\cdot\,|\,\mathbf{s}_0)$ for which the resulting KL divergence is zero, then $q_{\tilde{\mathcal{T}}_{0:t}|\mathbf{S}_0}(\cdot\,|\,\mathbf{s}_0)$ is the exact posterior. If the KL divergence is positive, then $q_{\tilde{\mathcal{T}}_{0:t}|\mathbf{S}_0}(\cdot\,|\,\mathbf{s}_0)$ is an approximate posterior. To solve the variational problem in Equation (A.5), we can define a factorized variational family

$$q_{\tilde{\mathcal{T}}_{0:t}|\mathbf{S}_0}(\tilde{\tau}_{0:t}\,|\,\mathbf{s}_0) \doteq \pi(\mathbf{a}_t\,|\,\mathbf{s}_t)\prod_{t'=0}^{t-1} q_{\mathbf{S}_{t'+1}|\mathbf{S}_{t'},\mathbf{A}_{t'}}(\mathbf{s}_{t'+1}\,|\,\mathbf{s}_{t'},\mathbf{a}_{t'})\pi(\mathbf{a}_{t'}\,|\,\mathbf{s}_{t'}), \tag{A.6}$$

where $\mathbf{A}_{0:t}$ and $\mathbf{S}_{1:t}$ are latent variables over which to infer an approximate posterior distribution, and the product is from $t = 0$ to $t = t^\star - 1$ to exclude the conditional distribution over the (observed) state $\mathbf{S}_{t+1} = \mathbf{g}$ from the variational distribution.

Returning to the variational problem in Equation (A.5), we can now write

$$\begin{aligned}
&\mathbb{D}_{\mathrm{KL}}(q_{\tilde{\mathcal{T}}_{0:t}|\mathbf{S}_0}(\cdot\,|\,\mathbf{s}_0)\,\|\,p_{\tilde{\mathcal{T}}_{0:t}|\mathbf{S}_0,\mathbf{S}_{t^\star}}(\cdot\,|\,\mathbf{s}_0,\mathbf{g}))\\
&= \int_{\mathcal{A}^{t+1}}\int_{\mathcal{S}^t} q_{\tilde{\mathcal{T}}_{0:t}|\mathbf{S}_0}(\tilde{\tau}_{0:t}\,|\,\mathbf{s}_0)\log\frac{q_{\tilde{\mathcal{T}}_{0:t}|\mathbf{S}_0}(\tilde{\tau}_{0:t}\,|\,\mathbf{s}_0)}{p_{\tilde{\mathcal{T}}_{0:t}|\mathbf{S}_0,\mathbf{S}_{t^\star}}(\tilde{\tau}_{0:t}|\mathbf{s}_0,\mathbf{g})}\,d\mathbf{s}_{1:t}d\mathbf{a}_{0:t}\\
&= -\bar{\mathcal{F}}(\pi,\mathbf{s}_0,\mathbf{g}) + \log p(\mathbf{g}|\mathbf{s}_0),
\end{aligned} \tag{A.7}$$

where

$$\bar{\mathcal{F}}(\pi,\mathbf{s}_0,\mathbf{g})$$

$$\doteq \mathbb{E}_{q_{\tilde{\mathcal{T}}_{0:t}|\mathbf{S}_0}(\tilde{\tau}_{0:t}\,|\,\mathbf{s}_0)}\Bigg[\log p_d(\mathbf{g}\,|\,\mathbf{s}_t,\mathbf{a}_t) + \log p(\mathbf{a}_t\,|\,\mathbf{s}_t) - \log\pi(\mathbf{a}_t\,|\,\mathbf{s}_t)$$

$$+ \sum_{t'=0}^{t-1}\log p(\mathbf{a}_{t'}\,|\,\mathbf{s}_{t'}) + \log p_d(\mathbf{s}_{t'+1}\,|\,\mathbf{s}_{t'},\mathbf{a}_{t'}) - \log\pi(\mathbf{a}_{t'}\,|\,\mathbf{s}_{t'}) - \log q_{\mathbf{S}_{t'+1}|\mathbf{S}_{t'},\mathbf{A}_{t'}}(\mathbf{s}_{t'+1}\,|\,\mathbf{s}_{t'},\mathbf{a}_{t'})\Bigg] \tag{A.8}$$

and

$$\log p(\mathbf{g}|\mathbf{s}_0) = \log\int_{\mathcal{A}^{t+1}}\int_{\mathcal{S}^t} p_d(\mathbf{g}\,|\,\mathbf{s}_t,\mathbf{a}_t)p_{\tilde{\mathcal{T}}_{0:t}|\mathbf{S}_0}(\tilde{\tau}_{0:t}\,|\,\mathbf{s}_0)\,d\mathbf{s}_{1:t}d\mathbf{a}_{0:t} \tag{A.9}$$

is a log-marginal likelihood. Following Haarnoja et al. [16], we define the variational distribution over next states to be the the true transition dynamics, that is, $q_{\mathbf{S}_{t+1}|\mathbf{S}_t,\mathbf{A}_t}(\mathbf{s}_{t+1}\,|\,\mathbf{s}_t,\mathbf{a}_t) = p_d(\mathbf{s}_{t+1}\,|\,\mathbf{s}_t,\mathbf{a}_t)$, so that

$$q_{\tilde{\mathcal{T}}_{0:t}|\mathbf{S}_0}(\tilde{\tau}_{0:t}\,|\,\mathbf{s}_0) \doteq \pi(\mathbf{a}_t\,|\,\mathbf{s}_t)\prod_{t'=0}^{t-1} p_d(\mathbf{s}_{t'+1}\,|\,\mathbf{s}_{t'},\mathbf{a}_{t'})\pi(\mathbf{a}_{t'}\,|\,\mathbf{s}_{t'}). \tag{A.10}$$

We can then simplify $\bar{\mathcal{F}}(\pi,\mathbf{s}_0,\mathbf{g})$ to

$$\bar{\mathcal{F}}(\pi,\mathbf{s}_0,\mathbf{g}) = \mathbb{E}_{q_{\tilde{\mathcal{T}}_{0:t}|\mathbf{S}_0}(\tilde{\tau}_{0:t}\,|\,\mathbf{s}_0)}\Bigg[\log p_d(\mathbf{g}\,|\,\mathbf{s}_t,\mathbf{a}_t) + \sum_{t'=0}^{t}\mathbb{D}_{\mathrm{KL}}(\pi(\cdot\,|\,\mathbf{s}_{t'})\,\|\,p(\cdot\,|\,\mathbf{s}_{t'}))\Bigg]. \tag{A.11}$$

Since $\log p(\mathbf{g}|\mathbf{s}_0)$ is constant in $\pi$, solving the variational optimization problem in Equation (A.5) is equivalent to maximizing the variational objective with respect to $\pi \in \Pi$, where $\Pi$ is a family of policy distributions. $\square$

**Corollary 1** (Fixed-Time Outcome-Driven Reward Function). *The objective in Equation* (4) *corresponds to KL-regularized reinforcement learning with a time-varying reward function given by*

$$r(\mathbf{s}_{t'},\mathbf{a}_{t'},\mathbf{g},t') \doteq \mathbb{I}\{t' = t\}\log p_d(\mathbf{g}\,|\,\mathbf{s}_{t'},\mathbf{a}_{t'}).$$

*Proof.* Let

$$r(\mathbf{s}_{t'},\mathbf{a}_{t'},\mathbf{g},t') \doteq \mathbb{I}\{t' = t\}\log p_d(\mathbf{g}\,|\,\mathbf{s}_{t'},\mathbf{a}_{t'}) \tag{A.12}$$

and note that the objective

$$\bar{\mathcal{F}}(\pi,\mathbf{s}_0,\mathbf{g}) = \mathbb{E}_{q_{\tilde{\mathcal{T}}_{0:T}|\mathbf{S}_0}(\cdot\,|\,\mathbf{s}_0)}\Bigg[\log p_d(\mathbf{g}\,|\,\mathbf{s}_t,\mathbf{a}_t) + \sum_{t=0}^{t}\mathbb{D}_{\mathrm{KL}}(\pi(\cdot\,|\,\mathbf{s}_t)\,\|\,p(\cdot\,|\,\mathbf{s}_t))\Bigg] \tag{A.13}$$

can equivalently written as

$$\bar{\mathcal{F}}(\pi,\mathbf{s}_0,\mathbf{g}) = \mathbb{E}_{q_{\tilde{\mathcal{T}}_{0:T}|\mathbf{S}_0}(\cdot\,|\,\mathbf{s}_0)}\Bigg[\sum_{t'=0}^{t} r(\mathbf{s}_{t'},\mathbf{a}_{t'},\mathbf{g},t') + \sum_{t'=0}^{t}\mathbb{D}_{\mathrm{KL}}(\pi(\cdot\,|\,\mathbf{s}_{t'})\,\|\,p(\cdot\,|\,\mathbf{s}_{t'}))\Bigg] \tag{A.14}$$

$$= \mathbb{E}_{q_{\tilde{\mathcal{T}}_{0:T}|\mathbf{S}_0}(\cdot\,|\,\mathbf{s}_0)}\Bigg[\sum_{t'=0}^{t} r(\mathbf{s}_{t'},\mathbf{a}_{t'},\mathbf{g},t') + \mathbb{D}_{\mathrm{KL}}(\pi(\cdot\,|\,\mathbf{s}_{t'})\,\|\,p(\cdot\,|\,\mathbf{s}_{t'}))\Bigg], \tag{A.15}$$

which, as shown in Haarnoja et al. [16], can be written in the form of Equation (1). $\square$

**Proposition 3** (Unknown-time Outcome-Driven Variational Objective). *Let* $q_{\tilde{\boldsymbol{\mathcal{T}}}_{0:T},T|\mathbf{S}_0}(\tilde{\boldsymbol{\tau}}_{0:t}, t \,|\, \mathbf{s}_0) = q_{\tilde{\boldsymbol{\mathcal{T}}}_{0:T}|T,\mathbf{S}_0}(\tilde{\boldsymbol{\tau}}_{0:t} \,|\, t, \mathbf{s}_0)q_T(t)$, *let* $q_T(t)$ *be a variational distribution defined on* $t \in \mathbb{N}_0$, *and let* $q_{\tilde{\boldsymbol{\mathcal{T}}}_{0:T}|T,\mathbf{S}_0}(\tilde{\boldsymbol{\tau}}_{0:t} \,|\, t, \mathbf{s}_0)$ *be as defined in Equation* (3). *Then, given any initial state* $\mathbf{s}_0$ *and outcome* $\mathbf{g}$, *we have that*

$$\mathbb{D}_{\mathrm{KL}}(q_{\tilde{\boldsymbol{\mathcal{T}}}_{0:T},T|\mathbf{S}_0}(\cdot \,|\, \mathbf{s}_0) \,\|\, p_{\tilde{\boldsymbol{\mathcal{T}}}_{0:T},T|\mathbf{S}_0,\mathbf{S}_{T^\star}}(\cdot \,|\, \mathbf{s}_0, \mathbf{g})) = \log p(\mathbf{g}|\mathbf{s}_0) - \mathcal{F}(\pi, q_T, \mathbf{s}_0, \mathbf{g}), \tag{A.16}$$

*where*

$$\mathcal{F}(\pi, q_T, \mathbf{s}_0, \mathbf{g})$$
$$\doteq \sum_{t=0}^{\infty} q_T(t) \, \mathbb{E}_{q_{\tilde{\boldsymbol{\mathcal{T}}}_{0:T}|T,\mathbf{S}_0}(\tilde{\boldsymbol{\tau}}_{0:t} \,|\, t,\mathbf{s}_0)} \left[ \log p_d(\mathbf{g} \,|\, \mathbf{s}_t, \mathbf{a}_t) - \mathbb{D}_{\mathrm{KL}}(q_{\tilde{\boldsymbol{\mathcal{T}}}_{0:T}|T,\mathbf{S}_0}(\cdot \,|\, \mathbf{s}_0) \,\|\, p_{\tilde{\boldsymbol{\mathcal{T}}}_{0:T}|T,\mathbf{S}_0}(\cdot \,|\, \mathbf{s}_0)) \right]$$
$$\tag{A.17}$$

*and* $\log p(\mathbf{g} \,|\, \mathbf{s}_0)$ *is constant in* $\pi$ *and* $q_T$.

*Proof.* In general, solving the variational problem

$$\min_{q \in \mathcal{Q}} \mathbb{D}_{\mathrm{KL}}(q_{\tilde{\boldsymbol{\mathcal{T}}}_{0:T},T|\mathbf{S}_0}(\cdot \,|\, \mathbf{s}_0) \,\|\, p_{\tilde{\boldsymbol{\mathcal{T}}}_{0:T},T|\mathbf{S}_0,\mathbf{S}_{T^\star}}(\cdot \,|\, \mathbf{s}_0, \mathbf{g})) \tag{A.18}$$

from Section 3.2 in closed form is challenging, but as in the fixed-time setting, we can take advantage of the fact that, by choosing a variational family parameterized by

$$q_{\tilde{\boldsymbol{\mathcal{T}}}_{0:T}|T,\mathbf{S}_0}(\tilde{\boldsymbol{\tau}}_{0:t} \,|\, t, \mathbf{s}_0) \doteq \pi(\mathbf{a}_t \,|\, \mathbf{s}_t) \prod_{t'=0}^{t-1} p_d(\mathbf{s}_{t'+1} \,|\, \mathbf{s}_{t'}, \mathbf{a}_{t'}) \, \pi(\mathbf{a}_{t'} \,|\, \mathbf{s}_{t'}), \tag{A.19}$$

with $\pi \in \Pi$, we can follow the same steps as in the proof for Proposition 1 and show that given any initial state $\mathbf{s}_0$ and outcome $\mathbf{g}$,

$$\mathbb{D}_{\mathrm{KL}}(q_{\tilde{\boldsymbol{\mathcal{T}}}_{0:T},T|\mathbf{S}_0}(\cdot \,|\, \mathbf{s}_0) \,\|\, p_{\tilde{\boldsymbol{\mathcal{T}}}_{0:T},T|\mathbf{S}_0,\mathbf{S}_{T^\star}}(\cdot \,|\, \mathbf{s}_0, \mathbf{g}))) = \log p(\mathbf{g} \,|\, \mathbf{s}_0) - \mathcal{F}(\pi, q_T, \mathbf{s}_0, \mathbf{g}), \tag{A.20}$$

where

$$\mathcal{F}(\pi, q_T, \mathbf{s}_0, \mathbf{g})$$
$$\doteq \sum_{t=0}^{\infty} q_T(t) \, \mathbb{E}_{q_{\tilde{\boldsymbol{\mathcal{T}}}_{0:T}|T,\mathbf{S}_0}(\tilde{\boldsymbol{\tau}}_{0:t} \,|\, t,\mathbf{s}_0)} \left[ \log p_d(\mathbf{g} \,|\, \mathbf{s}_t, \mathbf{a}_t) - \mathbb{D}_{\mathrm{KL}}(q_{\tilde{\boldsymbol{\mathcal{T}}}_{0:T}|T,\mathbf{S}_0}(\cdot \,|\, \mathbf{s}_0) \,\|\, p_{\tilde{\boldsymbol{\mathcal{T}}}_{0:T}|T,\mathbf{S}_0}(\cdot \,|\, \mathbf{s}_0)) \right],$$
$$\tag{A.21}$$

where $q_{\tilde{\boldsymbol{\mathcal{T}}}_{0:T},T|\mathbf{S}_0}(\tilde{\boldsymbol{\tau}}_{0:t}, t \,|\, \mathbf{s}_0) \doteq q_{\tilde{\boldsymbol{\mathcal{T}}}_{0:T}|T,\mathbf{S}_0}(\tilde{\boldsymbol{\tau}}_{0:t} \,|\, t, \mathbf{s}_0)q_T(t)$, and hence, solving the variational problem in Equation (6) is equivalent to maximizing $\mathcal{F}(\pi, q_T, \mathbf{s}_0, \mathbf{g})$ with respect to $\pi$ and $q_T$. □

## A.2 Recursive Variational Objective & Outcome-Driven Bellman Backup Operator

**Proposition 4** (Factorized Unknown-Time Outcome-Driven Variational Objective). *Let* $q_{\tilde{\boldsymbol{\mathcal{T}}}_{0:T},T}(\tilde{\boldsymbol{\tau}}_{0:t}, t \,|\, \mathbf{s}_0) = q_{\tilde{\boldsymbol{\mathcal{T}}}_{0:T}|T,\mathbf{S}_0}(\tilde{\boldsymbol{\tau}}_{0:t} \,|\, t, \mathbf{s}_0)q_T(t)$, *let* $q_T(t) = q_{\Delta_{t+1}}(\Delta_{t+1} = 1) \prod_{t'=1}^{t} q_{\Delta_{t'}}(\Delta_{t'} = 0)$ *be a variational distribution defined on* $t \in \mathbb{N}_0$, *and let* $q_{\tilde{\boldsymbol{\mathcal{T}}}_{0:T}|T,\mathbf{S}_0}(\tilde{\boldsymbol{\tau}}_{0:t} \,|\, t, \mathbf{s}_0)$ *be as defined in Equation* (3). *Then, given any initial state* $\mathbf{s}_0$ *and outcome* $\mathbf{g}$, *Equation* (A.17) *can be rewritten as*

$$\mathcal{F}(\pi, q_T, \mathbf{s}_0, \mathbf{g}) = \mathbb{E}_{q_{\tilde{\boldsymbol{\mathcal{T}}}_0 \,|\, \mathbf{S}_0}(\tilde{\boldsymbol{\tau}}_0 \,|\, \mathbf{s}_0)} \left[ \sum_{t=0}^{\infty} \left( \prod_{t'=1}^{t} q_{\Delta_{t'}}(\Delta_{t'} = 0) \right) \left( r(\mathbf{s}_t, \mathbf{a}_t, \mathbf{g}; q_\Delta) - \mathbb{D}_{\mathrm{KL}}(\pi(\cdot \,|\, \mathbf{s}_t) \,\|\, p(\cdot \,|\, \mathbf{s}_t)) \right) \right]$$
$$\tag{A.22}$$

*where*

$$r(\mathbf{s}_t, \mathbf{a}_t, \mathbf{g}; q_\Delta) \doteq q_{\Delta_{t+1}}(\Delta_{t+1} = 1) \log p_d(\mathbf{g} \,|\, \mathbf{s}_t, \mathbf{a}_t) - \mathbb{D}_{\mathrm{KL}}(q_{\Delta_{t+1}} \,\|\, p_{\Delta_{t+1}}), \tag{A.23}$$

*Proof.* Consider the variational objective $\mathcal{F}(\pi, q_T, \mathbf{s}_0, \mathbf{g})$ in Equation (A.17):

$$\mathcal{F}(\pi, q_T, \mathbf{s}_0, \mathbf{g})$$

$$= \sum_{t=0}^{\infty} q_T(t) \, \mathbb{E}_{q_{\tilde{\boldsymbol{\mathcal{T}}}_{0:T}|T,\mathbf{S}_0}(\tilde{\boldsymbol{\tau}}_{0:t} \,|\, t,\mathbf{s}_0)} \left[ \log p_d(\mathbf{g} \,|\, \mathbf{s}_t, \mathbf{a}_t) - \mathbb{D}_{\mathrm{KL}}(q_{\tilde{\boldsymbol{\mathcal{T}}}_{0:T},T|\mathbf{S}_0}(\cdot \,|\, \mathbf{s}_0) \,\|\, p_{\tilde{\boldsymbol{\mathcal{T}}}_{0:T},T|\mathbf{S}_0}(\cdot \,|\, \mathbf{s}_0)) \right]$$
$$\tag{A.24}$$

$$= \sum_{t=0}^{\infty} q_T(t) \, \mathbb{E}_{q_{\tilde{\boldsymbol{\mathcal{T}}}_{0:T}|T,\mathbf{S}_0}(\tilde{\boldsymbol{\tau}}_{0:t} \,|\, t,\mathbf{s}_0)} \left[ \log p_d(\mathbf{g} \,|\, \mathbf{s}_t, \mathbf{a}_t) - \log \frac{q_{\tilde{\boldsymbol{\mathcal{T}}}_{0:T}|T,\mathbf{S}_0}(\tilde{\boldsymbol{\tau}}_{0:t} \,|\, t, \mathbf{s}_0)q_T(t)}{p_{\tilde{\boldsymbol{\mathcal{T}}}_{0:T}|T,\mathbf{S}_0}(\tilde{\boldsymbol{\tau}}_{0:t} \,|\, t, \mathbf{s}_0)p_T(t)} d\tilde{\boldsymbol{\tau}}_{0:t} \right] \tag{A.25}$$

$$= \sum_{t=0}^{\infty} q_T(t) \, \mathbb{E}_{q_{\tilde{\boldsymbol{\mathcal{T}}}_{0:T}|T,\mathbf{S}_0}(\tilde{\boldsymbol{\tau}}_{0:t} \,|\, t,\mathbf{s}_0)} \left[ \log p_d(\mathbf{g} \,|\, \mathbf{s}_t, \mathbf{a}_t) - \log \frac{q_{\tilde{\boldsymbol{\mathcal{T}}}_{0:T}|T,\mathbf{S}_0}(\tilde{\boldsymbol{\tau}}_{0:t} \,|\, t, \mathbf{s}_0)}{p_{\tilde{\boldsymbol{\mathcal{T}}}_{0:T}|T,\mathbf{S}_0}(\tilde{\boldsymbol{\tau}}_{0:t} \,|\, t, \mathbf{s}_0)} \right] - \sum_{t=0}^{\infty} q_T(t) \log \frac{q_T(t)}{q_T(t)}.$$
$$\tag{A.26}$$

Noting that $\sum_{t=0}^{\infty} q_T(t) \log \frac{q_T(t)}{q_T(t)} = \mathbb{D}_{\mathrm{KL}}(q_T \parallel p_T)$, we can write

$$
\begin{aligned}
&\mathcal{F}(\pi, q_T, \mathbf{s}_0, \mathbf{g}) \\
&= \sum_{t=0}^{\infty} q_T(t)\, \mathbb{E}_{q_{\tilde{\boldsymbol{\mathcal{T}}}_{0:T}|T,\mathbf{S}_0}(\tilde{\boldsymbol{\tau}}_{0:t}\,|\,t,\mathbf{s}_0)} \left[ \log p_d(\mathbf{g}\,|\,\mathbf{s}_t, \mathbf{a}_t) - \log \frac{q_{\tilde{\boldsymbol{\mathcal{T}}}_{0:T}|T,\mathbf{S}_0}(\tilde{\boldsymbol{\tau}}_{0:t}\,|\,t,\mathbf{s}_0)}{p_{\tilde{\boldsymbol{\mathcal{T}}}_{0:T}|T,\mathbf{S}_0}(\tilde{\boldsymbol{\tau}}_{0:t}\,|\,t,\mathbf{s}_0)} \right] - \mathbb{D}_{\mathrm{KL}}(q_T \parallel p_T)
\end{aligned}
$$
(A.27)

$$
\begin{aligned}
&= \sum_{t=0}^{\infty} q_T(t)\, \mathbb{E}_{q_{\tilde{\boldsymbol{\mathcal{T}}}_{0:T}|T,\mathbf{S}_0}(\tilde{\boldsymbol{\tau}}_{0:t}\,|\,t,\mathbf{s}_0)} \left[ \log p_d(\mathbf{g}\,|\,\mathbf{s}_t, \mathbf{a}_t) \right] \\
&\quad - \sum_{t=0}^{\infty} q_T(t)\, \mathbb{E}_{q_{\tilde{\boldsymbol{\mathcal{T}}}_{0:T}|T,\mathbf{S}_0}(\tilde{\boldsymbol{\tau}}_{0:t}\,|\,t,\mathbf{s}_0)} \left[ \log \frac{q_{\tilde{\boldsymbol{\mathcal{T}}}_{0:T}|T,\mathbf{S}_0}(\tilde{\boldsymbol{\tau}}_{0:t}\,|\,t,\mathbf{s}_0)}{p_{\tilde{\boldsymbol{\mathcal{T}}}_{0:T}|T,\mathbf{S}_0}(\tilde{\boldsymbol{\tau}}_{0:t}\,|\,t,\mathbf{s}_0)} \right] - \mathbb{D}_{\mathrm{KL}}(q_T \parallel p_T).
\end{aligned}
$$
(A.28)

Further noting that for an infinite-horizon trajectory distribution

$$
q_{\tilde{\boldsymbol{\mathcal{T}}}_{t'}|\mathbf{S}_{t'}}(\tilde{\boldsymbol{\tau}}_{t'}\,|\,\mathbf{s}_{t'}) \doteq \prod_{t=t'}^{\infty} p_d(\mathbf{s}_{t+1}\,|\,\mathbf{s}_t, \mathbf{a}_t)\pi(\mathbf{a}_t\,|\,\mathbf{s}_t),
$$
(A.29)

trajectory realization $\tilde{\boldsymbol{\tau}}_{t+1} \doteq \{\boldsymbol{\tau}_{t'}\}_{t'=t+1}^{\infty}$, and any joint probability density $f(\mathbf{s}_t, \mathbf{a}_t)$,

$$
\sum_{t=0}^{\infty} q_T(t)\, \mathbb{E}_{q_{\tilde{\boldsymbol{\mathcal{T}}}_{0:T}|T,\mathbf{S}_0}(\tilde{\boldsymbol{\tau}}_{0:t}\,|\,t,\mathbf{s}_0)} \left[ f(\mathbf{s}_t, \mathbf{a}_t) \right]
$$
(A.30)

$$
= \sum_{t=0}^{\infty} \left( \int q_{\tilde{\boldsymbol{\mathcal{T}}}_{T+1}|\mathbf{S}_0}(\tilde{\boldsymbol{\tau}}_{t+1}\,|\,\mathbf{s}_0) \left( \int_{\mathcal{S}^t \times \mathcal{A}^{t+1}} q_{\tilde{\boldsymbol{\mathcal{T}}}_{0:t}|\mathbf{S}_0}(\tilde{\boldsymbol{\tau}}_{0:t}\,|\,\mathbf{s}_0) q_T(t) f(\mathbf{s}_t, \mathbf{a}_t)\, d\tilde{\boldsymbol{\tau}}_{0:t} \right) d\tilde{\boldsymbol{\tau}}_{t+1} \right),
$$
(A.31)

$$
= \sum_{t=0}^{\infty} \left( \mathbb{E}_{q_{\tilde{\boldsymbol{\mathcal{T}}}_{0:T}|T,\mathbf{S}_0}(\tilde{\boldsymbol{\tau}}_{0:t}\,|\,t,\mathbf{s}_0)} \left[ q_T(t) f(\mathbf{s}_t, \mathbf{a}_t) \right] \cdot \left( \underbrace{\int q_{\tilde{\boldsymbol{\mathcal{T}}}_{T+1}|\mathbf{S}_0}(\tilde{\boldsymbol{\tau}}_{t+1}\,|\,\mathbf{s}_0)\, d\tilde{\boldsymbol{\tau}}_{t+1}}_{=1} \right) \right)
$$
(A.32)

$$
= \sum_{t=0}^{\infty} \left( \left( \int_{\mathcal{S}^t \times \mathcal{A}^{t+1}} q(\tilde{\boldsymbol{\tau}}_{0:t}\,|\,\mathbf{s}_0) q_T(t) f(\mathbf{s}_t, \mathbf{a}_t)\, d\tilde{\boldsymbol{\tau}}_{0:t} \right) \cdot \left( \underbrace{\int q_{\tilde{\boldsymbol{\mathcal{T}}}_{T+1}|\mathbf{S}_0}(\tilde{\boldsymbol{\tau}}_{t+1}\,|\,\mathbf{s}_0)\, d\tilde{\boldsymbol{\tau}}_{t+1}}_{=1} \right) \right)
$$
(A.33)

$$
= \sum_{t=0}^{\infty} \int q_{\tilde{\boldsymbol{\mathcal{T}}}_0|\mathbf{S}_0}(\tilde{\boldsymbol{\tau}}_0\,|\,\mathbf{s}_0) q_T(t) f(\mathbf{s}_t, \mathbf{a}_t)\, d\tilde{\boldsymbol{\tau}}_0
$$
(A.34)

$$
= \int q_{\tilde{\boldsymbol{\mathcal{T}}}_0|\mathbf{S}_0}(\tilde{\boldsymbol{\tau}}_0\,|\,\mathbf{s}_0) \sum_{t=0}^{\infty} q_T(t) f(\mathbf{s}_t, \mathbf{a}_t)\, d\tilde{\boldsymbol{\tau}}_0,
$$
(A.35)

we can express Equation (A.28) in terms of the infinite-horizon state–action trajectory $q_{\tilde{\boldsymbol{\mathcal{T}}}_0|\mathbf{S}_0}(\tilde{\boldsymbol{\tau}}_0\,|\,\mathbf{s}_0) \doteq \prod_{t=0}^{\infty} p_d(\mathbf{s}_{t+1}\,|\,\mathbf{s}_t, \mathbf{a}_t)\pi(\mathbf{a}_t\,|\,\mathbf{s}_t)$ as

$$
\begin{aligned}
&\mathcal{F}(\pi, q_T, \mathbf{s}_0, \mathbf{g}) \\
&= \int q_{\tilde{\boldsymbol{\mathcal{T}}}_0|\mathbf{S}_0}(\tilde{\boldsymbol{\tau}}_0\,|\,\mathbf{s}_0) \sum_{t=0}^{\infty} q_T(t) \log p(\mathbf{g}\,|\,\mathbf{s}_t, \mathbf{a}_t)\, d\tilde{\boldsymbol{\tau}} \\
&\quad - \sum_{t=0}^{\infty} q_T(t) \mathbb{D}_{\mathrm{KL}}(q_{\tilde{\boldsymbol{\mathcal{T}}}_{0:T}|T,\mathbf{S}_0}(\cdot\,|\,t,\mathbf{s}_0) \parallel p_{\tilde{\boldsymbol{\mathcal{T}}}_{0:T}|T,\mathbf{S}_0}(\cdot\,|\,t,\mathbf{s}_0)) - \mathbb{D}_{\mathrm{KL}}(q_T \parallel p_T)
\end{aligned}
$$
(A.36)

$$
\begin{aligned}
&= \mathbb{E}_{q_{\tilde{\boldsymbol{\mathcal{T}}}_0|\mathbf{S}_0}(\tilde{\boldsymbol{\tau}}_0\,|\,\mathbf{s}_0)} \left[ \sum_{t=0}^{\infty} q_T(t) \Big( \log p(\mathbf{g}\,|\,\mathbf{s}_t, \mathbf{a}_t) \right. \\
&\quad \left. - \mathbb{D}_{\mathrm{KL}}(q_{\tilde{\boldsymbol{\mathcal{T}}}_{0:T}|T,\mathbf{S}_0}(\cdot\,|\,t,\mathbf{s}_0) \parallel p_{\tilde{\boldsymbol{\mathcal{T}}}_{0:T}|T,\mathbf{S}_0}(\cdot\,|\,t,\mathbf{s}_0)) \Big) \right] - \mathbb{D}_{\mathrm{KL}}(q_T \parallel p_T).
\end{aligned}
$$
(A.37)

Using Lemma 5 and the definition of $q_T(t)$ in Equation (7), we can rewrite this objective as

$$
\mathcal{F}(\pi, q_T, \mathbf{s}_0, \mathbf{g})
$$
$$
= \mathbb{E}_{q_{\tilde{\boldsymbol{\mathcal{T}}}_0 | \mathbf{S}_0}(\tilde{\boldsymbol{\tau}}_0 \, | \, \mathbf{s}_0)} \left[ \sum_{t=0}^{\infty} \Big( \prod_{t'=1}^{t} q_{\Delta_{t'}}(\Delta_{t'} = 0) \Big) q_{\Delta_{t'}}(\Delta_{t'} = 1) \Big( \log p(\mathbf{g} \, | \, \mathbf{s}_t, \mathbf{a}_t) \right.
$$
$$
\left. - \mathbb{D}_{\mathrm{KL}}(q_{\tilde{\boldsymbol{\mathcal{T}}}_{0:T} | T, \mathbf{S}_0}(\cdot \, | \, t, \mathbf{s}_0) \, \| \, p_{\tilde{\boldsymbol{\mathcal{T}}}_{0:T} | T, \mathbf{S}_0}(\cdot \, | \, t, \mathbf{s}_0)) \Big) \right] - \sum_{t=0}^{\infty} \Big( \prod_{t'=1}^{t} q_{\Delta_{t'}}(\Delta_{t'} = 0) \Big) \mathbb{D}_{\mathrm{KL}}(q_{\Delta_{t+1}} \, \| \, p_{\Delta_{t+1}})
$$
$$
\tag{A.38}
$$
$$
= \mathbb{E}_{q_{\tilde{\boldsymbol{\mathcal{T}}}_0 | \mathbf{S}_0}(\tilde{\boldsymbol{\tau}}_0 \, | \, \mathbf{s}_0)} \left[ \sum_{t=0}^{\infty} \left( \prod_{t'=1}^{t} q(\Delta_{t'} = 0) \right) \right.
$$
$$
\cdot \Big( q(\Delta_{t+1} = 1) \big( \log p(\mathbf{g} \, | \, \mathbf{s}_t, \mathbf{a}_t) - \mathbb{D}_{\mathrm{KL}}(q_{\tilde{\boldsymbol{\mathcal{T}}}_{0:T} | T, \mathbf{S}_0}(\cdot \, | \, t, \mathbf{s}_0) \, \| \, p_{\tilde{\boldsymbol{\mathcal{T}}}_{0:T} | T, \mathbf{S}_0}(\cdot \, | \, t, \mathbf{s}_0)) \big) \tag{A.39}
$$
$$
\left. - \mathbb{D}_{\mathrm{KL}}(q_{\Delta_{t+1}} \, \| \, p_{\Delta_{t+1}}) \Big) \right],
$$

with

$$
\mathbb{D}_{\mathrm{KL}}(q_{\Delta_{t+1}} \, \| \, p_{\Delta_{t+1}})
$$
$$
= q_{\Delta_{t+1}}(\Delta_{t+1} = 0) \log \frac{q_{\Delta_{t+1}}(\Delta_{t+1} = 0)}{p_{\Delta_{t+1}}(\Delta_{t+1} = 0)} + (1 - q_{\Delta_{t+1}}(\Delta_{t+1} = 0)) \log \frac{1 - q_{\Delta_{t+1}}(\Delta_{t+1} = 0)}{1 - p_{\Delta_{t+1}}(\Delta_{t+1} = 0)}.
$$
$$
\tag{A.40}
$$

Next, to re-express $\mathbb{D}_{\mathrm{KL}}(q_{\tilde{\boldsymbol{\mathcal{T}}}_{0:T} | T, \mathbf{S}_0}(\cdot \, | \, t, \mathbf{s}_0) \, \| \, p_{\tilde{\boldsymbol{\mathcal{T}}}_{0:T} | T, \mathbf{S}_0}(\cdot \, | \, t, \mathbf{s}_0))$ as a sum over Kullback-Leibler divergences between distributions over single action random variables, we note that

$$
\mathbb{D}_{\mathrm{KL}}(q_{\tilde{\boldsymbol{\mathcal{T}}}_{0:T} | T, \mathbf{S}_0}(\cdot \, | \, t, \mathbf{s}_0) \, \| \, p_{\tilde{\boldsymbol{\mathcal{T}}}_{0:T} | T, \mathbf{S}_0}(\cdot \, | \, t, \mathbf{s}_0))
$$
$$
= \int_{\mathcal{S}^t \times \mathcal{A}^{t+1}} q_{\tilde{\boldsymbol{\mathcal{T}}}_{0:T} | T, \mathbf{S}_0}(\tilde{\boldsymbol{\tau}}_{0:t} \, | \, t, \mathbf{s}_0) \log \frac{q_{\tilde{\boldsymbol{\mathcal{T}}}_{0:T} | T, \mathbf{S}_0}(\tilde{\boldsymbol{\tau}}_{0:t} \, | \, t, \mathbf{s}_0)}{p_{\tilde{\boldsymbol{\mathcal{T}}}_{0:T} | T, \mathbf{S}_0}(\tilde{\boldsymbol{\tau}}_{0:t} \, | \, t, \mathbf{s}_0)} d\tilde{\boldsymbol{\tau}}_{0:t} \tag{A.41}
$$
$$
= \int_{\mathcal{S}^t \times \mathcal{A}^{t+1}} q_{\tilde{\boldsymbol{\mathcal{T}}}_{0:T} | T, \mathbf{S}_0}(\tilde{\boldsymbol{\tau}}_{0:t} \, | \, t, \mathbf{s}_0) \log \frac{\prod_{t'=1}^{t} \pi(\mathbf{a}_{t'} \, | \, \mathbf{s}_{t'})}{\prod_{t'=1}^{t} p(\mathbf{a}_{t'} \, | \, \mathbf{s}_{t'})} d\tilde{\boldsymbol{\tau}}_{0:t} \tag{A.42}
$$
$$
= \int_{\mathcal{S}^t \times \mathcal{A}^{t+1}} q_{\tilde{\boldsymbol{\mathcal{T}}}_{0:T} | T, \mathbf{S}_0}(\tilde{\boldsymbol{\tau}}_{0:t} \, | \, t, \mathbf{s}_0) \sum_{t'=0}^{t} \log \frac{\pi(\mathbf{a}_{t'} \, | \, \mathbf{s}_{t'})}{p(\mathbf{a}_{t'} \, | \, \mathbf{s}_{t'})} d\tilde{\boldsymbol{\tau}}_{0:t} \tag{A.43}
$$
$$
= \mathbb{E}_{q_{\tilde{\boldsymbol{\mathcal{T}}}_0 | \mathbf{S}_0}(\tilde{\boldsymbol{\tau}}_0 \, | \, \mathbf{s}_0)} \left[ \sum_{t'=0}^{t} \int_{\mathcal{A}} \pi(\mathbf{a}_{t'} \, | \, \mathbf{s}_{t'}) \log \frac{\pi(\mathbf{a}_{t'} \, | \, \mathbf{s}_{t'})}{p(\mathbf{a}_{t'} \, | \, \mathbf{s}_{t'})} d\mathbf{a}_{t'} \right] \tag{A.44}
$$
$$
= \mathbb{E}_{q_{\tilde{\boldsymbol{\mathcal{T}}}_0 | \mathbf{S}_0}(\tilde{\boldsymbol{\tau}}_0 \, | \, \mathbf{s}_0)} \left[ \sum_{t'=0}^{t} \mathbb{D}_{\mathrm{KL}}(\pi(\cdot \, | \, \mathbf{s}_{t'}) \, \| \, p(\cdot \, | \, \mathbf{s}_{t'})) \right], \tag{A.45}
$$

where we have used the same marginalization trick as above to express the expression in terms of an infinite-horizon trajectory distribution, which allows us to express Equation (A.39) as

$$
\mathcal{F}(\pi, q_T, \mathbf{s}_0, \mathbf{g})
$$
$$
= \mathbb{E}_{q_{\tilde{\boldsymbol{\mathcal{T}}}_0 | \mathbf{S}_0}(\tilde{\boldsymbol{\tau}}_0 \, | \, \mathbf{s}_0)} \left[ \sum_{t=0}^{\infty} \left( \prod_{t'=1}^{t} q(\Delta_{t'} = 0) \right) \right.
$$
$$
\cdot \Big( q_{\Delta_{t+1}}(\Delta_{t+1} = 1) \Big( \log p(\mathbf{g} \, | \, \mathbf{s}_t, \mathbf{a}_t) - \mathbb{E}_{q_{\tilde{\boldsymbol{\mathcal{T}}}_0 | \mathbf{S}_0}(\tilde{\boldsymbol{\tau}}_0 \, | \, \mathbf{s}_0)} \left[ \sum_{t'=0}^{t} \mathbb{D}_{\mathrm{KL}}(\pi(\cdot \, | \, \mathbf{s}_{t'}) \, \| \, p(\cdot \, | \, \mathbf{s}_{t'})) \right] \Big) \tag{A.46}
$$
$$
\left. - \mathbb{D}_{\mathrm{KL}}(q_{\Delta_{t+1}} \, \| \, p_{\Delta_{t+1}}) \Big) \right].
$$

Rearranging and dropping redundant expectation operators, we can now express the objective as

$$\mathcal{F}(\pi, q_T, \mathbf{s}_0, \mathbf{g})$$

$$= \mathbb{E}_{q_{\tilde{\mathcal{T}}_0 \mid \mathbf{S}_0}(\tilde{\tau}_0 \mid \mathbf{s}_0)} \left[ \sum_{t=0}^{\infty} \left( \prod_{t'=1}^{t} q_{\Delta_{t+1}}(\Delta_{t'} = 0) \right) \right.$$

$$\left. \cdot \left( q_{\Delta_{t+1}}(\Delta_{t+1} = 1) \left( \log p(\mathbf{g} \mid \mathbf{s}_t, \mathbf{a}_t) - \mathbb{E}_{q_{\tilde{\mathcal{T}}_0 \mid \mathbf{S}_0}(\tilde{\tau}_0 \mid \mathbf{s}_0)} \left[ \sum_{t'=0}^{t} \mathbb{D}_{\mathrm{KL}}(\pi(\cdot \mid \mathbf{s}_{t'}) \parallel p(\cdot \mid \mathbf{s}_{t'})) \right] \right) \right.\right. \quad \text{(A.47)}$$

$$\left.\left. - \mathbb{D}_{\mathrm{KL}}(q_{\Delta_{t+1}} \parallel p_{\Delta_{t+1}}) \right) \right].$$

$$= \mathbb{E}_{q_{\tilde{\mathcal{T}}_0 \mid \mathbf{S}_0}(\tilde{\tau}_0 \mid \mathbf{s}_0)} \left[ \sum_{t=0}^{\infty} \left( \prod_{t'=1}^{t} q_{\Delta_{t'}}(\Delta_{t'} = 0) \right) \right.$$

$$\left. \cdot \left( q_{\Delta_{t+1}}(\Delta_{t+1} = 1) \log p(\mathbf{g} \mid \mathbf{s}_t, \mathbf{a}_t) - \mathbb{D}_{\mathrm{KL}}(q_{\Delta_{t+1}} \parallel p_{\Delta_{t+1}}) \right) \right]$$

$$- \sum_{t=0}^{\infty} \underbrace{\left( \prod_{t'=1}^{t} q_{\Delta_{t'}}(\Delta_{t'} = 0) q_{\Delta_{t+1}}(\Delta_{t+1} = 1) \right)}_{= q_T(t)} \mathbb{E}_{q_{\tilde{\mathcal{T}}_0 \mid \mathbf{S}_0}(\tilde{\tau}_0 \mid \mathbf{s}_0)} \left[ \sum_{t'=0}^{t} \mathbb{D}_{\mathrm{KL}}(\pi(\cdot \mid \mathbf{s}_{t'}) \parallel p(\cdot \mid \mathbf{s}_{t'})) \right],$$

$$\text{(A.48)}$$

whereupon we note that the negative term can be expressed as

$$\sum_{t=0}^{\infty} q_T(t) \, \mathbb{E}_{q_{\tilde{\mathcal{T}}_0 \mid \mathbf{S}_0}(\tilde{\tau}_0 \mid \mathbf{s}_0)} \left[ \sum_{t'=0}^{t} \mathbb{D}_{\mathrm{KL}}(\pi(\cdot \mid \mathbf{s}_{t'}) \parallel p(\cdot \mid \mathbf{s}_{t'})) \right]$$

$$\text{(A.49)}$$

$$= \mathbb{E}_{q_{\tilde{\mathcal{T}}_0 \mid \mathbf{S}_0}(\tilde{\tau}_0 \mid \mathbf{s}_0)} \left[ \sum_{t=0}^{\infty} \sum_{t'=0}^{t} q_T(t) \mathbb{D}_{\mathrm{KL}}(\pi(\cdot \mid \mathbf{s}_{t'}) \parallel p(\cdot \mid \mathbf{s}_{t'})) \right]$$

$$= \mathbb{E}_{q_{\tilde{\mathcal{T}}_0 \mid \mathbf{S}_0}(\tilde{\tau}_0 \mid \mathbf{s}_0)} \left[ \sum_{t=0}^{\infty} q(T \geq t) \mathbb{D}_{\mathrm{KL}}(\pi(\cdot \mid \mathbf{s}_t) \parallel p(\cdot \mid \mathbf{s}_t)) \right] \quad \text{(A.50)}$$

$$= \mathbb{E}_{q_{\tilde{\mathcal{T}}_0 \mid \mathbf{S}_0}(\tilde{\tau}_0 \mid \mathbf{s}_0)} \left[ \sum_{t=0}^{\infty} \underbrace{\left( \prod_{t'=1}^{t} q_{\Delta_{t'}}(\Delta_{t'} = 0) \right)}_{\text{(by Lemma 2)}} \mathbb{D}_{\mathrm{KL}}(\pi(\cdot \mid \mathbf{s}_t) \parallel p(\cdot \mid \mathbf{s}_t)) \right], \quad \text{(A.51)}$$

where the second line follows from expanding the sums and regrouping terms. By substituting the expression in Equation (A.51) into Equation (A.48), we obtain an objective expressed entirely in terms of distributions over single-index random variables:

$$\mathcal{F}(\pi, q_T, \mathbf{s}_0, \mathbf{g})$$

$$= \mathbb{E}_{q_{\tilde{\mathcal{T}}_0 \mid \mathbf{S}_0}(\tilde{\tau}_0 \mid \mathbf{s}_0)} \left[ \sum_{t=0}^{\infty} \left( \prod_{t'=1}^{t} q_{\Delta_{t'}}(\Delta_{t'} = 0) \right) \right.$$

$$\text{(A.52)}$$

$$\left. \cdot \left( q_{\Delta_{t+1}}(\Delta_{t+1} = 1) \log p_d(\mathbf{g} \mid \mathbf{s}_t, \mathbf{a}_t) - \mathbb{D}_{\mathrm{KL}}(q_{\Delta_{t+1}} \parallel p_{\Delta_{t+1}}) - \mathbb{D}_{\mathrm{KL}}(\pi(\cdot \mid \mathbf{s}_t) \parallel p(\cdot \mid \mathbf{s}_t)) \right) \right]$$

$$= \mathbb{E}_{q_{\tilde{\mathcal{T}}_0 \mid \mathbf{S}_0}(\tilde{\tau}_0 \mid \mathbf{s}_0)} \left[ \sum_{t=0}^{\infty} \left( \prod_{t'=1}^{t} q_{\Delta_{t'}}(\Delta_{t'} = 0) \right) \left( r(\mathbf{s}_t, \mathbf{a}_t, \mathbf{g}; q_\Delta) - \mathbb{D}_{\mathrm{KL}}(\pi(\cdot \mid \mathbf{s}_t) \parallel p(\cdot \mid \mathbf{s}_t)) \right) \right], \quad \text{(A.53)}$$

where we defined

$$r(\mathbf{s}_t, \mathbf{a}_t, \mathbf{g}; q_\Delta) \doteq q_{\Delta_{t+1}}(\Delta_{t+1} = 1) \log p_d(\mathbf{g} \mid \mathbf{s}_t, \mathbf{a}_t) - \mathbb{D}_{\mathrm{KL}}(q_{\Delta_{t+1}} \parallel p_{\Delta_{t+1}}), \quad \text{(A.54)}$$

which concludes the proof. $\qquad \square$

**Theorem 1** (Outcome-Driven Variational Inference). *Let $q_T(t)$ and $q_{\tilde{\mathcal{T}}_{0:t} \mid T}(\tilde{\tau}_{0:t} \mid t, \mathbf{s}_0)$ be as defined in Equation (3) and Equation (7), and define*

$$V^\pi(\mathbf{s}_t, \mathbf{g}; q_T) \doteq \mathbb{E}_{\pi(\mathbf{a}_t \mid \mathbf{s}_t)}[Q^\pi(\mathbf{s}_t, \mathbf{a}_t, \mathbf{g}; q_T)] - \mathbb{D}_{\mathrm{KL}}(\pi(\cdot \mid \mathbf{s}_t) \parallel p(\cdot \mid \mathbf{s}_t)), \quad \text{(A.55)}$$

$$Q^\pi(\mathbf{s}_t, \mathbf{a}_t, \mathbf{g}; q_T) \doteq r(\mathbf{s}_t, \mathbf{a}_t, \mathbf{g}; q_\Delta) + q(\Delta_{t+1} = 0) \mathbb{E}_{p_d(\mathbf{s}_{t+1} \mid \mathbf{s}_t, \mathbf{a}_t)}[V^\pi(\mathbf{s}_{t+1}, \mathbf{g}; \pi, q_T)], \quad \text{(A.56)}$$

$$r(\mathbf{s}_t, \mathbf{a}_t, \mathbf{g}; q_\Delta) \doteq q_{\Delta_{t+1}}(\Delta_{t+1} = 1) \log p_d(\mathbf{g} \mid \mathbf{s}_t, \mathbf{a}_t) - \mathbb{D}_{\mathrm{KL}}(q_{\Delta_{t+1}} \parallel p_{\Delta_{t+1}}). \quad \text{(A.57)}$$

*Then given any initial state* $\mathbf{s}_0$ *and outcome* $\mathbf{g}$,

$$\mathbb{D}_{\mathrm{KL}}(q_{\tilde{\mathcal{T}}_{0:T},T|\mathbf{S}_0}(\cdot\,|\,\mathbf{s}_0)\,\|\,p_{\tilde{\mathcal{T}}_{0:T},T|\mathbf{S}_0,\mathbf{S}_{T^\star}}(\cdot\,|\,\mathbf{s}_0,\mathbf{g})) = -\mathcal{F}(\pi,q_T,\mathbf{s}_0,\mathbf{g}) + C = -V^\pi(\mathbf{s}_0,\mathbf{g};q_T) + C,$$

*where* $C \doteq \log p(\mathbf{g}\,|\,\mathbf{s}_0)$ *is independent of* $\pi$ *and* $q_T$, *and hence maximizing* $V^\pi(\mathbf{s}_0,\mathbf{g};\pi,q_T)$ *is equivalent to minimizing Equation* (6). *In other words,*

$$\begin{aligned}
&\underset{\pi\in\Pi,q_T\in\mathcal{Q}_T}{\arg\min}\ \{\mathbb{D}_{\mathrm{KL}}(q_{\tilde{\mathcal{T}}_{0:T},T|\mathbf{s}_0}(\cdot\,|\,\mathbf{s}_0)\,\|\,p_{\tilde{\mathcal{T}}_{0:T},T|\mathbf{S}_0,\mathbf{S}_{T^\star}}(\cdot\,|\,\mathbf{s}_0,\mathbf{g}))\}\\
&=\ \underset{\pi\in\Pi,q_T\in\mathcal{Q}_T}{\arg\max}\ \mathcal{F}(\pi,q_T,\mathbf{s}_0,\mathbf{g})\\
&=\ \underset{\pi\in\Pi,q_T\in\mathcal{Q}_T}{\arg\max}\ V^\pi(\mathbf{s}_0,\mathbf{g};q_T).
\end{aligned}$$

*Proof.* Consider the objective derived in Proposition 4,

$$\begin{aligned}
&\mathcal{F}(\pi,q_T,\mathbf{s}_0,\mathbf{g})\\
&= \mathbb{E}_{q_{\tilde{\mathcal{T}}_0|\mathbf{S}_0}(\tilde{\tau}_0\,|\,\mathbf{s}_0)}\left[\sum_{t=0}^{\infty}\left(\prod_{t'=1}^{t}q_{\Delta_{t'}}(\Delta_{t'}=0)\right)\right.\\
&\qquad\cdot\underbrace{\left(q_{\Delta_{t+1}}(\Delta_{t+1}=1)\log p_d(\mathbf{g}\,|\,\mathbf{s}_t,\mathbf{a}_t) - \mathbb{D}_{\mathrm{KL}}(q_{\Delta_{t+1}}\,\|\,p_{\Delta_{t+1}})\right)}_{\doteq\ r(\mathbf{s}_t,\mathbf{a}_t,\mathbf{g};q_\Delta)} \left. -\mathbb{D}_{\mathrm{KL}}(\pi(\mathbf{a}_t\,|\,\mathbf{s}_t)\,\|\,p(\mathbf{a}_t|\mathbf{s}_t))\right],
\end{aligned}$$
(A.58)

and recall that, by Proposition 2,

$$\mathbb{D}_{\mathrm{KL}}(q_{\tilde{\mathcal{T}}_{0:T},T|\mathbf{S}_0}(\cdot\,|\,\mathbf{s}_0)\,\|\,p_{\tilde{\mathcal{T}}_{0:T},T|\mathbf{S}_0,\mathbf{S}_{T^\star}}(\cdot\,|\,\mathbf{s}_0,\mathbf{g})) = -\mathcal{F}(\pi,q_T,\mathbf{s}_0,\mathbf{g}) + \log p(\mathbf{g}|\mathbf{s}_0). \tag{A.59}$$

Therefore, to prove the result that

$$\mathbb{D}_{\mathrm{KL}}(q_{\tilde{\mathcal{T}}_{0:T},T|\mathbf{S}_0}(\cdot\,|\,\mathbf{s}_0)\,\|\,p_{\tilde{\mathcal{T}}_{0:T},T|\mathbf{S}_0,\mathbf{S}_{T^\star}}(\cdot\,|\,\mathbf{s}_0,\mathbf{g})) = -V^\pi(\mathbf{s}_0,\mathbf{g};q_T) + \log p(\mathbf{g}|\mathbf{s}_0),$$

we just need to show that $\mathcal{F}(\pi,q_T,\mathbf{s}_0,\mathbf{g}) = V^\pi(\mathbf{s}_0,\mathbf{g};q_T)$ for $V^\pi(\mathbf{s}_0,\mathbf{g};q_T)$ as defined in the theorem. To do so, we start from the objective $\mathcal{F}(\pi,q_T,\mathbf{s}_0,\mathbf{g})$ and and unroll it for $t=0$:

$$\begin{aligned}
&\mathcal{F}(\pi,q_T,\mathbf{s}_0,\mathbf{g})\\
&= \mathbb{E}_{q_{\tilde{\mathcal{T}}_0|\mathbf{S}_0}(\tilde{\tau}_0\,|\,\mathbf{s}_0)}\left[\sum_{t=0}^{\infty}\left(\prod_{t'=1}^{t}q_{\Delta_{t'}}(\Delta_{t'}=0)\right)r(\mathbf{s}_t,\mathbf{a}_t,\mathbf{g};q_\Delta) - \mathbb{D}_{\mathrm{KL}}(\pi(\mathbf{a}_t\,|\,\mathbf{s}_t)\,\|\,p(\mathbf{a}_t|\mathbf{s}_t))\right] \qquad\text{(A.60)}\\
&= \mathbb{E}_{\pi(\mathbf{a}_0\,|\,\mathbf{s}_0)}\left[r(\mathbf{s}_0,\mathbf{a}_0,\mathbf{g};q_\Delta) + \mathbb{E}_{q(\tau_1\,|\,\mathbf{s}_0,\mathbf{a}_0)}\left[\sum_{t=1}^{\infty}\prod_{t'=1}^{t}q_{\Delta_{t'}}(\Delta_{t'}=0)\Big(r(\mathbf{s}_t,\mathbf{a}_t,\mathbf{g};q_\Delta)\right.\right.\\
&\qquad \left.\left.- \mathbb{D}_{\mathrm{KL}}(\pi(\cdot\,|\,\mathbf{s}_t)\,\|\,p(\cdot\,|\,\mathbf{s}_t))\Big)\right]\right] - \mathbb{D}_{\mathrm{KL}}(\pi(\cdot\,|\,\mathbf{s}_0)\,\|\,p(\cdot\,|\,\mathbf{s}_0)).
\end{aligned}$$
(A.61)

With this expression at hand, we now define

$$\begin{aligned}
&Q_{\mathrm{sum}}^\pi(\mathbf{s}_0,\mathbf{a}_0,\mathbf{g};q_T)\\
&\doteq r(\mathbf{s}_0,\mathbf{a}_0,\mathbf{g};q_\Delta) + \mathbb{E}_{q(\tau_1|\mathbf{s}_0,\mathbf{a}_0)}\left[\sum_{t=1}^{\infty}\prod_{t'=1}^{t}q_{\Delta_{t'}}(\Delta_{t'}=0)\Big(r(\mathbf{s}_t,\mathbf{a}_t,\mathbf{g};q_\Delta) - \mathbb{D}_{\mathrm{KL}}(\pi(\cdot\,|\,\mathbf{s}_t)\,\|\,p(\cdot\,|\,\mathbf{s}_t))\Big)\right],
\end{aligned}$$
(A.62)

and note that $\mathcal{F}(\pi,q_T,\mathbf{s}_0,\mathbf{g}) = \mathbb{E}_{\pi(\mathbf{a}_0\,|\,\mathbf{s}_0)}[Q_{\mathrm{sum}}^\pi(\mathbf{s}_0,\mathbf{a}_0,\mathbf{g};q_T)] - \mathbb{D}_{\mathrm{KL}}(\pi(\cdot\,|\,\mathbf{s}_0)\,\|\,p(\cdot\,|\,\mathbf{s}_0)) = V^\pi(\mathbf{s}_0,\mathbf{g};q_T)$, as per the definition of $V^\pi(\mathbf{s}_0,\mathbf{g};q_T)$. To prove the theorem from this intermediate result, we now have to show that $Q_{\mathrm{sum}}^\pi(\mathbf{s}_0,\mathbf{a}_0,\mathbf{g};q_T)$ as defined in Equation (A.62) can in fact be expressed recursively as $Q_{\mathrm{sum}}^\pi(\mathbf{s}_t,\mathbf{a}_t,\mathbf{g};q_T) = Q^\pi(\mathbf{s}_0,\mathbf{a}_0,\mathbf{g};q_T)$ with

$$Q^\pi(\mathbf{s}_0,\mathbf{a}_0,\mathbf{g};q_T) = r(\mathbf{s}_t,\mathbf{a}_t,\mathbf{g};q_\Delta) + q(\Delta_{t+1}=0)\,\mathbb{E}_{p_d(\mathbf{s}_{t+1}\,|\,\mathbf{s}_t,\mathbf{a}_t)}[V^\pi(\mathbf{s}_{t+1},\mathbf{g};\pi,q_T)]. \tag{A.63}$$

To see that this is the case, first, unroll $Q^\pi(\mathbf{s}_0, \mathbf{a}_0, \mathbf{g}; q_T)$ for $t = 1$,

$$Q^\pi_{\text{sum}}(\mathbf{s}_0, \mathbf{a}_0, \mathbf{g}; q_T)$$

$$= r(\mathbf{s}_0, \mathbf{a}_0, \mathbf{g}; q_\Delta) + \mathbb{E}_{q(\boldsymbol{\tau}_1|\mathbf{s}_0,\mathbf{a}_0)}\left[\sum_{t=1}^\infty \prod_{t'=1}^t q_{\Delta_{t'}}(\Delta_{t'} = 0)\Big(r(\mathbf{s}_t, \mathbf{a}_t, \mathbf{g}; q_\Delta) - \mathbb{D}_{\text{KL}}(\pi(\cdot\,|\,\mathbf{s}_t)\,\|\,p(\cdot\,|\,\mathbf{s}_t))\Big)\right]$$

(A.64)

$$= r(\mathbf{s}_0, \mathbf{a}_0, \mathbf{g}; q_\Delta) + \mathbb{E}_{p_d(\mathbf{s}_1|\mathbf{a}_0,\mathbf{a}_0)}\Bigg[\mathbb{E}_{q(\boldsymbol{\tau}_1|\mathbf{s}_0,\mathbf{a}_0)}\bigg[\sum_{t=1}^\infty \prod_{t'=1}^t q_{\Delta_{t'}}(\Delta_{t'} = 0)\Big(r(\mathbf{s}_t, \mathbf{a}_t, \mathbf{g}; q_\Delta)$$

(A.65)

$$- \mathbb{D}_{\text{KL}}(\pi(\cdot\,|\,\mathbf{s}_t)\,\|\,p(\cdot\,|\,\mathbf{s}_t))\Big)\bigg]\Bigg]$$

$$= r(\mathbf{s}_0, \mathbf{a}_0, \mathbf{g}; q_\Delta) + \mathbb{E}_{p_d(\mathbf{s}_1|\mathbf{a}_0,\mathbf{a}_0)}\Bigg[\mathbb{E}_{\pi(\mathbf{a}_1\,|\,\mathbf{s}_1)}\bigg[q_{\Delta_1}(\Delta_1 = 0)\left(r(\mathbf{s}_1, \mathbf{a}_1, \mathbf{g}; q_\Delta) - \mathbb{D}_{\text{KL}}(\pi(\cdot\,|\,\mathbf{s}_1)\,\|\,p(\cdot\,|\,\mathbf{s}_1))\right)$$

$$+ \mathbb{E}_{q(\boldsymbol{\tau}_2|\mathbf{s}_1,\mathbf{a}_1)}\bigg[\sum_{t=2}^\infty \prod_{t'=2}^t q_{\Delta_{t'}}(\Delta_{t'} = 0)\Big(r(\mathbf{s}_t, \mathbf{a}_t, \mathbf{g}; q_\Delta) - \mathbb{D}_{\text{KL}}(\pi(\cdot\,|\,\mathbf{s}_t)\,\|\,p(\cdot\,|\,\mathbf{s}_t))\Big)\bigg]\bigg]\Bigg],$$

(A.66)

and note that we can rearrange this expression to obtain the recursive relationship

$$Q^\pi_{\text{sum}}(\mathbf{s}_0, \mathbf{a}_0, \mathbf{g}; q_T)$$

$$= r(\mathbf{s}_0, \mathbf{a}_0, \mathbf{g}; q_\Delta) + q_{\Delta_1}(\Delta_1 = 0)\,\mathbb{E}_{p_d(\mathbf{s}_{0+1}\,|\,\mathbf{s}_0,\mathbf{a}_0)}\Bigg[-\mathbb{D}_{\text{KL}}(\pi(\cdot\,|\,\mathbf{s}_1)\,\|\,p(\cdot\,|\,\mathbf{s}_1))$$

$$+ \mathbb{E}_{\pi(\mathbf{a}_1\,|\,\mathbf{s}_1)}\bigg[r(\mathbf{s}_1, \mathbf{a}_1, \mathbf{g}; q_\Delta) + \mathbb{E}\bigg[\sum_{t=2}^\infty \left(\prod_{t'=2}^t q_{\Delta_{t'}}(\Delta_{t'} = 0)\right)\Big(r(\mathbf{s}_t, \mathbf{a}_t, \mathbf{g}; q_\Delta)$$

(A.67)

$$- \mathbb{D}_{\text{KL}}(\pi(\cdot\,|\,\mathbf{s}_t)\,\|\,p(\cdot\,|\,\mathbf{s}))\Big)\bigg]\bigg]\Bigg],$$

where the innermost expectation is taken with respect to $q(\boldsymbol{\tau}_2|\mathbf{s}_1, \mathbf{a}_1)$. With this result and noting that

$$Q^\pi_{\text{sum}}(\mathbf{s}_1, \mathbf{a}_1, \mathbf{g}; q_T)$$

$$= r(\mathbf{s}_1, \mathbf{a}_1, \mathbf{g}; q_\Delta) + \mathbb{E}\left[\sum_{t=2}^\infty \left(\prod_{t'=2}^t q_{\Delta_{t'}}(\Delta_{t'} = 0)\right)\Big(r(\mathbf{s}_t, \mathbf{a}_t, \mathbf{g}; q_\Delta) - \mathbb{D}_{\text{KL}}(\pi(\cdot\,|\,\mathbf{s}_t)\,\|\,p(\cdot\,|\,\mathbf{s}))\Big)\right],$$

(A.68)

where the expectation is again taken with respect to $q(\boldsymbol{\tau}_2|\mathbf{s}_1, \mathbf{a}_1)$, we see that

$$Q^\pi_{\text{sum}}(\mathbf{s}_0, \mathbf{a}_0, \mathbf{g}; q_T)$$

$$= r(\mathbf{s}_0, \mathbf{a}_0, \mathbf{g}; q_\Delta) + q_{\Delta_1}(\Delta_1 = 0)\,\mathbb{E}_{p_d(\mathbf{s}_{0+1}\,|\,\mathbf{s}_0,\mathbf{a}_0)}\Big[\mathbb{E}_{\pi(\mathbf{a}_1|\mathbf{s}_1)}\left[Q^\pi_{\text{sum}}(\mathbf{s}_1, \mathbf{a}_1, \mathbf{g}; q_T)\right]$$

(A.69)

$$- \mathbb{D}_{\text{KL}}(\pi(\cdot\,|\,\mathbf{s}_1)\,\|\,p(\cdot\,|\,\mathbf{s}_1))\Big]$$

$$= r(\mathbf{s}_0, \mathbf{a}_0, \mathbf{g}; q_\Delta) + q_{\Delta_1}(\Delta_1 = 0)\,\mathbb{E}_{p_d(\mathbf{s}_1|\mathbf{s},\mathbf{a})}\left[V^\pi(\mathbf{s}_1, \mathbf{g}; q_T)\right],$$

(A.70)

for $V(\mathbf{s}_{t+1}, \mathbf{g}; q_T)$ as defined above, as desired. In other words, we have that

$$\mathcal{F}(\pi, q_T, \mathbf{s}_0, \mathbf{g}) = \mathbb{E}_{\pi(\mathbf{a}_0\,|\,\mathbf{s}_0)}[Q^\pi_{\text{sum}}(\mathbf{s}_0, \mathbf{a}_0, \mathbf{g}; q_T)] - \mathbb{D}_{\text{KL}}(\pi(\cdot\,|\,\mathbf{s}_0)\,\|\,p(\cdot\,|\,\mathbf{s}_0)) = V^\pi(\mathbf{s}_0, \mathbf{g}; q_T). \quad \text{(A.71)}$$

Combining this result with Proposition 2 and Proposition 4, we finally conclude that

$$\mathbb{D}_{\text{KL}}(q_{\tilde{\boldsymbol{\mathcal{T}}}_{0:T},T|\mathbf{S}_0}(\cdot\,|\,\mathbf{s}_0)\,\|\,p_{\tilde{\boldsymbol{\mathcal{T}}}_{0:T},T|\mathbf{S}_0,\mathbf{S}_{T^\star}}(\cdot\,|\,\mathbf{s}_0, \mathbf{g})) = -\mathcal{F}(\pi, q_T, \mathbf{s}_0, \mathbf{g}) + C = -V^\pi(\mathbf{s}_0, \mathbf{g}; q_T) + C,$$

(A.72)

where $C \doteq \log p(\mathbf{g}\,|\,\mathbf{s}_0)$ is independent of $\pi$ and $q_T$. Hence, maximizing $V^\pi(\mathbf{s}_0, \mathbf{g}; \pi, q_T)$ is equivalent to minimizing the objective in Equation (6). In other words,

$$\underset{\pi\in\Pi, q_T\in\mathcal{Q}_T}{\arg\min}\ \{\mathbb{D}_{\text{KL}}(q_{\tilde{\boldsymbol{\mathcal{T}}}_{0:T},T|\mathbf{S}_0}(\cdot\,|\,\mathbf{s}_0)\,\|\,p_{\tilde{\boldsymbol{\mathcal{T}}}_{0:T},T|\mathbf{S}_0,\mathbf{S}_{T^\star}}(\cdot\,|\,\mathbf{s}_0, \mathbf{g}))\}$$

(A.73)

$$= \underset{\pi\in\Pi, q_T\in\mathcal{Q}_T}{\arg\max}\ \mathcal{F}(\pi, q_T, \mathbf{s}_0, \mathbf{g}) = \underset{\pi\in\Pi, q_T\in\mathcal{Q}_T}{\arg\max}\ V^\pi(\mathbf{s}_0, \mathbf{g}; q_T).$$

This concludes the proof. $\qquad\qquad\square$

**Corollary 2** (Fixed-Discount Outcome-Driven Variational Inference). *Let $q_T = p_T$, assume that $p_T$ is a Geometric distribution with parameter $\gamma \in (0, 1)$. Then the inference problem in Equation (6) of finding a goal-directed variational trajectory distribution simplifies to maximizing the following recursively defined variational objective with respect to $\pi$:*

$$\bar{V}^\pi(\mathbf{s}_0, \mathbf{g}; \gamma) \doteq \mathbb{E}_{\pi(\mathbf{a}_0 \mid \mathbf{s}_0)}[Q(\mathbf{s}_0, \mathbf{a}_0, \mathbf{g}; \gamma)] - \mathbb{D}_{\mathrm{KL}}(\pi(\cdot \mid \mathbf{s}_0) \| p(\cdot \mid \mathbf{s})0)), \tag{A.74}$$

*where*

$$\bar{Q}^\pi(\mathbf{s}_0, \mathbf{a}_0, \mathbf{g}; \gamma) \doteq (1 - \gamma) \log p_d(\mathbf{g} \mid \mathbf{s}_0, \mathbf{a}_0) + \gamma \, \mathbb{E}_{p_d(\mathbf{s}_1 \mid \mathbf{s}_0, \mathbf{a}_0)}[V(\mathbf{s}_1, \mathbf{g}; \gamma)]. \tag{A.75}$$

*Proof.* The result follows immediately when replacing $q_\Delta$ in Theorem 1 by $p_\Delta$ and noting that $\mathbb{D}_{\mathrm{KL}}(p_\Delta \| p_\Delta) = 0$. □

## A.3 Optimal Variational Posterior over $T$

**Proposition 2** (Optimal Variational Distribution over $T$). *The optimal variational distribution $q_T^\star$ with respect to Equation (8) is defined recursively in terms of $q_{\Delta_{t+1}}^\star(\Delta_{t+1} = 0) \forall t \in \mathbb{N}_0$ by*

$$q_{\Delta_{t+1}}^\star(\Delta_{t+1} = 0; \pi, Q^\pi) = \sigma\left(\Lambda(\mathbf{s}_t, \pi, q_T, Q^\pi) + \sigma^{-1}\left(p_{\Delta_{t+1}}(\Delta_{t+1} = 0)\right)\right), \tag{A.76}$$

*where*

$$\Lambda(\mathbf{s}_t, \pi, q_T, Q^\pi) \doteq \mathbb{E}_{\pi(\mathbf{a}_{t+1} \mid \mathbf{s}_{t+1}) p_d(\mathbf{s}_{t+1} \mid \mathbf{s}_t, \mathbf{a}_t) \pi(\mathbf{a}_t \mid \mathbf{s}_t)}[Q^\pi(\mathbf{s}_{t+1}, \mathbf{a}_{t+1}, \mathbf{g}; q_T) - \log p_d(\mathbf{g} \mid \mathbf{s}_t, \mathbf{a}_t)]$$

*and $\sigma(\cdot)$ is the sigmoid function, that is, $\sigma(x) = \frac{1}{e^{-x}+1}$ and $\sigma^{-1}(x) = \log \frac{x}{1-x}$.*

*Proof.* Consider $\mathcal{F}(\pi, q_T, \mathbf{s}_0, \mathbf{g})$:

$$\begin{aligned}
\mathcal{F}(\pi, q_T, \mathbf{s}_t, \mathbf{g}) &= \mathbb{E}_{\pi(\mathbf{a}_t \mid \mathbf{s}_t)}[Q^\pi(\mathbf{s}_t, \mathbf{a}_t, \mathbf{g}; q_T)] \\
&= \mathbb{E}_{\pi(\mathbf{a}_t \mid \mathbf{s}_t)}[r(\mathbf{s}_t, \mathbf{a}_t, \mathbf{g}; q_\Delta) + q_{\Delta_{t+1}}(\Delta_{t+1} = 0) \mathbb{E}[V(\mathbf{s}_{t+1}, \mathbf{g}; q_T)]].
\end{aligned} \tag{A.77}$$

Since the variational objective $\mathcal{F}(\pi, q_T, \mathbf{s}_t, \mathbf{g})$ can be expressed recursively as

$$V^\pi(\mathbf{s}_t, \mathbf{g}; q_T) \doteq \mathbb{E}_{\pi(\mathbf{a}_t \mid \mathbf{s}_t)}[Q(\mathbf{s}_t, \mathbf{a}_t, \mathbf{g}; q_T)] - \mathbb{D}_{\mathrm{KL}}(\pi(\cdot \mid \mathbf{s}_t) \| p(\cdot \mid \mathbf{s}_t)),$$

with

$$Q^\pi(\mathbf{s}_t, \mathbf{a}_t, \mathbf{g}; q_T) = r(\mathbf{s}_t, \mathbf{a}_t, \mathbf{g}; q_\Delta) + q_{\Delta_{t+1}}(\Delta_{t+1} = 0) \mathbb{E}_{p_d(\mathbf{s}_{t+1} \mid \mathbf{s}_t, \mathbf{a}_t)}[V^\pi(\mathbf{s}_{t+1}, \mathbf{g}; q_T)],$$

$$r(\mathbf{s}_t, \mathbf{a}_t, \mathbf{g}; q_\Delta) = q_{\Delta_{t+1}}(\Delta_{t+1} = 1) \log p_d(\mathbf{g} \mid \mathbf{s}_t, \mathbf{a}_t) - \mathbb{D}_{\mathrm{KL}}(q_{\Delta_{t+1}} \| p_{\Delta_{t+1}}),$$

and since $\mathbb{D}_{\mathrm{KL}}(q_{\Delta_{t+1}} \| p_{\Delta_{t+1}})$ is strictly convex in $q_{\Delta_{t+1}}(\Delta_{t+1} = 0)$, we can find the globally optimal Bernoulli distribution parameters $q_{\Delta_{t+1}}(\Delta_{t+1} = 0)$ for all $t \in \mathbb{N}_0$ recursively. That is, it is sufficient to solve the problem

$$q_{\Delta_{t+1}}^\star(\Delta_{t+1} = 0) \doteq \underset{q_{\Delta_{t+1}}(\Delta_{t+1}=0)}{\arg\max} \{\mathcal{F}(\pi, q_T, \mathbf{s}_0, \mathbf{g})\} = \underset{q_{\Delta_{t+1}}(\Delta_{t+1}=0)}{\arg\max} \{\mathcal{F}(\pi, q_{\Delta_1}, \dots, q_{\Delta_{t+1}}, \dots, \mathbf{s}_0, \mathbf{g})\} \tag{A.78}$$

for a fixed $t + 1$. To do so, we take the derivative of $\mathcal{F}(\pi, q_{\Delta_1}, \dots, q_{\Delta_{t+1}}, \dots, \mathbf{s}_0, \mathbf{g})$, which—defined recursively—is given by

$$\begin{aligned}
&\mathbb{E}_{\pi(\mathbf{a}_t \mid \mathbf{s}_t)}[Q(\mathbf{s}_t, \mathbf{a}_t, \mathbf{g}; q_T) - \mathbb{D}_{\mathrm{KL}}(\pi(\cdot \mid \mathbf{s}_t) \| p(\cdot \mid \mathbf{s}_t)) \\
&= \mathbb{E}_{\pi(\mathbf{a}_t \mid \mathbf{s}_t)}\left[r(\mathbf{s}_t, \mathbf{a}_t, \mathbf{g}; q_\Delta) + q_{\Delta_{t+1}}(\Delta_{t+1} = 0) \mathbb{E}_{p_d(\mathbf{s}_{t+1} \mid \mathbf{s}_t, \mathbf{a}_t)}[V^\pi(\mathbf{s}_{t+1}, \mathbf{g}; q_T)]\right] \\
&\qquad - \mathbb{D}_{\mathrm{KL}}(\pi(\cdot \mid \mathbf{s}_t) \| p(\cdot \mid \mathbf{s}_t))
\end{aligned} \tag{A.79}$$

$$\begin{aligned}
&= \mathbb{E}_{\pi(\mathbf{a}_t \mid \mathbf{s}_t)}\Bigg[q_{\Delta_{t+1}}(\Delta_{t+1} = 1) \log p_d(\mathbf{g} \mid \mathbf{s}_t, \mathbf{a}_t) - \mathbb{D}_{\mathrm{KL}}(q_{\Delta_{t+1}} \| p_{\Delta_{t+1}}) \\
&\qquad + q_{\Delta_{t+1}}(\Delta_{t+1} = 0) \mathbb{E}_{p_d(\mathbf{s}_{t+1} \mid \mathbf{s}_t, \mathbf{a}_t)}[V^\pi(\mathbf{s}_{t+1}, \mathbf{g}; q_T)]\Bigg] - \mathbb{D}_{\mathrm{KL}}(\pi(\cdot \mid \mathbf{s}_t) \| p(\cdot \mid \mathbf{s}_t))
\end{aligned} \tag{A.80}$$

$$\begin{aligned}
&= \mathbb{E}_{\pi(\mathbf{a}_t \mid \mathbf{s}_t)}\Bigg[(1 - q_{\Delta_{t+1}}(\Delta_{t+1} = 0)) \log p_d(\mathbf{g} \mid \mathbf{s}_t, \mathbf{a}_t) - \mathbb{D}_{\mathrm{KL}}(q_{\Delta_{t+1}} \| p_{\Delta_{t+1}}) \\
&\qquad + q_{\Delta_{t+1}}(\Delta_{t+1} = 0) \mathbb{E}_{p_d(\mathbf{s}_{t+1} \mid \mathbf{s}_t, \mathbf{a}_t)}[V^\pi(\mathbf{s}_{t+1}, \mathbf{g}; q_T)]\Bigg] - \mathbb{D}_{\mathrm{KL}}(\pi(\cdot \mid \mathbf{s}_t) \| p(\cdot \mid \mathbf{s}_t)),
\end{aligned} \tag{A.81}$$

with respect to $q_{\Delta_{t+1}}(\Delta_{t+1} = 0)$ and set it to zero, which yields

$$\begin{aligned}
0 = &-\mathbb{E}_{\pi(\mathbf{a}_t \mid \mathbf{s}_t)}\left[\log p_d(\mathbf{g} \mid \mathbf{s}_t, \mathbf{a}_t) + \mathbb{E}_{\pi(\mathbf{a}_{t+1} \mid \mathbf{s}_{t+1}) p_d(\mathbf{s}_{t+1} \mid \mathbf{s}_t, \mathbf{a}_t)}[Q^\pi(\mathbf{s}_{t+1}, \mathbf{a}_{t+1}, \mathbf{g}; q_T)]\right] \\
&+ \log \frac{1 - q_{\Delta_{t+1}}^\star(\Delta_{t+1} = 0)}{1 - p_{\Delta_{t+1}}(\Delta_{t+1} = 0)} - \log \frac{q_{\Delta_{t+1}}^\star(\Delta_{t+1} = 0)}{p_{\Delta_{t+1}}(\Delta_{t+1} = 0)}.
\end{aligned} \tag{A.82}$$

Rearranging, we get

$$\frac{q_{\Delta_{t+1}}^{\star}(\Delta_{t+1}=0)}{1-q_{\Delta_{t+1}}^{\star}(\Delta_{t+1}=0)} = \exp\left(\mathbb{E}[Q^\pi(\mathbf{s}_{t+1},\mathbf{a}_{t+1},\mathbf{g};q_T) - \log p_d(\mathbf{g}\,|\,\mathbf{s}_t,\mathbf{a}_t)] + \log\frac{p_{\Delta_{t+1}}(\Delta_{t+1}=0)}{1-p_{\Delta_{t+1}}(\Delta_{t+1}=0)}\right),$$
(A.83)

where the expectation is taken with respect to $\pi(\mathbf{a}_{t+1}\,|\,\mathbf{s}_{t+1})p_d(\mathbf{s}_{t+1}\,|\,\mathbf{s}_t,\mathbf{a}_t)\pi(\mathbf{a}_t\,|\,\mathbf{s}_t)$ and the $Q$-function depends on $q(\Delta_{t'})$ with $t' > t$, but not on $q_{\Delta_{t+1}}^{\star}(\Delta_{t+1}=0)$. Solving for $q_{\Delta_{t+1}}^{\star}(\Delta_{t+1}=0)$. Solving for $q_{\Delta_{t+1}}^{\star}(\Delta_{t+1}=0)$, we obtain

$$q_{\Delta_{t+1}}^{\star}(\Delta_{t+1}=0)$$

$$= \frac{\exp(\mathbb{E}_{p_{\pi p_d}\pi(\mathbf{a}_t\,|\,\mathbf{s}_t)}[Q^\pi(\mathbf{s}_{t+1},\mathbf{a}_{t+1},\mathbf{g};q_T) - \log p_d(\mathbf{g}\,|\,\mathbf{s}_t,\mathbf{a}_t)] + \log\frac{p_{\Delta_{t+1}}(\Delta_{t+1}=0)}{1-p_{\Delta_{t+1}}(\Delta_{t+1}=0)})}{1+\exp(\mathbb{E}_{p_{\pi p_d}\pi(\mathbf{a}_t\,|\,\mathbf{s}_t)}[Q^\pi(\mathbf{s}_{t+1},\mathbf{a}_{t+1},\mathbf{g};q_T) - \log p_d(\mathbf{g}\,|\,\mathbf{s}_t,\mathbf{a}_t)] + \log\frac{p_{\Delta_{t+1}}(\Delta_{t+1}=0)}{1-p_{\Delta_{t+1}}(\Delta_{t+1}=0)})}$$
(A.84)

$$= \sigma\left(\mathbb{E}_{p_{\pi p_d}}[Q^\pi(\mathbf{s}_{t+1},\mathbf{a}_{t+1},\mathbf{g};q_T)] - \mathbb{E}_{\pi(\mathbf{a}_t\,|\,\mathbf{s}_t)}[\log p_d(\mathbf{g}\,|\,\mathbf{s}_t,\mathbf{a}_t)] + \sigma^{-1}\left(p_{\Delta_{t+1}}(\Delta_{t+1}=0)\right)\right),$$
(A.85)

where $p_{\pi p_d} \doteq \pi(\mathbf{a}_{t+1}\,|\,\mathbf{s}_{t+1})p_d(\mathbf{s}_{t+1}\,|\,\mathbf{s}_t,\mathbf{a}_t)$, $\sigma(\cdot)$ is the sigmoid function with $\sigma(x) = \frac{1}{e^{-x}+1}$ and $\sigma^{-1}(x) = \log\frac{x}{1-x}$. This concludes the proof. $\qquad\square$

**Remark 1.** *As can be seen from [Proposition 2], the optimal approximation to the posterior over $T$ trades off short-term rewards via $\mathbb{E}_{\pi(\mathbf{a}_t\,|\,\mathbf{s}_t)}[r(\mathbf{s}_t,\mathbf{a}_t,\mathbf{g};q_\Delta)]$, long-term rewards via $\mathbb{E}_{\pi(\mathbf{a}_{t+1}\,|\,\mathbf{s}_{t+1})p_d(\mathbf{s}_{t+1}\,|\,\mathbf{s}_t,\mathbf{a}_t)}[Q^\pi(\mathbf{s}_{t+1},\mathbf{a}_{t+1},\mathbf{g};q_T)]$, and the prior log-odds of not achieving the outcome at a given point in time conditioned on the outcome not having been achieved yet, $\frac{p_{\Delta_{t+1}}(\Delta_{t+1}=0)}{1-p_{\Delta_{t+1}}(\Delta_{t+1}=0)}$.*

## A.4 Outcome-Driven Policy Iteration

**Theorem 2** (Variational Outcome-Driven Policy Iteration)**.** *Assume $|\mathcal{A}| < \infty$ and that the MDP is ergodic.*

1. *Outcome-Driven Policy Evaluation (ODPE): Given policy $\pi$ and a function $Q^0 : \mathcal{S} \times \mathcal{A} \times \mathcal{S} \to \mathbb{R}$, define $Q^{i+1} = \mathcal{T}^\pi Q^i$. Then the sequence $Q^i$ converges to the lower bound in [Theorem 1].*

2. *Outcome-Driven Policy Improvement (ODPI): The policy*

$$\pi^+ = \arg\max_{\pi'\in\Pi}\left\{\mathbb{E}_{\pi'(\mathbf{a}_t\,|\,\mathbf{s}_t)}[Q^\pi(\mathbf{s}_t,\mathbf{a}_t,\mathbf{g};q_T)] - \mathbb{D}_{\mathrm{KL}}(\pi'(\cdot\,|\,\mathbf{s}_t)\,\|\,p(\cdot\,|\,\mathbf{s}_t))\right\}$$
(A.86)

*and the variational distribution over $T$ recursively defined in terms of*

$$q^+(\Delta_{t+1}=0\,|\,\mathbf{s}_0;\pi,Q^\pi)$$
$$= \sigma\Big(\mathbb{E}_{\pi(\mathbf{a}_{t+1}\,|\,\mathbf{s}_{t+1})p_d(\mathbf{s}_{t+1}\,|\,\mathbf{s}_t,\mathbf{a}_t)}[Q^\pi(\mathbf{s}_{t+1},\mathbf{a}_{t+1},\mathbf{g};q_T)] - \mathbb{E}_{\pi(\mathbf{a}_t\,|\,\mathbf{s}_t)}[\log p_d(\mathbf{g}\,|\,\mathbf{s}_t,\mathbf{a}_t)]$$
$$+ \sigma^{-1}\left(p_{\Delta_{t+1}}(\Delta_{t+1}=0)\right)\Big)$$
(A.87)

*improve the variational objective. In other words, $\mathcal{F}(\pi^+,q_T,\mathbf{s}_0) \geq \mathcal{F}(\pi,q_T,\mathbf{s}_0)$ and $\mathcal{F}(\pi,q_T^+,\mathbf{s}_0) \geq \mathcal{F}(\pi,q_T,\mathbf{s}_0)$ for all $\mathbf{s}_0 \in \mathcal{S}$.*

3. *Alternating between ODPE and ODPI converges to a policy $\pi^\star$ and a variational distribution over $T$, $q_T^\star$, such that $Q^{\pi^\star}(\mathbf{s},\mathbf{a},\mathbf{g};q_T^\star) \geq Q^\pi(\mathbf{s},\mathbf{a},\mathbf{g};q_T)$ for all $(\pi,q_T) \in \Pi \times \mathcal{Q}_T$ and any $(\mathbf{s},\mathbf{a}) \in \mathcal{S} \times \mathcal{A}$.*

*Proof.* Parts of this proof are adapted from the proof given in Haarnoja et al. [16], modified for the Bellman operator proposed in [Definition 1].

1. Outcome-Driven Policy Evaluation (ODPE): Instead of absorbing the entropy term into the $Q$-function, we can define an entropy-augmented reward as

$$r^\pi(\mathbf{s}_t,\mathbf{a}_t,\mathbf{g};q_\Delta) \doteq q_{\Delta_{t+1}}(\Delta_{t+1}=1)\log p_d(\mathbf{g}\,|\,\mathbf{s}_t,\mathbf{a}_t) - \mathbb{D}_{\mathrm{KL}}(q_{\Delta_{t+1}}\,\|\,p_{\Delta_{t+1}})$$
$$+ q_{\Delta_{t+1}}(\Delta_{t+1}=0)\,\mathbb{E}_{p_d(\mathbf{s}_{t+1}\,|\,\mathbf{s}_t,\mathbf{a}_t)}[\mathbb{D}_{\mathrm{KL}}(\pi(\cdot\,|\,\mathbf{s}_{t+1})\,\|\,p(\cdot\,|\,\mathbf{s}_{t+1}))].$$
(A.88)

We can then write an update rule according to [Definition 1] as

$$\tilde{Q}(\mathbf{s}_t,\mathbf{a}_t,\mathbf{g};q_T) \leftarrow r^\pi(\mathbf{s}_t,\mathbf{a}_t,\mathbf{g};q_\Delta)$$
$$+ q_{\Delta_{t+1}}(\Delta_{t+1}=0)\,\mathbb{E}_{\pi(\mathbf{a}_{t+1}\,|\,\mathbf{s}_{t+1})p_d(\mathbf{s}_{t+1}\,|\,\mathbf{s}_t,\mathbf{a}_t)}[\tilde{Q}(\mathbf{s}_{t+1},\mathbf{a}_{t+1},\mathbf{g};q_T)],$$
(A.89)

where $q_{\Delta_{t+1}}(\Delta_{t+1} = 0) \leq 1$. This update is similar to a Bellman update [43], but with a discount factor given by $q_{\Delta_{t+1}}(\Delta_{t+1} = 0)$. In general, this discount factor $q_{\Delta_{t+1}}(\Delta_{t+1} = 0)$ can be computed dynamically based on the current state and action, such as in Equation (11). As discussed in White [53], this Bellman operator is still a contraction mapping so long as the Markov chain induced by the current policy is ergodic and there exists a state such that $q_{\Delta_{t+1}}(\Delta_{t+1} = 0) < 1$. The first condition is true by assumption. The second condition is true since $q_{\Delta_{t+1}}(\Delta_{t+1} = 0)$ is given by Equation (11), which is always strictly between 0 and 1. Therefore, we apply convergence results for policy evaluation with transition-dependent discount factors [53] to this contraction mapping, and the result immediately follows.

2. Outcome-Driven Policy Improvement (ODPI): Let $\pi_{\text{old}} \in \Pi$ and let $Q^{\pi_{\text{old}}}$ and $V^{\pi_{\text{old}}}$ be the outcome-driven state and state-action value functions from Definition 1, let $q_T$ be some variational distribution over $T$, and let $\pi_{\text{new}}$ be given by

$$\pi_{\text{new}}(\mathbf{a}_t|\mathbf{s}_t) = \arg\max_{\pi' \in \Pi} \left\{ \mathbb{E}_{\pi'(\mathbf{a}_t \mid \mathbf{s}_t)} \left[ Q^{\pi_{\text{old}}}(\mathbf{s}_t, \mathbf{a}_t, \mathbf{g}; q_T) \right] - \mathbb{D}_{\text{KL}}(\pi'(\cdot \mid \mathbf{s}_t) \,\|\, p(\cdot \mid \mathbf{s}_t)) \right\} \tag{A.90}$$

$$= \arg\max_{\pi' \in \Pi} \mathcal{J}_{\pi_{\text{old}}}(\pi'(\mathbf{a}_t, \mathbf{s}_t), q_T). \tag{A.91}$$

Then, it must be true that $\mathcal{J}_{\pi_{\text{old}}}(\pi_{\text{old}}(\mathbf{a}_t|\mathbf{s}_t); q_T) \leq \mathcal{J}_{\pi_{\text{old}}}(\pi_{\text{new}}(\mathbf{a}_t|\mathbf{s}_t); q_T)$, since one could set $\pi_{\text{new}} = \pi_{\text{old}} \in \Pi$. Thus,

$$\begin{aligned}
\mathbb{E}_{\pi_{\text{new}}(\mathbf{a}_t|\mathbf{s}_t)} & \left[ Q^{\pi_{\text{old}}}(\mathbf{s}_t, \mathbf{a}_t, \mathbf{g}; q_T) \right] - \mathbb{D}_{\text{KL}}(\pi_{\text{new}}(\cdot \mid \mathbf{s}_t) \,\|\, p(\cdot \mid \mathbf{s}_t)) \\
& \geq \mathbb{E}_{\pi_{\text{old}}(\mathbf{a}_t|\mathbf{s}_t)} \left[ Q^{\pi_{\text{old}}}(\mathbf{s}_t, \mathbf{a}_t, \mathbf{g}; q_T) \right] - \mathbb{D}_{\text{KL}}(\pi_{\text{old}}(\cdot \mid \mathbf{s}_t) \,\|\, p(\cdot \mid \mathbf{s}_t)),
\end{aligned} \tag{A.92}$$

and since

$$V^{\pi_{\text{old}}}(\mathbf{s}_t, \mathbf{g}; q_T) = \mathbb{E}_{\pi_{\text{old}}(\mathbf{a}_t|\mathbf{s}_t)} \left[ Q^{\pi_{\text{old}}}(\mathbf{s}_t, \mathbf{a}_t, \mathbf{g}; q_T) \right] - \mathbb{D}_{\text{KL}}(\pi_{\text{old}}(\cdot \mid \mathbf{s}_t) \,\|\, p(\cdot \mid \mathbf{s}_t)), \tag{A.93}$$

we get

$$\mathbb{E}_{\pi_{\text{new}}(\mathbf{a}_t|\mathbf{s}_t)} \left[ Q^{\pi_{\text{old}}}(\mathbf{s}_t, \mathbf{a}_t, \mathbf{g}; q_T) \right] - \mathbb{D}_{\text{KL}}(\pi_{\text{new}}(\cdot \mid \mathbf{s}_t) \,\|\, p(\cdot \mid \mathbf{s}_t)) \geq V^{\pi_{\text{old}}}(\mathbf{s}_t, \mathbf{g}; q_T). \tag{A.94}$$

We can now write the Bellman equation as

$$\begin{aligned}
& Q^{\pi_{\text{old}}}(\mathbf{s}_t, \mathbf{a}_t, \mathbf{g}; q_T) \\
& = q_{\Delta_{t+1}}(\Delta_{t+1} = 1) \log p_d(\mathbf{g} \mid \mathbf{s}_t, \mathbf{a}_t) + q_{\Delta_{t+1}}(\Delta_{t+1} = 0) \mathbb{E}_{p_d(\mathbf{s}_{t+1} \mid \mathbf{s}_t, \mathbf{a}_t)} [V^{\pi_{\text{old}}}(\mathbf{s}_{t+1}, \mathbf{g}; q_T)]
\end{aligned} \tag{A.95}$$

$$\begin{aligned}
& \leq q_{\Delta_{t+1}}(\Delta_{t+1} = 1) \log p_d(\mathbf{g} \mid \mathbf{s}_t, \mathbf{a}_t) \\
& \quad + q_{\Delta_{t+1}}(\Delta_{t+1} = 0) \mathbb{E}_{p(\mathbf{s}_{t'} \mid \mathbf{s}_t, \mathbf{a}_t)} [\mathbb{E}_{\pi_{\text{new}}(\mathbf{a}_{t'} \mid \mathbf{s}_{t'})} [Q^{\pi_{\text{old}}}(\mathbf{s}_{t'}, \mathbf{a}_{t'}, \mathbf{g}; q_T)] \\
& \quad - \mathbb{D}_{\text{KL}}(\pi_{\text{new}}(\cdot \mid \mathbf{s}_{t'}) \,\|\, p(\cdot \mid \mathbf{s}_{t'}))],
\end{aligned} \tag{A.96}$$

$$\vdots$$

$$\leq Q^{\pi_{\text{new}}}(\mathbf{s}_t, \mathbf{a}_t, \mathbf{g}; q_T) \tag{A.97}$$

where we defined $t' \doteq t + 1$, repeatedly applied the Bellman backup operator defined in Definition 1 and used the bound in Equation (A.94). Convergence follows from Outcome-Driven Policy Evaluation above.

3. Locally Optimal Variational Outcome-Driven Policy Iteration: Define $\pi^i$ to be a policy at iteration $i$. By ODPI for a given $q_T$, the sequence of state-action value functions $\{Q^{\pi^i}(q_T)\}_{i=1}^{\infty}$ is monotonically increasing in $i$. Since the reward is finite and the negative KL divergence is upper bounded by zero, $Q^\pi(q_T)$ is upper bounded for $\pi \in \Pi$ and the sequence $\{\pi^i\}_{i=1}^{\infty}$ converges to some $\pi^\star$. To see that $\pi^\star$ is an optimal policy, note that it must be the case that $\mathcal{J}_{\pi^\star}(\pi^\star(\mathbf{a}_t|\mathbf{s}_t); q_T) > \mathcal{J}_{\pi^\star}(\pi(\mathbf{a}_t \mid \mathbf{s}_t); q_T)$ for any $\pi \in \Pi$ with $\pi \neq \pi^\star$. By the argument used in ODPI above, it must be the case that the outcome-driven state-action value of the converged policy is higher than that of any other non-converged policy in $\Pi$, that is, $Q^{\pi^\star}(\mathbf{s}_t, \mathbf{a}_t; q_T) > Q^\pi(\mathbf{s}_t, \mathbf{a}_t; q_T)$ for all $\pi \in \Pi$ and any $q_T^i \in \mathcal{Q}_T$ and $(\mathbf{s}, \mathbf{a}) \in \mathcal{S} \times \mathcal{A}$. Therefore, given $q_T$, $\pi^\star$ must be optimal in $\Pi$, which concludes the proof.

4. Globally Optimal Variational Outcome-Driven Policy Iteration: Let $\pi^i$ be a policy and let $q_T^i$ be variational distributions over $T$ at iteration $i$. By Locally Optimal Variational Outcome-Driven Policy Iteration, for a *fixed* $q_T^i$ with $q_T^i = q_T^j \forall i, j \in \mathbb{N}_0$, the sequence of $\{(\pi^i, q_T^i)\}_{i=1}^{\infty}$ increases the objective Equation (A.16) at each iteration and converges to a stationary point in $\pi^i$, where $Q^{\pi^\star}(\mathbf{s}_t, \mathbf{a}_t; q_T^i) > Q^\pi(\mathbf{s}_t, \mathbf{a}_t; q_T^i)$ for all $\pi \in \Pi$ and any $q_T^i \in \mathcal{Q}_T$ and $(\mathbf{s}, \mathbf{a}) \in \mathcal{S} \times \mathcal{A}$. Since the objective in Equation (A.16) is concave in $q_T$, it must be the case that for, $q_T^{\star^i} \in \mathcal{Q}_T$, the optimal variational distribution over $T$ at iteration $i$, defined recursively by

$$\begin{aligned}
q^{\star^i}(\Delta_{t+1} = 0; \pi^i, Q^{\pi^i}) = \sigma \Big( & \mathbb{E}_{\pi(\mathbf{a}_{t+1} \mid \mathbf{s}_{t+1}) p_d(\mathbf{s}_{t+1} \mid \mathbf{s}_t, \mathbf{a}_t)} [Q^{\pi^i}(\mathbf{s}_{t+1}, \mathbf{a}_{t+1}, \mathbf{g}; q_T(\pi^i, Q^{\pi^i}))] \\
& - \mathbb{E}_{\pi(\mathbf{a}_t \mid \mathbf{s}_t)} [\log p_d(\mathbf{g} \mid \mathbf{s}_t, \mathbf{a}_t)] + \sigma^{-1}(p_{\Delta_{t+1}}(\Delta_{t+1} = 0)) \Big),
\end{aligned}$$

for $t \in \mathbb{N}_0$, $Q^\pi(\mathbf{s}_t, \mathbf{a}_t; q_T^\star) > Q^\pi(\mathbf{s}_t, \mathbf{a}_t; q_T)$ for all $\pi \in \Pi$ and any $(\mathbf{s}, \mathbf{a}) \in \mathcal{S} \times \mathcal{A}$. Note that $q_T$ is defined implicitly in terms of $\pi^i$ and $Q^{\pi^i}$, that is, the optimal variational distribution over $T$ at iteration $i$ is defined as a function of the policy and $Q$-function at iteration $i$. Hence, it must then be true that for $Q^{\pi^\star}(\mathbf{s}_t, \mathbf{a}_t; q_T^\star) > Q^{\pi^\star}(\mathbf{s}_t, \mathbf{a}_t; q_T)$ for all $q_T^\star(\pi^\star, Q^{\pi^\star}) \in \mathcal{Q}_T$ and for any $\pi^\star \in \Pi$ and $(\mathbf{s}, \mathbf{a}) \in \mathcal{S} \times \mathcal{A}$. In other words, for an optimal policy and corresponding $Q$-function, there exists an optimal variational distribution over $T$ that maximizes the $Q$-function, given the optimal policy. Repeating locally optimal variational outcome-driven policy iteration under the new variational distribution $q_T^\star(\pi^\star, Q^{\pi^\star})$ will yield an optimal policy $\pi^{\star\star}$ and computing the corresponding optimal variational distribution, $q_T^{\star\star}(\pi^{\star\star}, Q^{\pi^{\star\star}})$ will further increase the variational objective such that for $\pi^{\star\star}) \in \Pi$ and $q_T^{\star\star}(\pi^{\star\star}, Q^{\pi^{\star\star}}) \in \mathcal{Q}_T$, we have that

$$Q^{\pi^{\star\star}}(\mathbf{s}_t, \mathbf{a}_t; q_T^{\star\star}) > Q^{\pi^{\star\star}}(\mathbf{s}_t, \mathbf{a}_t; q_T^\star) > Q^{\pi^\star}(\mathbf{s}_t, \mathbf{a}_t; q_T^\star) > Q^{\pi^\star}(\mathbf{s}_t, \mathbf{a}_t; q_T) \qquad \text{(A.98)}$$

for any $\pi^\star \in \Pi$ and $(\mathbf{s}, \mathbf{a}) \in \mathcal{S} \times \mathcal{A}$. Hence, global optimal variational outcome-driven policy iteration increases the variational objective at every step. Since the objective is upper bounded (by virtue of the rewards being finite and the negative KL divergence being upper bounded by zero) and the sequence of $\{(\pi^i, q_T^i)\}_{i=1}^\infty$ increases the objective Equation (A.16) at each iteration, by the monotone convergence theorem, the objective value converges to a supremum and since the objective function is concave the supremum is unique. Hence, since the supremum is unique and obtained via global optimal variational outcome–driven policy iteration on $(\pi, q_T) \in \Pi \times \mathcal{Q}_T$, the sequence of $\{(\pi^i, q_T^i)\}_{i=1}^\infty$ converges to a unique stationary point $(\pi^\star, q_T^\star) \in \Pi \times \mathcal{Q}_T$, where $Q^{\pi^\star}(\mathbf{s}_t, \mathbf{a}_t; q_T^i) > Q^\pi(\mathbf{s}_t, \mathbf{a}_t; q_T^i)$ for all $\pi \in \Pi$ and any $q_T^i \in \mathcal{Q}_T$ and $(\mathbf{s}, \mathbf{a}) \in \mathcal{S} \times \mathcal{A}$.

$\square$

**Corollary 3** (Optimality of Variational Outcome Driven Policy Iteration). *Variational Outcome-Driven Policy Iteration on $(\pi, q_T) \in \Pi \times \mathcal{Q}_T$ results in an optimal policy at least as good or better than any optimal policy attainable from policy iteration on $\pi \in \Pi$ alone.*

**Remark 2.** *The convergence proof of ODPE assumes a transition-dependent discount factor [53], because the variational distribution used in Equation (11) depends on the next state and action as well as on the desired outcome.*

## A.5  Lemmas

**Lemma 1.** *Let $q(T = t) \doteq q(T = t | T \geq t) \prod_{i=1}^t q(T \neq i - 1 | T \geq i - 1)$ be a discrete probability distribution with support $\mathbb{N}_0$. Then for any $t \in \mathbb{N}_0$, we have that*

$$q(T \geq t) = \sum_{i=t}^\infty q(T = i | T \geq i) \prod_{j=1}^i q(T \neq j - 1 | T \geq j - 1) = \prod_{i=1}^t q(T \neq i - 1 | T \geq i - 1). \qquad \text{(A.99)}$$

*Proof.* We proof the statement by induction on $t$.

Base case: For $t = 0$, $q(T \geq 0) = 1$ by definition of the empty product.

Inductive case: Note that $q(T \leq t) = \prod_{i=1}^t q(T = i - 1 | T \geq i - 1)$. Show that

$$q(T \geq t) = \prod_{i=1}^t q(T \neq i - 1 | T \geq i - 1) \implies q(T \geq t + 1) = \prod_{i=1}^{t+1} q(T \neq i - 1 | T \geq i - 1). \qquad \text{(A.100)}$$

Consider $q(T \geq t + 1) = \sum_{i=t+1}^\infty q(T = i | T \geq i) \prod_{j=1}^i q(T \neq j - 1 | T \geq j - 1)$. To proof the inductive hypothesis, we need to show that the following equality is true:

$$\sum_{i=t+1}^\infty q(T = i | T \geq i) \prod_{j=1}^i q(T \neq j - 1 | T \geq j - 1) = \prod_{i=1}^{t+1} q(T \neq i - 1 | T \geq i - 1) \qquad \text{(A.101)}$$

$$\Longleftrightarrow \sum_{i=t}^\infty q(T = i | T \geq i) \prod_{j=1}^i q(T \neq j - 1 | T \geq j - 1) - q(T = t | T \geq t) \prod_{j=1}^t q(T \neq j - 1 | T \geq j - 1)$$

$$= q(T \neq t | T \geq t) \prod_{i=1}^t q(T \neq i - 1 | T \geq i - 1).$$

$$\text{(A.102)}$$

By the inductive hypothesis,

$$q(T \geq t) = \sum_{i=t}^{\infty} q(T = i | T \geq i) \prod_{j=1}^{i} q(T \neq j - 1 | T \geq j - 1) = \prod_{i=1}^{t} q(T \neq i - 1 | T \geq i - 1), \quad \text{(A.103)}$$

and so

$$\text{Equation (A.102)} \iff \prod_{j=1}^{t} q(T \neq j | T \geq j) - q(T \neq t + 1 | T \geq t + 1) \prod_{j=1}^{t} q(T = j | T \geq j) \quad \text{(A.104)}$$

$$= q(T \neq t | T \geq t) \prod_{i=1}^{t} q(T \neq i - 1 | T \geq i - 1). \quad \text{(A.105)}$$

Factoring out $\prod_{i=1}^{t} q(T \neq i - 1 | T \geq i - 1)$, we get

$$\iff \prod_{j=1}^{t} q(T \neq j - 1 | T \geq j - 1) \underbrace{(1 - q(T = t | T \geq t))}_{= q(T \neq t | T \geq t)} = q(T \neq t | T \geq t) \prod_{j=1}^{t} q(T = j - 1 | T \geq j - 1)$$

$$\text{(A.106)}$$

$$\iff q(T \neq t | T \geq t) \prod_{j=1}^{t} q(T \neq j - 1 | T \geq j - 1) = q(T \neq t | T \geq t) \prod_{j=1}^{t} q(T \neq j - 1 | T \geq j - 1),$$

$$\text{(A.107)}$$

which proves the inductive hypothesis. $\qquad \square$

**Lemma 2.** *Let $q_T(t)$ and $p_T(t)$ be discrete probability distributions with support $\mathbb{N}_0$, let $\Delta_t$ be a Bernoulli random variable, with success defined as $T = t + 1$ given that $T \geq t$, and let $q_{\Delta_t}$ be a discrete probability distribution over $\Delta_t$ for $t \in \mathbb{N} \backslash \{0\}$, so that*

$$\begin{aligned} q_{\Delta_{t+1}}(\Delta_{t+1} = 0) &\doteq q(T \neq t \,|\, T \geq t) \\ q_{\Delta_{t+1}}(\Delta_{t+1} = 1) &\doteq q(T = t \,|\, T \geq t). \end{aligned} \quad \text{(A.108)}$$

*Then we can write $q(T = t) = q_{\Delta_{t+1}}(\Delta_{t+1} = 1) \prod_{i=1}^{t} q_{\Delta_i}(\Delta_i = 0)$ for any $t \in \mathbb{N}_0$ and have that*

$$q(T \geq t) = \sum_{i=t}^{\infty} q_{\Delta_{i+1}}(\Delta_{i+1} = 1) \prod_{j=1}^{i} q_{\Delta_j}(\Delta_j = 0) = \prod_{i=1}^{t} q_{\Delta_i}(\Delta_i = 0). \quad \text{(A.109)}$$

*Proof.* By Lemma 1, we have that for any $t \in \mathbb{N}_0$

$$q(T \geq t) = \sum_{i=t}^{\infty} q(T = i | T \geq i) \prod_{j=1}^{i} q(T \neq j - 1 | T \geq j - 1) = \prod_{i=1}^{t} q(T \neq i - 1 | T \geq i - 1). \quad \text{(A.110)}$$

The result follows by replacing $q(T = i | T \geq i)$ by $q_{\Delta_{i+1}}(\Delta_{i+1} = 1)$, $q(T \neq j - 1 | T \geq j - 1)$ by $q_{\Delta_j}(\Delta_j = 0)$, and $q(T \neq i - 1 | T \geq i - 1)$ by $q_{\Delta_i}(\Delta_i = 0)$. $\qquad \square$

**Lemma 3.** *Let $q_T(t)$ and $p_T(t)$ be discrete probability distributions with support $\mathbb{N}_0$. Then for any $k \in \mathbb{N}_0$,*

$$\begin{aligned} &\mathbb{E}_{t \sim q(T \,|\, T \geq k)} \left[ \log \frac{q(T = t \,|\, T \geq k)}{p(T = t \,|\, T \geq k)} \right] \\ &= f(q, p, k) + q(T \neq k \,|\, T \geq k) \, \mathbb{E}_{t \sim q(T \,|\, T \geq k+1)} \left[ \log \frac{q(T = t \,|\, T \geq k + 1)}{p(T = t \,|\, T \geq k + 1)} \right]. \end{aligned} \quad \text{(A.111)}$$

*Proof.* Consider $\mathbb{E}_{t \sim q(T \,|\, T \geq k)} \left[ \log \frac{q(T=t \,|\, T \geq k)}{p(T=t \,|\, T \geq k)} \right]$ and note that by the law of total expectation we can rewrite it as

$$\begin{aligned} &\mathbb{E}_{t \sim q(T \,|\, T \geq k)} \left[ \log \frac{q(T = t \,|\, T \geq k)}{p(T = t \,|\, T \geq k)} \right] \\ &= q(T = k \,|\, T \geq k) \, \mathbb{E}_{t \sim q(T \,|\, T = k)} \left[ \log \frac{q(T = t \,|\, T \geq k)}{p(T = t \,|\, T \geq k)} \right] \\ &\quad + q(T \neq k \,|\, T \geq k) \, \mathbb{E}_{t \sim q(T \,|\, T \geq k+1)} \left[ \log \frac{q(T = t \,|\, T \geq k)}{p(T = t \,|\, T \geq k)} \right] \\ &= q(T = k \,|\, T \geq k) \log \frac{q(T = k \,|\, T \geq k)}{p(T = k \,|\, T \geq k)} + q(T \neq k \,|\, T \geq k) \, \mathbb{E}_{t \sim q(T | T \geq k+1)} \left[ \log \frac{q(T = t \,|\, T \geq k)}{p(T = t \,|\, T \geq k)} \right]. \end{aligned} \quad \text{(A.112)}$$

$$\text{(A.113)}$$

For all values of $T \geq k + 1$, we have that

$$q(T = t \,|\, T \geq k) = q(T = t \,|\, T \geq k + 1)q(T \neq k \,|\, T \geq k) \qquad \text{(A.114)}$$

$$p(T = t \,|\, T \geq k) = p(T = t \,|\, T \geq k + 1)p(T \neq k \,|\, T \geq k) \qquad \text{(A.115)}$$

and so we can rewrite the expectation in Equation (A.113) as

$$\mathbb{E}_{t \sim q(T \,|\, T \geq k+1)} \left[ \log \frac{q(T = t \,|\, T \geq k)}{p(T = t \,|\, T \geq k)} \right] = \mathbb{E}_{t \sim q(T \,|\, T \geq k+1)} \left[ \log \frac{q(T = t \,|\, T \geq k)}{p(T = t \,|\, T \geq k)} + \log \frac{q(T \neq k \,|\, T \geq k)}{p(T \neq k \,|\, T \geq k)} \right]$$
$$\text{(A.116)}$$

$$= \mathbb{E}_{t \sim q(T \,|\, T \geq k+1)} \left[ \log \frac{q(T = t \,|\, T \geq k)}{p(T = t \,|\, T \geq k)} \right] + \log \frac{q(T \neq k \,|\, T \geq k)}{p(T \neq k \,|\, T \geq k)}$$
$$\text{(A.117)}$$

Combining Equation (A.117) with Equation (A.113), we have

$$\mathbb{E}_{t \sim q(T \,|\, T \geq k)} \left[ \log \frac{q(T = t \,|\, T \geq k)}{p(T = t \,|\, T \geq k)} \right]$$

$$= \underbrace{q(T = k \,|\, T \geq k) \log \frac{q(T = k \,|\, T \geq k)}{p(T = k \,|\, T \geq k)} + q(T \neq k \,|\, T \geq k) \log \frac{q(T \neq k \,|\, T \geq k)}{p(T \neq k \,|\, T \geq k)}}_{\doteq\, f(q,p,k)} \qquad \text{(A.118)}$$

$$+ q(T \neq k \,|\, T \geq k) \, \mathbb{E}_{t \sim q(T \,|\, T \geq k+1)} \left[ \log \frac{q(T = t \,|\, T \geq k + 1)}{p(T = t \,|\, T \geq k + 1)} \right],$$

which concludes the proof. $\qquad \square$

**Lemma 4.** *Let $q_T(t)$ and $p_T(t)$ be discrete probability distributions with support $\mathbb{N}_0$. Then the KL divergence from $q_T$ to $p_T$ can be written as*

$$\mathbb{D}_{\text{KL}}(q_T \,\|\, p_T) = \sum_{t=0}^{\infty} q(T \geq t) f(q_T, p_T, t) \qquad \text{(A.119)}$$

*where $f(q_T, p_T, t)$ is shorthand for*

$$f(q_T, p_T, t) = q(T = t \,|\, T \geq t) \log \frac{q(T = t \,|\, T \geq t)}{p(T = t \,|\, T \geq t)} + q(T \neq t \,|\, T \geq t) \log \frac{q(T \neq t \,|\, T \geq t)}{p(T \neq t \,|\, T \geq t)}. \qquad \text{(A.120)}$$

*Proof.* Note that $q(T = k)$ denotes the probability that the distribution $q$ assigns to the event $T = k$ and $q(T \geq m)$ denotes the tail probability, that is, $q(T \geq m) = \sum_{t=m}^{\infty} q(T = t)$. We will write $q(T|T \geq m)$ to denote the conditional distribution of $q$ given $T \geq m$, that is, $q(T = k | T \geq m) = \mathbb{1}[k \geq m] q(T = k) / q(T \geq m)$. We will use analogous notation for $p$.

By the definition of the KL divergence and using the fact that, since the support is lowerbounded by $T = 0$, $q(T = 0) = q(T = 0 \,|\, T \geq 0)$, we have

$$\mathbb{D}_{\text{KL}}(q_T \,\|\, p_T) = \mathbb{E}_{t \sim q(T)} \left[ \log \frac{q(T = t)}{p(T = t)} \right] = \mathbb{E}_{t \sim q(T \,|\, T \geq 0)} \left[ \log \frac{q(T = t \,|\, T \geq 0)}{p(T = t \,|\, T \geq 0)} \right]. \qquad \text{(A.121)}$$

Using Lemma 3 with $k = 0, 1, 2, 3, \ldots$, we can expand the above expression to get

$$\mathbb{D}_{\text{KL}}(q_T \parallel p_T) \tag{A.122}$$

$$= f(q_T, p_T, 0) + q(T \neq 0 \mid T \geq 0) \, \mathbb{E}_{t \sim q(T \mid T \geq 1)} \left[ \log \frac{q(T = t \mid T \geq 1)}{p(T = t \mid T \geq 1)} \right] \tag{A.123}$$

$$= f(q, p, 0) + q(T \neq 0 \mid T \geq 1) f(q_T, p_T, 1)$$
$$\qquad + q(T \neq 0 \mid T \geq 0) q(T \neq 1 \mid T \geq 1) \, \mathbb{E}_{t \sim q(T \mid T \geq 2)} \left[ \log \frac{q(T = t \mid T \geq 2)}{p(T = t \mid T \geq 2)} \right] \tag{A.124}$$

$$= \underbrace{1}_{=q(T \geq 0)} \cdot f(q, p, 0)$$
$$\qquad + \underbrace{q(T \neq 0 \mid T \geq 0)}_{=q(T \geq 1)} f(q, p, 1)$$
$$\qquad + \underbrace{q(T \neq 0 \mid T \geq 0) q(T \neq 1 \mid T \geq 1)}_{=q(T \geq 2)} f(q_T, p_T, 2)$$
$$\qquad + \underbrace{q(T \neq 0 \mid T \geq 0) q(T \neq 1 \mid T \geq 1) q(T \neq 2 \mid T \geq 2)}_{=q(T \geq 3)} \mathbb{E}_{t \sim q(T \mid T \geq 3)} \left[ \log \frac{q(T = t \mid T \geq 3)}{p(T = t \mid T \geq 3)} \right]$$

$$\tag{A.125}$$

$$= \sum_{t=0}^{\infty} q(T \geq t) f(q_T, p_T, t), \tag{A.126}$$

where $f(q_T, p_T, t)$ is shorthand for

$$f(q_T, p_T, t) = q(T = t \mid T \geq t) \log \frac{q(T = t \mid T \geq t)}{p(T = t \mid T \geq t)} + q(T \neq t \mid T \geq t) \log \frac{q(T \neq t \mid T \geq t)}{p(T \neq t \mid T \geq t)}. \tag{A.127}$$

and we used the fact that, by Lemma 1,

$$q(T \geq t) = \prod_{k=1}^{t} q(T \neq k - 1 \mid T \geq k - 1). \tag{A.128}$$

This completes the proof. $\qquad\square$

**Lemma 5.** *Let $q_T(t)$ and $p_T(t)$ be discrete probability distributions with support $\mathbb{N}_0$, let $\Delta_t$ be a Bernoulli random variable, with success defined as $T = t$ given that $T \geq t$, and let $q_{\Delta_t}$ and $p_{\Delta_t}$ be discrete probability distributions over $\Delta_t$ for $t \in \mathbb{N}_0 \backslash \{0\}$, so that*

$$q_{\Delta_{t+1}}(\Delta_{t+1} = 0) \doteq q(T \neq t \mid T \geq t) \qquad q_{\Delta_{t+1}}(\Delta_{t+1} = 1) \doteq q(T = t \mid T \geq t) \tag{A.129}$$

$$p_{\Delta_{t+1}}(\Delta_{t+1} = 0) \doteq p(T \neq t \mid T \geq t) \qquad p_{\Delta_{t+1}}(\Delta_{t+1} = 1) \doteq p(T = t \mid T \geq t). \tag{A.130}$$

*Then the KL divergence from $q_T$ to $p_T$ can be written as*

$$\mathbb{D}_{\text{KL}}(q_T \parallel p_T) = \sum_{t=0}^{\infty} \Big( \prod_{k=1}^{t} q_{\Delta_t}(\Delta_t = 0) \Big) \mathbb{D}_{\text{KL}}(q_{\Delta_{t+1}} \parallel p_{\Delta_{t+1}}) \tag{A.131}$$

*Proof.* The result follows from Lemma 4, Equation (A.128), Equation (A.129), and the definition of $f$.

In detail, from Lemma 1, and Equation (A.129) we have that

$$q(T \geq t) = \prod_{k=1}^{t} q(T \neq k - 1 \mid T \geq k - 1) = \prod_{k=1}^{t} q_{\Delta_k}(\Delta_k = 0). \tag{A.132}$$

From the definition of $f(q_T, p_T, t)$, we have

$$f(q_T, p_T, t) = q(T = t \mid T \geq t) \log \frac{q(T = t \mid T \geq t)}{p(T = t \mid T \geq t)} + q(T \neq t \mid T \geq t) \log \frac{q(T \neq t \mid T \geq t)}{p(T \neq t \mid T \geq t)} \tag{A.133}$$

$$= q_{\Delta_{t+1}}(\Delta_{t+1} = 0) \log \frac{q_{\Delta_{t+1}}(\Delta_{t+1} = 0)}{p_{\Delta_{t+1}}(\Delta_{t+1} = 0)} + q(\Delta_{t+1} = 1) \log \frac{q_{\Delta_{t+1}}(\Delta_{t+1} = 1)}{p_{\Delta_{t+1}}(\Delta_{t+1} = 1)} \tag{A.134}$$

$$= \mathbb{D}_{\text{KL}}(q_{\Delta_{t+1}} \parallel p_{\Delta_{t+1}}). \tag{A.135}$$

Combining Equation (A.132), Equation (A.135), and Equation (A.119) completes the proof. $\qquad\square$

# Appendix B    Additional Experiments

## B.1    Further Ablation Study Results

We show the full ablation learning curves in Figure 5. We see that ODAC consistently performs the best, and that ODAC with a fixed model also performs well. However, on a few tasks, and in particular the Fetch Push and Sawyer Faucet tasks, we see that using a fixed $q_T$ hurts the performance, suggesting that our derived formula in Equation (11) results in better empirical performance.

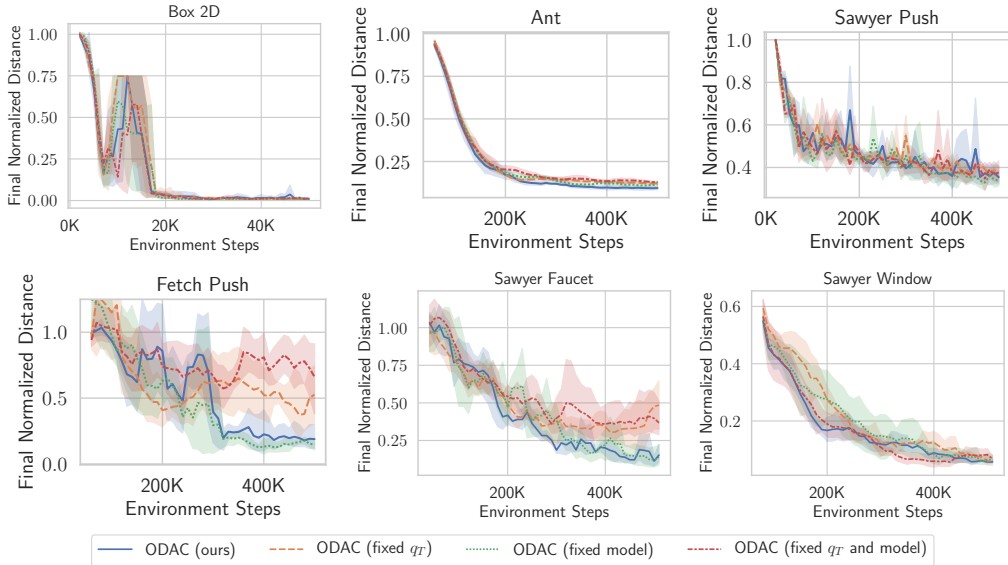

**Figure 5:** Ablation results across all six environments. We see that using our derived $q_T$ optimality equation is important for best performance across all six tasks and that ODAC is not sensitive to the quality of the dynamics model.

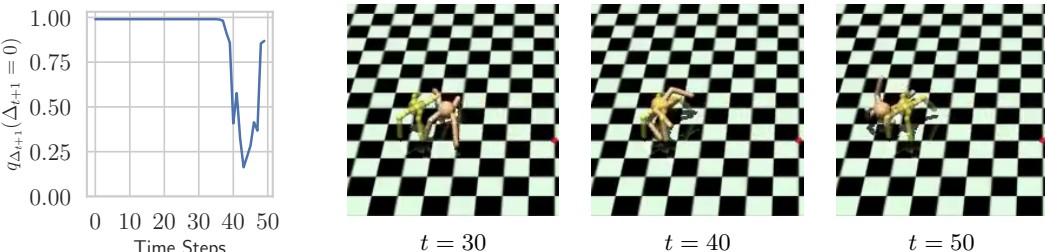

**Figure 6:** The inferred $q_{\Delta_{t+1}}(\Delta_{t+1} = 0)$ versus time during an example trajectory in the Ant environment. As the ant robot falls over, $q_{\Delta_{t+1}}(\Delta_{t+1} = 0)$ drops in value. We see that the optimal posterior $q^\star_{\Delta_{t+1}}(\Delta_{t+1} = 0)$ given in Proposition 2 automatically assigns a high likelihood of terminating when this irrecoverable state is first reached, effectively acting as a dynamic discount factor.

## B.2    Comparisons under Oracle Goal Sampling

For exploration, Andrychowicz et al. [2] explore the benefits of HER either using a single, fixed goal during exploration (see Section 4.3 of Andrychowicz et al. [2]) or using oracle goal sampling, that is, during exploration, a new goal is sampled each episode from a uniform distribution over the set of all reachable goals in the environment. As such, oracle goal sampling requires knowledge of the environment to sample several reachable goals. For example, in the 2D box experiment (Figure 3a), points inside the grey block in the center

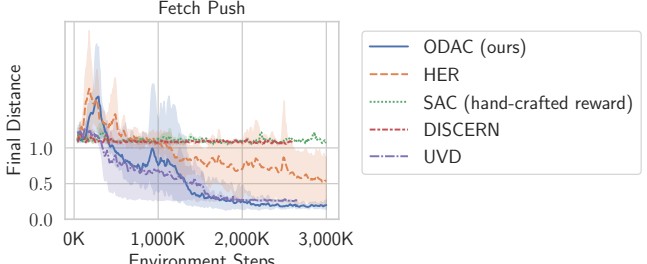

**Figure 7:** Comparison of different methods for desired outcomes **g** sampled uniformly from the set of admissible states.

are not reachable goal states, and this additional information must be available when performing oracle goal sampling.

To demonstrate the impact of sampling the desired outcome **g** during exploration, we evaluate ODAC and related methods on the Fetch task when using oracle goal sampling. As shown in Figure 7, the performances of UVD and ODAC are similar and both outperform other methods. These results suggest that UVD depends more heavily on sampling outcomes from the set of desired outcomes than ODAC. The significant decrease in performance when the desired outcome **g** is fixed may be due to the fact that uniformly sampling **g** implicitly provides a curriculum for learning. For example, in the Box 2D environment, goal states sampled above the box can train the agent to move around the obstacle, making it easier to learn how to reach the other side of the box. Without this guidance, prior methods often "get stuck" on the other side of the box. In contrast, ODAC consistently performs well in this more challenging setting, suggesting that the log-likelihood signal provides good guidance to the policy.

As shown in Figure 4, ODAC performs well on both this setting and the harder setting where the desired outcome **g** was fixed during exploration, suggesting that ODAC does not rely as heavily on the uniform sampling of **g** to learn a good policy than do other methods.

## B.3  Comparison to Model-Based Planning

ODAC learns a dynamics model but does not use it for planning and instead relies on the derived Bellman updates to obtain a policy. However, a natural question is whether or not the method would benefit from using this model to perform model-based planning, as in Janner et al. [21]. We assess this by comparing ODAC with model-based baseline that uses a 1-step look-ahead. In particular, we follow the training procedure in Janner et al. [21] with $k = 1$. To ensure a fair comparison, we use the exact same dynamics model architecture as in ODAC and match the update-to-environment step ratio to be 4-to-1 for both methods.

Table 2 shows the final distance to the goal (best results in bold). Using the same dynamics model, ODAC, which does not use the dynamics model to perform planning and only uses it to compute rewards, outperforms the model-based planning method. While a better model might lead to better performance for the model-based baseline, these results suggest that ODAC is not sensitive to model quality to the same degree as model-based planning methods.

**Table 2:** Normalized final distances (lower is better) across four random seeds, multiplied by a factor of 100.

| Environment | ODAC (Mean + Standard Error) | Dyna (Mean + Standard Error) |
|---|---|---|
| Box 2D | 0.74 (0.091) | 0.87 (0.058) |
| Ant | 33 (27) | 102 (0.83) |
| Sawyer Faucet | 14 (6.3) | 100 (5) |
| Fetch Push | 12 (3.7) | 96 (3.8) |
| Sawyer Push | 58 (8.7) | 96 (0.39) |
| Sawyer Window | 4.4 (1.5) | 116 (14) |

## B.4  Reward Visualization

We visualize the reward for the Box 2D environment in Figure 8. We see that over the course of training, the reward function initially flattens out near **g**, making learning easier by encouraging the policy to focus on moving just out of the top left corner of the environment. Later in training (around 16,000 steps), the policy learns to move out of the top left corner, and we see that the reward changes to have a stronger reward gradient near **g**. We also note that the reward are much more negative for being far **g** at the end of training: the top left region changes from having a penalty of $-1.6$ to $-107$. Overall, these visualizations show that the reward function automatically changes during training and provides a strong reward signal for different parts of the state space depending on the behavior of the policy.

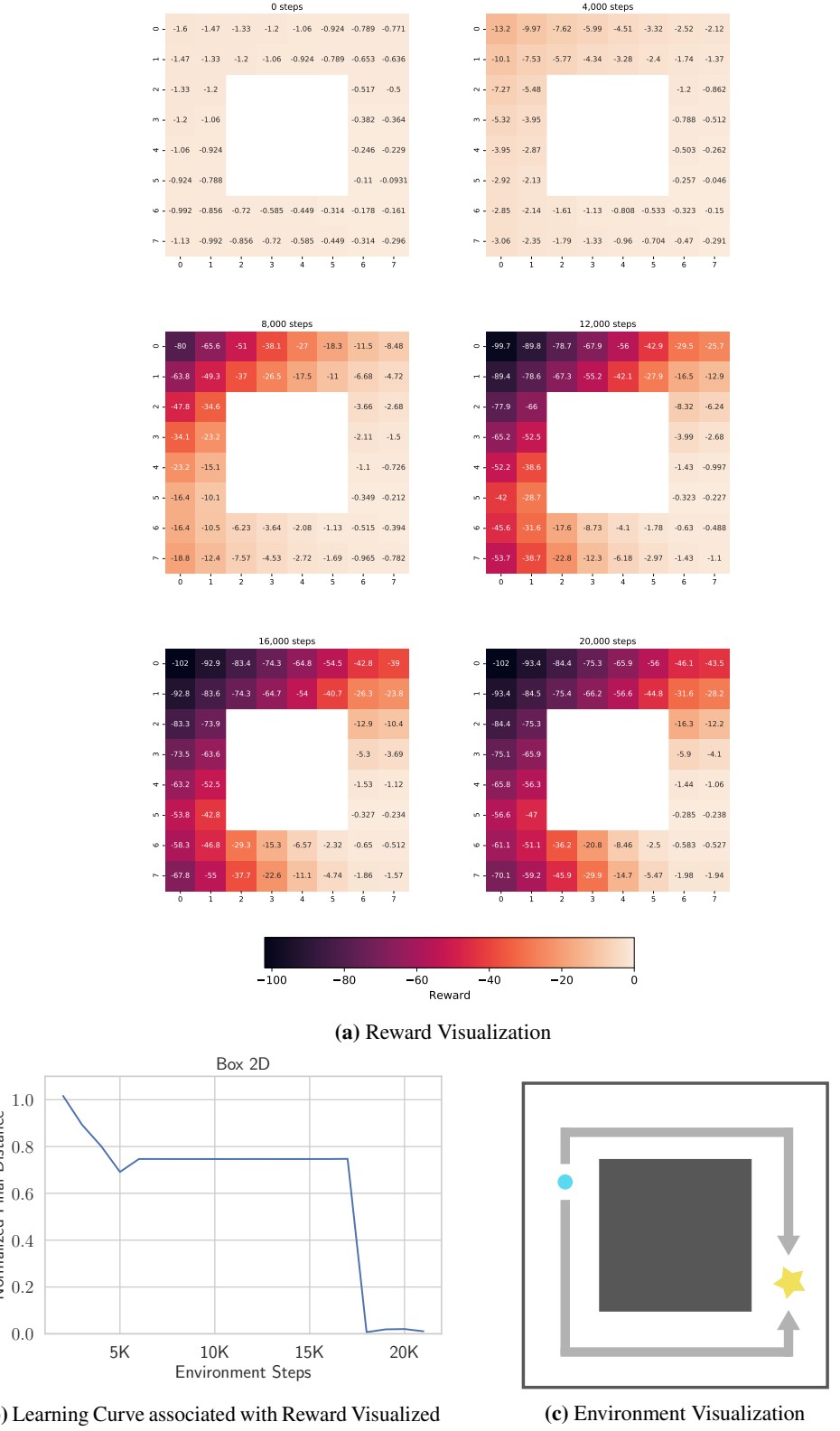

**(a)** Reward Visualization

**(b)** Learning Curve associated with Reward Visualized

**(c)** Environment Visualization

**Figure 8:** We visualize the rewards over the course of training on a single random seed for the Box 2D environment. To visualize the reward, we discretize the continuous state space and evaluate $r(\mathbf{s}_t, \mathbf{a}_t, \mathbf{g}; q_\Delta)$ for $\mathbf{a} = \vec{0}$ at different states. As shown in Figure 8c, the desired outcome $\mathbf{g}$ is near the bottom right and the states in the center are invalid. After 4-8 thousand environment steps, the reward is more flat near $\mathbf{g}$, and only provides a reward gradient far from $\mathbf{g}$. After 20 thousand environment steps, the reward gradient is much larger again near the end, and the penalty for being in the top left corner has changed from $-1.6$ to $-107$.

# Appendix C   Experimental Details

## C.1   Environment

**Ant.**   This Ant domain is based on the "Ant-V3" OpenAI Gym [4] environment, with three modifications: the gear ratio is reduced from 150 to 120, the contact force sensors are removed from the state, and there is no termination condition and the episode only terminates after a fixed amount of time. In this environment, the state space is 23 dimensional, consistent of the XYZ coordinate of the center of the torso, the orientation of the ant (in quaternion), and the angle and angular velocity of all 8 joints. The action space is 8-dimensional and corresponds to the torque to apply to each joint. The desired outcome consists of the desired XYZ, orientation, and joint angles at a position that is 5 meters down and to the right of the initial position. This desired pose is shown in Figure 4.

**Sawyer Push.**   In this environment, the state and goal space is 4 dimensional and the action space is 2 dimension. The state and goal consists of the XY end effector (EE) and the XY position of the puck. The object is on a 40cm x 50cm table and starts 20 cm in front of the hand. The goal puck position is fixed to 15 cm forward and 30 cm to the right of the initial hand position, while the goal hand position is 5cm behind and 20 cm to the right of the initial hand position. The action is the change in position in each XY direction, with a maximum change of 3 cm per direction at each time step. The episode horizon is 100.

**Sawyer Window and Faucet.**   In this environment, the state and goal space is 6 dimensional and the action space is 2 dimension. The state and goal consists of the XYZ end effector (EE) and the XYZ position of the window or faucet end endpoint. The hand is initialized away from the window and faucet. The EE goal XYZ position is set to the initial window or faucet position. The action is the change in position in each XYZ direction. For the window task, the goal positions is to close the window, and for the faucet task, the goal position is to rotate the faucet 90 degrees counter-clockwise from above.

**Box 2D.**   In this environment, the state is a $4x4$ with a $2x2$ box in the middle. The policy is initialized to to $(-3.5, -2)$ and the desired outcome is $(3.5, 2)$. The action is the XY velocity of the agent, with wall collisions taken into account and maximum velocity of 0.2 in each direction. To make the environment stochastic, we add Gaussian noise to actions with mean zero and standard deviation that's 10% of the maximum action magnitude.

**Tabular Box 2D (Figure 1).**   We implemented a tabular version of ODAC and applied it to the 2D environment shown in Figure 3a. We discretize the environment into an $8 \times 8$ grid of states. The action correspond to moving up, down, left, or right. If probability $1 - \epsilon$, this action is taken. If the agent runs into a wall or boundary, the agent stays in its current state. With probability $\epsilon = 0.1$, the commanded action is ignored and a neighboring state grid (including the current state) is uniformly sampled as the next state. The policy and $Q$-function are represented with look-up tables and randomly initialized. The entropy reward is weighted by 0.01 and the time prior $p_T$ is geometric with parameter 0.5. The dynamics model, $p_d^{(0)}$ is initialized to give a uniform probability to each states for every state and action. Each iteration, we simulate data collection by updating the dynamics model with the running average update $p_d^{(t+1)} = 0.99p_d^{(t)} + 0.01p_d$, where $p_d$ is the true dynamics and update the policy and $Q$-function according to Equation (16) and Equation (14), respectively. Figure 1 shows that, in contrast to the binary-reward setting, the learned reward provides shaping for the policy, which solves the task within 100 iterations.

## C.2   Algorithm

Pseudocode for the complete algorithm is shown in Algorithm 2.

## C.3   Implementation Details

**Dynamics model.**   For the Ant and Sawyer experiments, we train a neural network to output the mean and standard deviation of a Laplace distribution. This distribution is then used to model the distribution over the *difference* between the current state and the next state, which we found to be more reliable than predicting the next state. So, the overall distribution is given by a Laplace distribution with learned mean $\mu$ and fixed standard deviation $\sigma$ computed via

$$p_\psi = \text{Laplace}(\mu = g_\psi(\mathbf{s}, \mathbf{a}) + f(\mathbf{s}), \sigma = 0.00001)$$

where $g$ is the output of a network and $f$ is a function that maps a state into a goal.

For the 2D Navigation experiment, we use a Gaussian distribution. The dynamics neural network has hidden units of size $[64, 64]$ with a ReLU hidden activations. For the Ant and Sawyer experiments, there is no output activation. For the linear-Gaussiand and 2D Navigation experiments, we have a tanh output, so that the mean

---

**Algorithm 2** Outcome-Driven Actor Critic

---

**Require:** Policy $\pi_\theta$, Q-function $Q_\phi$, dynamics model $p_\psi$, replay buffer $\mathcal{R}$, and map from state to achieved goal $f$.

  **for** $n = 0, \ldots N - 1$ episodes **do**
    Sample initial state $\mathbf{s}_0$ from environment.
    Sample goal $\mathbf{g}$ from environment.
    **for** $t = 0, \ldots, H - 1$ steps **do**
      Get action $\mathbf{a}_t \sim \pi_\theta(\mathbf{s}_t, \mathbf{g})$.
      Get next state $\mathbf{s}_{t+1} \sim p(\cdot \,|\, \mathbf{s}_t, \mathbf{a}_t)$.
      Store $(\mathbf{s}_t, \mathbf{a}_t, \mathbf{s}_{t+1}, \mathbf{g})$ into replay buffer $\mathcal{R}$.
      Sample transition $(\mathbf{s}, \mathbf{a}, \mathbf{s}', \mathbf{g}) \sim \mathcal{R}$.
      Compute reward $r = \log p_\psi(\mathbf{g} \,|\, \mathbf{s}, \mathbf{a}) - \mathbb{D}_{\mathrm{KL}}(q_\Delta(\cdot | \mathbf{s}_t, \mathbf{a}_t) \,\|\, p(\Delta))$.
      Compute $q(\Delta_t = 0 | \mathbf{s}, \mathbf{a})$ using Equation (11).
      Update $Q_\phi$ using Equation (17) and data $(\mathbf{s}, \mathbf{a}, \mathbf{s}', \mathbf{g}, r)$.
      Update $\pi_\theta$ using Equation (18) and data $(\mathbf{s}, \mathbf{a}, \mathbf{g})$.
      Update $p_\psi$ using Equation (19) and data $(\mathbf{s}, \mathbf{a}, \mathbf{g})$.
    **end for**
    **for** $t = 0, ..., H - 1$ steps **do**
      **for** $i = 0, ..., k - 1$ steps **do**
        Sample future state $\mathbf{s}_{h_i}$, where $t < h_i \le H - 1$.
        Store $(\mathbf{s}_t, \mathbf{a}_t, \mathbf{s}_{t+1}, f(\mathbf{s}_{h_i}))$ into $\mathcal{R}$.
      **end for**
    **end for**
  **end for**

---

and standard To bound the standard deviation outputted by the network, the standard-deviation tanh is multiplied by two with the standard deviation be between limited to between

**Reward normalization.** Because the different experiments have rewards of very different scale, we normalize the rewards by dividing by a running average of the maximum reward magnitude. Specifically, for every reward $r$ in the $i$th batch of data, we replace the reward with

$$\hat{r} = r/C_i$$

where we update the normalizing coefficient $C_i$ using each batch of reward $\{r_b\}_{b=1}^B$:

$$C_{i+1} \leftarrow (1 - \lambda) \times C_i + \lambda \max_{b \in [1, \ldots, B]} |r_b|$$

and $C_i$ is initialized to 1. In our experiments, we use $\lambda = 0.001$.

**Target networks.** To train our Q-function, we use the technique from Fujimoto et al. [12] in which we train two separate Q-networks with target networks and take the minimum over two to compute the bootstrap value. The target networks are updated using a slow, moving average of the parameters after every batch of data:

$$\bar{\phi}_{i+1} = (1 - \tau)\bar{\phi}_i + \times \phi_i.$$

In our experiments, we used $\tau = 0.001$.

**Automatic entropy tuning.** We use the same technique as in Haarnoja et al. [17] to weight the rewards against the policy entropy term. Specifically, we pre-multiply the entropy term in

$$\hat{V}(\mathbf{s}', \mathbf{g}) \approx Q_{\bar{\phi}}(\mathbf{s}', \mathbf{a}', \mathbf{g}) - \log \pi(\mathbf{a}' | \mathbf{s}'; \mathbf{g}),$$

by a parameter $\alpha$ that is updated to ensure that the policy entropy is above a minimum threshold. The parameter $\alpha$ is updated by taking a gradient step on the following function with each batch of data:

$$\mathcal{F}_\alpha(\alpha) = -\alpha \left( \log \pi(\mathbf{a} \,|\, \mathbf{s}, \mathbf{g}) + \mathcal{H}_{\mathrm{target}} \right)$$

and where $\mathcal{H}_{\mathrm{target}}$ is the target entropy of the policy. We follow the procedure in Haarnoja et al. [17] to choose $\mathcal{H}_{\mathrm{target}}$ and choose $\mathcal{H}_{\mathrm{target}} = -D_{\mathrm{action}}$, where $D_{\mathrm{action}}$ is the dimension of the action space.

**Exploration policy.** Because ODAC is an off-policy algorithm, we are free to use any exploration policy. It may be beneficial to add For the Ant and Sawyer tasks, we simply sample current policy. For the 2D Navigation task, at each time step, the policy takes a random action with probability 0.3 and repeats its

**Evaluation policy.** For evaluation, we use the mean of the learned policy for selecting actions.

**Hyperparameters.** Table 4 lists the hyperparameters that were shared across the experiments. Table 3 lists hyper-parameters specific to each environment.

**Table 3:** Environment specific hyper-parameters.

| Environment | horizon | $Q$-function and policy network sizes (hidden units) |
|---|---|---|
| Box 2D | 100 | [64, 64] |
| Ant | 100 | [400, 300] |
| Fetch Push | 50 | [64, 64] |
| Sawyer Push | 100 | [400, 300] |
| Sawyer Window | 100 | [400, 300] |
| Sawyer Faucet | 100 | [400, 300] |

**Table 4:** General hyperparameters used for all experiments.

| Hyperparameter | Value |
|---|---|
| # training batches per environment step | 1 |
| batch size | 256 |
| discount Factor | 0.99 |
| policy hidden activation | ReLU |
| $Q$-function hidden activation | ReLU |
| replay buffer size | 1 million |
| hindsight relabeling strategy | future |
| hindsight relabeling probability | 80% |
| target network update speed $\tau$ | 0.001 |
| reward scale update speed $\lambda$ | 0.001 |