# OpenReview forum: "Outcome-Driven Reinforcement Learning via Variational Inference"
_NeurIPS.cc/2021/Conference — NeurIPS 2021 Poster_

### Official Review · Reviewer_4Y6i · 2021-07-08

**Rating:** 8
**Confidence:** 4

**Summary:**

This paper derives goal based reinforcement learning as a task of variational inference.
They propose a novel bellman operator and learning scheme and provide extensive evaluations across various complex tasks.
Empirical evidence backs the theoretical results showing that this method works in practice.

**Limitations And Societal Impact:**

-

**Main Review:**

I find this line of work to be very interesting and important.
The problem formulation and paper are well written and the experiments show the ability of this approach to solve problems in practice.

Two minor comments:

1) I'd like to see a clean analysis in a tabular setting, while deep learning experiments are impressive they usually require a lot of "hacks" and it isn't always clear why the method works better/worse than prior work. This is not to say the method does not achieve impressive results nor that the inability/ability to break SOTA results in deep RL are sufficient/relevant for acceptance/rejection.

2) The numerical table is a bit misleading. I know that the common practice recently is to bold methods with the better mean performance. I'd urge the authors to re-think this broken approach and highlight all results that *might* be optimal (overlapping confidence intervals at the top).

**Time Spent Reviewing:**

4

---

> ### Author Response · Authors · 2021-08-10
> **Response to Reviewer 4Y6i**
>
> **General comments**
>
> Thank you for the detailed review. We were happy to read that you find “this line of work to be very interesting and important” and the “problem formulation and paper [...] well written.”
>
> We hope that comments below address your concerns, and we would be more than happy to answer any follow-up questions you may have.
>
> **1. “I'd like to see a clean analysis in a tabular setting, while deep learning experiments are impressive they usually require a lot of "hacks" and it isn't always clear why the method works better/worse than prior work.”**
>
> We appreciate the question. Theorem 2 studies the proposed algorithm’s convergence to an optimal policy in tabular settings. The theorem states that in tabular settings, alternating between outcome-driven policy evaluation (ODPE) and outcome-driven policy improvement (ODPI) leads to an optimal policy (and to an optimal dynamic discount factor).
>
> **2. “The numerical table is a bit misleading. I know that the common practice recently is to bold methods with the better mean performance. I'd urge the authors to re-think this broken approach and highlight all results that might be optimal (overlapping confidence intervals at the top).”**
>
> We share the sentiment and will remove using boldface in the results table for cases where there is no statistically significant difference.

---

### Official Review · Reviewer_DJiQ · 2021-07-15

**Rating:** 6
**Confidence:** 4

**Summary:**

The paper proposes ODAC, a new RL algorithm for goal-directed tasks. Using a variational inference approach, ODAC automatically infer a dense environment reward from the goal and derives a probabilistic Bellman backup operator. The latter is used to develop an off-policy RL algorithm. The paper presents detailed theoretical derivation and empirical results on a suite of environments.


**Limitations And Societal Impact:**

As the proposed method is quite complex, a general limitation is the difficulty in understanding the actual working of the method, and its advantages over the baseline methods. Several related issues are discussed above, such as why the learned dynamic model performs worse than a fixed one, whether HER/goal relabeling is a critical component for ODAC as well, and how much of the observed performance improvements comes from restricting goal states during training.

**Main Review:**

The paper is well presented. The theoretical derivation is detailed and easy to follow. In addition, the derived practical reward appears intuitive, including a  log-likelihood term for reaching the goal in the next step and a KL term for long-term reachability. The empirical results shows improved performance over existing methods.

**Presentation**: There appears to be some inconsistencies in the presentation of the method. For the reward function, Eq. 10 is different from Eq. 18 (the subscript $t$ in the KL term is off by 1), and again different from Alg. 2 in the appendix (the coefficient $q(\Delta=1)$ for the log-likelihood $\log p(g|s, a)$ is missing). Could the authors please clarify which reward function is used in practice. If approximation is used, please provide some intuition/motivation for the approximation. For the value function, Eq. 8 and Eq. 13 have a different sign for the KL term. It appears that Eq. 13 has the incorrect sign.

**Method**: Alg 1 and Alg 2 in the appendix are different. It appears that the general idea of hindsight experience replay (HER) is part of Alg 2, but not part of Alg 1. It is important to highlight if HER plays a significant role in the practical algorithm, and study it in the ablation (e.g. ODAC without HER vs ODAC with HER).

**Empirical Evaluation**: The experiment setting from the paper appears to restrict the goal to a single state (i.e. no oracle goal sampling) during training. It is difficult to understand why this setting is chosen, given that oracle goal sampling appears easily implementable, and was used by several previous methods originally. Could the authors provide the comparison results for more environments (in addition to Fetch Push in the appendix), allowing oracle goal sampling? This would allow a more comprehensive comparison with the baseline methods and a better understanding of the ODAC’s advantages over existing methods.

Table 1 appears to also suggest that the learned dynamic model $p_d $  performs noticeably worse than a fixed one, wrt to the policy learning. Specifically, the learned dynamic model obtains worse mean performance and noticeably larger standard deviation in 4 out of 6 tasks. Could the authors provide some intuition as to why this is the case, since this is rather unexpected. I cannot support the current interpretation that the proposed method is insensitive to the dynamic model.

Minor:
In A.15, t should be t’ in the summation sign. Line 808 has a missing latex reference.

------------- After responses ---------------------
My concerns are mostly addressed. I have revised my score.

**Time Spent Reviewing:**

7

---

> ### Author Response · Authors · 2021-08-10
> **Response to Reviewer DJiQ**
>
> **General comments**
>
> Thank you for the detailed review and for highlighting that the overall method seemed complex and that it was unclear how each component of the method contributed to the overall performance of the method. Below, we clarify that the overall algorithm only has two simple differences from the standard soft actor-critic method and discuss how our ablations and comparisons test the impact that these two factors have on performance. We have also added a comparison of ODAC with and without HER, which shows that HER is not essential. Lastly, we also address the concern regarding the oracle goal sampling.
>
> We believe this has addressed all of the issues raised in your review, but please let us know if any other concerns remain.
>
> **1. “The experiment setting from the paper appears to restrict the goal to a single state (i.e. no oracle goal sampling) during training. It is difficult to understand why this setting is chosen, given that oracle goal sampling appears easily implementable, and was used by several previous methods originally. [...] Could the authors provide the comparison results for more environments (in addition to Fetch Push in the appendix), allowing oracle goal sampling?”**
>
> ​​Prior work is generally split on whether it assumes oracle goal sampling or not. Some work does (Andrychowicz et al., 2016), while many works do not assume it, and instead treat discovering goals as an important part of the exploration problem (Teh et al., 2017 [T2017], Pong et al., 2019 [P2019]). We believe the latter is an important problem setting: In many real-world applications, we do not have oracle access to the set of valid states any more than we have oracle access to the dynamics---everything must be learned via interaction with the environment. This setting is also more challenging, because oracle goal sampling facilitates significantly easier exploration.
>
> ​​Nevertheless, we opted to include oracle goal sampling for the Fetch Push environment because it is taken from prior work that also does goal sampling, which is not the case for the other environments. While we were unable to add oracle sampling experiments during the rebuttal due to practical limitations (the author who ran the experiments was away from the office during the rebuttal period), we will aim to add them in the updated paper.
>
>
> **2. “Specifically, the learned dynamic model obtains worse mean performance and noticeably larger standard deviation in 4 out of 6 tasks. Could the authors provide some intuition as to why this is the case, since this is rather unexpected.”**
>
> We agree that these results are rather unexpected. Our main hypothesis for why this might be happening is that the cost associated with the instability introduced by a changing reward function (as the model of the environment’s transition dynamics is being updated) may---on some environments---outweigh the benefit of having a more accurate model of the environment’s transition dynamics. This would be more likely to be the case for environments where a static “distance-based” reward function is sufficient. We will aim to add additional analysis to understand this issue, for example by using a pretrained model that does not change over the course of training as a baseline.
>
> We would emphasize, however, that at a high level such tradeoffs are not unexpected. This can be interpreted as an instance of a bias/variance tradeoff, which is common in machine learning: the learned model has less bias, but higher variance, while the fixed model is biased, but has less variance (since it has no learned parameters).
>
>
> **3. “As the proposed method is quite complex, a general limitation is the difficulty in understanding the actual working of the method”**
>
> Although our derivation is somewhat involved in order to provide a clear justification of the method as optimizing a variational objective for Type II maximum likelihood estimation, the final method is quite simple. The actual algorithm, implementation-wise, just corresponds to an actor--critic algorithm with rewards provided by the log-likelihood of reaching the goal under a certain dynamics model for a given state--action pair, and the discount factor is provided by a closed-form equation. The ablation studies show that using the derived discount factor consistently improves performance. Moreover, although it is unclear whether it is better to use a learned or fixed dynamics model, the ablations do show that---regardless of the dynamics model---ODAC outperforms SAC with hand-crafted rewards and fixed discount factor.
>
> The method is comparable in complexity of implementation to standard model-free actor--critic methods.
>
>
> **4. “Alg 1 and Alg 2 in the appendix are different. It appears that the general idea of hindsight experience replay (HER) is part of Alg 2, but not part of Alg 1. It is important to highlight if HER plays a significant role in the practical algorithm, and study it in the ablation (e.g. ODAC without HER vs ODAC with HER).”**
>
> Thank you for raising this point. Algorithm 2 is correct and used in practice. We will clarify this in the main paper, but we would like to emphasize that for our proposed algorithm, we found that using HER does not in fact have a noticeable impact on the performance at the end of training but mostly appears to affect the number of epochs needed to converge to an optimal policy.
>
> In the figure here [(anonymized link)](https://drive.google.com/file/d/1ZMGDgdIJRWT7_jto3aPOv5qgHtiufxxG/view?usp=sharing), we ablated the use of HER on the ant environment and the performance is similar to that of SAC with a ground-truth reward, demonstrating that HER is not a critical component of ODAC. We will include these results in the final version of the paper.
>
>
> **5. “For the reward function, Eq. 10 is different from Eq. 18 (the subscript t in the KL term is off by 1), and again different from Alg. 2 in the appendix (the coefficient $q(\Delta=1)$ for the log-likelihood $\log⁡ p(g | s,a)$ is missing). Could the authors please clarify which reward function is used in practice.”**
>
> Thank you for catching these typos. Equation (10) is correct, and the subscripts in the KL divergences shown in Equation (18) and in the reward in Algorithm 2 should be $t+1$. This reward is used in practice. The reason for the $+1$ increment in the index is the application of Lemma 5 when going from Equation A.36 to Equation A.37 in Appendix A.2.
>
> **6. “For the value function, Eq. 8 and Eq. 13 have a different sign for the KL term. It appears that Eq. 13 has the incorrect sign.”**
>
> Thank you for catching this typo. You are correct, Equation (13) has the incorrect sign.
>
> --
>
> References:
>
> [T2017] Yee Whye Teh, Victor Bapst, Wojciech Marian Czarnecki, John Quan,
> James Kirkpatrick, Raia Hadsell, Nicolas Heess, Razvan Pascanu. Distral: Robust Multitask Reinforcement Learning. Advances in Neural Information Processing Systems 30, 2017.
>
> [P2019]  Vitchyr H. Pong, Murtaza Dalal, Steven Lin, Ashvin Nair, Shikhar Bahl, Sergey Levine. Skew-Fit: State-Covering Self-Supervised Reinforcement Learning. Proceedings of the 37th International Conference on Machine Learning, 2019.

---

> > ### Comment · Reviewer_DJiQ · 2021-08-16
> > **Question on HER and goal sampling**
> >
> > I thank the authors for the clarifications. I have a follow-up question on HER and goal sampling.
> >
> > To my understanding, HER in general allows different methods to sample new goals (which are also valid goals since they are reachable by the agent). Is this different from "oracle goal sampling" as described in the paper?

---

> > > ### Author Response · Authors · 2021-08-17
> > > **Clarification about HER and goal sampling**
> > >
> > > Thank you for the follow-up question!
> > >
> > > ### Oracle Goal Sampling
> > >
> > > HER involves sampling goals in two different ways: First, during exploration, HER samples goals from the environment are then provided to a goal-directed policy. Second, when updating the policy and/or Q-function, HER performs “goal relabeling” by sampling states from the replay buffer and converting them into goals. Could you clarify whether by “sample new goals,” you are referring to sampling new goals during exploration or for relabeling?
> > >
> > > Andrychowicz et al. (2018) (where HER was introduced) explore different ways to sample goals for __**both**__ relabeling and exploration. For relabeling, they found that the “future-style” relabeling method performed the best. We used this future-style relabeling for our method and for the methods that we compared to, ensuring that we have a fair comparison.
> > >
> > > For exploration, Andrychowicz et al. (2018) explore the benefits of HER when either (1) using a single, fixed goal during exploration (see Section 4.3 of Andrychowicz et al. (2018)) or (2) using what we call “oracle goal sampling” during exploration. By "oracle goal sampling" we mean that during exploration, a new goal is sampled each episode from a uniform distribution over the set of __**all reachable**__  goals in the environment. As such, oracle goal sampling requires knowledge of the environment to sample __several__ reachable goals. For example, in the 2D box experiment (Figure 3a), points inside the grey block in the center are __not__ reachable goal states, and this additional information must be available when performing oracle goal sampling.
> > >
> > > Rather than assuming that we can perform oracle goal sampling (which is challenging in complex real-world environments), we assume that the agent is provided with a single goal at the beginning of exploration as part of the problem formulation.
> > >
> > > One potential source of confusion is whether “relabeling” in HER can only resample goals from the exploration goal distribution. We clarify that this is not the case. The future-style goal-relabeling procedure used in HER is orthogonal to the goal-sampling procedure used for exploration.
> > >
> > > In summary, ODAC and the methods that we compare to use the future-style relabeling during training, and do not use oracle goal sampling.
> > >
> > > ### Additional Comments on Goal Sampling & Experiments
> > >
> > > We clarify that the ablation posted in our previous message, studies whether or not it is important to perform future-style relabeling. “ODAC with HER” is the same as ODAC in our paper, which uses future-style relabeling, and “ODAC without HER” removes the future-style relabeling. In both cases, we do not use oracle goal sampling. The results here ([anonymized link](https://drive.google.com/file/d/1ZMGDgdIJRWT7_jto3aPOv5qgHtiufxxG/view?usp=sharing)) demonstrate that our method does not require the future-style relabeling to learn.
> > >
> > > If you are wondering what happens when we add the assumption that we can perform oracle goal sampling during exploration, we provide results with oracle goal sampling in Appendix C (Figure 6). They show that UVD and ODAC are similar in performance and that both outperform other methods. Moreover, they also show that there is an improvement in performance when oracle goal sampling is used, as compared to the setting where only a single goal is given as part of the problem formulation, which indicates that (as expected) oracle goal sampling makes the problem easier. We suspect that this improvement may be due to oracle goal sampling implicitly providing a curriculum for learning: when the only exploration goal is the final task goal, learning is difficult, but when the goals are selected randomly from the known set of possible goals by the oracle, some of them might be much easier. Hence, if we care about settings in which oracle goal sampling is not feasible, using oracle goal sampling may lead to a misleadingly good performance, and hence we avoid this setting in our main experiments.
> > >
> > > Please let us know if there are any remaining questions. We hope that we have addressed your questions and that you will consider revising your score.

---

> > > > ### Comment · Reviewer_DJiQ · 2021-08-18
> > > > **Thank you**
> > > >
> > > > Thanks for the explanation. I have revised my score.

---

> > > > > ### Author Response · Authors · 2021-08-18
> > > > > **Thank you**
> > > > >
> > > > > Thank you for engaging with our response and for revising your score. It's much appreciated.

---

### Official Review · Reviewer_SSfs · 2021-07-15

**Rating:** 7
**Confidence:** 3

**Summary:**

This paper combines the recent work on viewing RL as Probabilistic Inference and the goal-conditioned RL setting. The proposed approach assigns rewards based on the likelihood of reaching the desired state-action pair and propagates this via learning a dynamic discount function based on time elapsed in the episode. The authors propose solving the objective via variational inference and demonstrate this leads to a probabilistic policy evaluation operator. An actor-critic approach is then taken to perform approximate policy iteration.  The authors show the advantage of their methods on high-dimensional continuous control tasks.


**Limitations And Societal Impact:**

yes

**Main Review:**

### Originality:

The combination of control as inference take on the off-policy goal-driven RL is novel to the best of my knowledge and is also motivated sufficiently. I have some questions regarding the approach, in particular:

- The connection between viewing the log-likelihood of reaching particular state-action pair as the reward has been explored also in other works that also take a probabilistic inference view on different RL settings and propose solving it via variational inference such as [1].  Also, the authors mention that the difference between [3] and this work is that [3] requires on-policy samples whereas the proposed approach is off-policy. Can the authors elaborate on why extending [3] to off-policy is particularly tough or challenging? Even though the proposed approach is based on off-policy updates it is still online, so a comparison could be made. What advantage does the proposed approach provides compared to an off-policy version of [3] or the general approach in [1]?

- The mechanism through which the rewards/values are propagated is via learning a state-dependent discounting function. I think the authors should also mention some of the other works that have look at a similar approach for learning the discount factor such as [2,5].

### Quality:

The work is technically sound and empirically supported with sufficient ablation studies. However, I have a few questions regarding the choice of baselines. The authors motivate in Sec 5 that the main difference between the prior work on control as probabilistic inference assumes that the manual reward is required ($r_{\text{manual}}$) whereas the main advantage of the proposed approach is that instead of requiring manual reward function as input it requires manual goals as input. Based on this, I would have expected some of the goal-based version of the existing control as probabilistic inference as some of the baselines. For instance, why was VIREL [4] excluded from the baselines? The VIREL methodology should be extendable to this setting? If not, some discussion of why the other work prior work can’t be extended to this setting should be provided.

### Clarity:
The paper is mostly clear. The paper references outcomes throughout the text without formally defining them. It wasn’t clear to me what is the difference between goal-directed and outcome-driven? Does outcome-driven signify the probabilistic nature of the original goal-directed setting?


### Significance:

The paper looks at an important problem setting and the results seem to demonstrate that the proposed approach is able to achieve its goal. I have some reservations about the originality and the need for this approach as mentioned in the points above. I think if the authors can shed some light on the questions raised above, then that will be helpful in raising the usefulness of this work to the community.

### References:

- [1] Weber, Théophane, et al. "Credit assignment techniques in stochastic computation graphs." The 22nd International Conference on Artificial Intelligence and Statistics. PMLR, 2019.
- [2] Pitis, Silviu. "Rethinking the discount factor in reinforcement learning: A decision theoretic approach." Proceedings of the AAAI Conference on Artificial Intelligence. Vol. 33. No. 01. 2019.
- [3] Justin Fu, Avi Singh, Dibya Ghosh, Larry Yang, and Sergey Levine. Variational inverse control 387 with events: A general framework for data-driven reward definition. In S. Bengio, H. Wallach, 388 H. Larochelle, K. Grauman, N. Cesa-Bianchi, and R. Garnett, editors, Neural Information 389 Processing Systems (NeurIPS), pages 8538–8547. 2018.
- [4] Matthew Fellows, Anuj Mahajan, Tim G. J. Rudner, and Shimon Whiteson. VIREL: A Varia384 tional Inference Framework for Reinforcement Learning. In Advances in Neural Information 385 Processing Systems 32, 2019.
- [5] Xu, Z., van Hasselt, H. P., and Silver, D. Meta-gradient reinforcement learning. In Advances in neural information processing systems, pp. 2396–2407, 2018.




**Time Spent Reviewing:**

4

---

> ### Author Response · Authors · 2021-08-10
> **Response to Reviewer SSfs**
>
> **General comments**
>
> Thank you for your thoughtful and detailed review. We were pleased you thought our approach to “off-policy goal-driven RL is novel [...] and [...] motivated sufficiently” as well as “technically sound and empirically supported with sufficient ablation studies.”
>
> We hope our clarifications below address your concerns regarding the significance of the technical contributions of the paper and would be more than happy to answer any follow-up questions you may have.
>
>
> **1. “Can the authors elaborate on why extending [3] to off-policy is particularly tough or challenging?”**
>
> The derivation of VICE does not enable off-policy learning because VICE does not perform inference over $T$, the time when the goal is reached. Instead, VICE performs inference over the states and actions and marginalizes out the random variable $T$. By doing so, the (approximate) posterior distribution over every state and action becomes dependent, and so the resulting inference cannot be expressed recursively in terms of expectations over actions and states at only a single time step. Therefore, VICE [3] uses an on-policy policy-gradient method and relies on trajectories sampled according to the current policy.
>
> Our method can be seen as the ingredient to making VICE off-policy. In Appendix A.2, we show that by treating the time at which the outcome is achieved as a random variable and inferring a variational distribution over it, we are able to obtain a recursively-defined objective expressed in terms expectations over actions and states at only a single time step (see Theorem 1, derived in Appendix A.2).
>
>
> **2. “What advantage does the proposed approach provides compared to an off-policy version of [3] or the general approach in [1]”**
>
> In addition to allowing off-policy learning, our work addresses further shortcomings of VICE: (i) the derivations in VICE contain subtle technical errors (specifically, the Q-function derived on page 13 of the VICE appendix is only valid for probability mass functions and not for probability density function, but is incorrectly used in continuous state-space environments) not present in our work (our variational objective is valid for both discrete and continuous state spaces) and (ii) VICE introduces discount factors in an ad-hoc manner, whereas we show that discount factors emerge naturally when defining the probabilistic model in terms of a trans-dimensional distribution and performing variational inference over the time at which the outcome is achieved, $T$ (see Section 3.2). On top of these contributions, we also derive an optimal dynamic discount factor (Proposition 2) and provide a proof of convergence to an optimal policy under the proposed algorithm for tabular settings---neither of which was provided in [3].
>
> Regarding [1], could you clarify what kind of comparison you would like us to draw? Corollary 2 in our appendix describes the relationship to SAC, which is an RL algorithm discussed in [1] and most closely-related to our work. Otherwise, [1] and our paper are quite distinct algorithmically and in terms of their derivation. Further clarification would be much appreciated.
>
>
> **3. “I would have expected some of the goal-based version of the existing control as probabilistic inference as some of the baselines.”**
>
> Thank you for raising this point. Our empirical evaluation does in fact include this baseline: The method referred to as only “HER” in Figure 4 is in fact SAC with hindsight experience replay (HER) and is based on the code provided publicly in the RLkit GitHub repo ([link to examples/her/her_sac_gym_fetch_reach.py in RLkit repo](https://github.com/rail-berkeley/rlkit/blob/master/examples/her/her_sac_gym_fetch_reach.py)). We will update the caption and the description in Section 6 and appreciate your bringing this to our attention.
>
>
> **4. “For instance, why was VIREL [4] excluded from the baselines? The VIREL methodology should be extendable to this setting? If not, some discussion of why the other work prior work can’t be extended to this setting should be provided.”**
>
> The same way SAC was used with HER in our experiments (as described above), it would be possible to use VIREL with HER. We appreciate the suggestion and are working on including VIREL with HER as a baseline.**
>
>
> **5. “It wasn’t clear to me what is the difference between goal-directed and outcome-driven? Does outcome-driven signify the probabilistic nature of the original goal-directed setting?”**
>
> Your interpretation is correct. The term outcome is chosen to signify the probabilistic nature of the goal-directed setting. Specifically, in lines 94--96 (Section 3.1) we introduce an outcome as a specific state realization $g \in \mathcal{S}$ at a specific time step $t^{\ast}$ when starting from an initial state $s_{0}$. In other words, we define a (desired) outcome as a state realization $S_{t^{\ast}} = g$, where $S_{t^{\ast}}$ is a random variable. In Section 3.2, we generalize this notion to cases where the time $T$ at which the state realization $S_{T} = g$ occurs is unknown and itself treated as a random variable.
>
> --
>
> References:
> [1] Weber, Théophane, et al. “Credit assignment techniques in stochastic computation graphs.” The 22nd International Conference on Artificial Intelligence and Statistics. PMLR, 2019.
>
> [3] Justin Fu, Avi Singh, Dibya Ghosh, Larry Yang, and Sergey Levine. Variational inverse control with events: A general framework for data-driven reward definition. Neural Information 389 Processing Systems (NeurIPS), pages 8538–8547, 2018.

---

> > ### Comment · Reviewer_SSfs · 2021-08-19
> > **Thanks**
> >
> > Thanks for the response. I appreciate the clarifications regarding VICE and HER baseline. My concerns regarding the baselines have been mostly addressed. I will wait for the comparisons with VIREL before revising the score.

---

> > > ### Author Response · Authors · 2021-08-21
> > > **VIREL Comparison Results**
> > >
> > > We have completed the comparison to VIREL. Thank you for your patience.
> > >
> > > We used [publicly available VIREL code released by the authors](https://github.com/AnujMahajanOxf/VIREL/tree/master/VIREL_code), consulted with the authors on how to tune the method, and got their assistance in implementing it, so we are confident that the implementation is correct. The VIREL paper has two variants of the algorithm: one with an automatically tuned “beta” parameter that acts as a dynamic weight on the entropy term, and another with a static entropy term weight. We ran both versions of their algorithm and also tuned the reward scale by choosing the best of either 1 or 1/(average reward magnitude). We then reported the best performing variant.
> > >
> > > We plot the learning curves of ODAC and VIREL with HER across all six environments. The other methods from the paper are omitted for clarity. See the results in this anonymous link: https://imgur.com/a/KtHnQhf
> > >
> > > We found that ODAC consistently outperforms VIREL with HER across every task. Please let us know if you have any remaining questions.

---

> > > > ### Comment · Reviewer_SSfs · 2021-08-23
> > > > **Thank you!**
> > > >
> > > > Thanks for the response. My concerns regarding the baselines have been alleviated now. I've updated my score to reflect that.

---

> > > > > ### Author Response · Authors · 2021-08-23
> > > > > **Thank you!**
> > > > >
> > > > > Thank you for engaging with our response, for your helpful suggestions, and for revising your score. It's much appreciated.

---

### Official Review · Reviewer_3mzu · 2021-07-16

**Rating:** 7
**Confidence:** 3

**Summary:**

This paper proposes to formulate goal-conditional reinforcement learning as a variational inference problem, termed as outcome-driven RL, and shows that the variational objective naturally generates shaped reward signals for learning. The variational objective results in an off-policy temporal-difference algorithm. Experimental results in robotic control tasks show the advantages of the novel algorithm.



**Limitations And Societal Impact:**

The authors mentioned the limitations of the proposed work and potential future work to improve it. The authors did not comment on potential societal impacts.

**Main Review:**

The problem of goal-conditional RL is an active area of research and many recent progresses have been made. This work proposes a novel formulation of the problem and demonstrates that such a formulation leads to an algorithm that outperforms prior methods. The paper is well-written with clarity. The related work section does a good job of overviewing relevant literature in both RL and control. The experiments cover six task domains with different characteristics and demonstrate ODAC performs comparably or outperforms prior methods in most of the tasks. An ablation study is also performed to provide insights into the limitations of ODAC and show that ODAC is not sensitive to the accuracy of the dynamics model.





**Time Spent Reviewing:**

2

---

> ### Author Response · Authors · 2021-08-10
> **Response to Reviewer 3mzu**
>
> We appreciate the positive review and are happy that you found the paper well-written and thought it “does a good job of overviewing relevant literature in both RL and control.” We would be more than happy to answer any questions or respond to any comments you may have during the discussion phase.

---

### Decision · Program_Chairs · 2021-09-27

**Decision:**

Accept (Poster)

**Comment:**

The paper proposes a new variational inference approach for goal-based reinforcement learning. The author's goal is to more directly tackle reinforcement learning problems where the goal is to reach a certain outcome, without resorting to manual reward shaping. Similarly to some previous work, they define an inference objective which is maximize using techniques from variational optimization. Different from such works, in their inference formulation they condition on the goal being reached, rather than using the reward function as pseudo-likelihood.

The reviewers consider the considered problem (goal-conditioned RL) important and very relevant for the machine learning community.
The resulting approach was considered novel by the reviewers, especially the extension of a framework similar to VICE [3] to an off-policy setting.
The reviewers also considered the paper technically sound and sufficiently well supported by experiments and ablations (that show the method outperforms e.g. SAC, another method in this space).
Also, the reviewers consider the paper well-written.